# A Tale of Two Structures:
# Do LLMs Capture the Fractal Complexity of Language?

**Ibrahim Alabdulmohsin** [1]  **Andreas Steiner** [1]

## Abstract

Language exhibits a fractal structure in its information-theoretic complexity (i.e. bits per token), with self-similarity across scales and long-range dependence (LRD). In this work, we investigate whether large language models (LLMs) can replicate such fractal characteristics and identify conditions—such as temperature setting and prompting method—under which they may fail. Moreover, we find that the fractal parameters observed in natural language are contained within a *narrow* range, whereas those of LLMs' output vary widely, suggesting that fractal parameters might prove helpful in detecting a non-trivial portion of LLM-generated texts. Notably, these findings, and many others reported in this work, are robust to the choice of the architecture; e.g. Gemini 1.0 Pro, Mistral-7B and Gemma-2B. We also release a dataset comprising over 240,000 articles generated by various LLMs (both pretrained and instruction-tuned) with different decoding temperatures and prompting methods, along with their corresponding human-generated texts. We hope that this work highlights the complex interplay between fractal properties, prompting, and statistical mimicry in LLMs, offering insights for generating, evaluating and detecting synthetic texts.

## 1. Introduction

The information-theoretic complexity of language (i.e., its bits or "surprise") has been shown to exhibit both self-similarity and long-range dependence (LRD) (Alabdulmohsin et al., 2024). In simple terms, a stochastic process is considered self-similar if its statistical properties remain consistent across different scales, regardless of the level of magnification applied. A well-known example of such behavior

is found in Ethernet traffic (Crovella & Bestavros, 1995; Leland et al., 1994; Paxson & Floyd, 1995; Willinger et al., 1997), where self-similarity manifests as burstiness across all time scales, thereby impacting the design of network device buffers (Leland & Wilson, 1991). In language, such self-similarity is attributed to its recursive structure (Altmann et al., 2012). On the other hand, a stochastic process is called long-range dependent (LRD) if its future is influenced by the *distant* past, with no particular characteristic context length.

Self-similarity and long-range dependence (LRD) can be quantified using the Hölder and Hurst exponents, respectively. The Hölder exponent, which we denote by S for self-similarity, characterizes the rate of decay in the autocorrelation function, with *smaller* values of S indicating a *more* significant self-similar structure (heavier tail) (Watkins, 2019). By contrast, larger values of the Hurst exponent $H \gg 0.5$ indicate more dependence across time (Hurst, 1951). We refer the reader to Appendix B for the exact definitions of these quantities. A natural question that arises, next, is: How different are S and H in LLM-generated texts from natural language? In this work, we investigate this question in depth, aiming to identify conditions—such as temperature settings, instruction-tuning, prompting and model size—that influence an LLM's ability to replicate such fractal characteristics.

Before doing that, however, let us consider some arguments for why LLMs may or may not be capable of replicating the fractal structure of language. One argument for why they should be capable of doing so lies in the chain rule of probability. LLMs are reasonably calibrated at the token level (Kadavath et al., 2022), implying that auto-regressive decoding should theoretically capture the structure of natural language as long as token-level probability scores remain well-calibrated. By the chain rule, each subsequent token's probability is conditioned on the prior tokens, and this process should *ideally* reflect the self-similarity and long-range dependence inherent in language.

However, this idealized view may not hold in practice. A critical issue can arise, for example, from the mismatch between how LLMs are trained and how they are used at inference, as pointed out by (Bachmann & Nagarajan, 2024).

[1]Google Deepmind, Zürich, Switzerland. Correspondence to: Ibrahim Alabdulmohsin <ibomohsin@google.com>.

*Proceedings of the 42nd International Conference on Machine Learning*, Vancouver, Canada. PMLR 267, 2025.

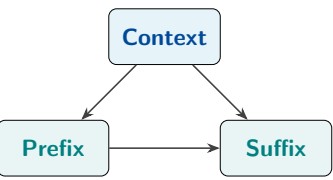

*Figure 1.* A causal model we consider, in which a latent "context" generates a prefix and both produce a suffix.

During training, LLMs use teacher-forcing, where the correct previous tokens are always provided. This ensures that errors do not accumulate during training, but during inference, models must rely on their own predictions, leading potentially to compounding errors that can distort the fractal structure. Formally speaking, whereas LLMs can be well-calibrated at the *next token* level, their ability to accurately predict the distribution over longer sequences of tokens might degrade during inference.

Another potential challenge lies in how language is generated by humans. Humans typically generate texts by first conceptualizing an underlying "*context*" and then constructing sentences based on it, rather than improvising one token at a time with no regard to the overall intent. This is captured by the causal model in Figure 1, where a latent "context" generates a "prefix" (beginning of text), and, in conjunction with this prefix, both produce a "suffix" (continuation).

Formally, under this hypothetical causal model, prompting corresponds to an *interventional* (or "**do**") query. Let $\mathcal{V}$ be a finite vocabulary of tokens, where texts correspond to finite sequences $x \in \mathcal{V}^N$. Now, divide $x$ into a prefix $x_{0:n-1}$ and a suffix $x_{n:N}$. Assuming for simplicity that the set of possible latent contexts is finite, one classical result in causal inference states that the "interventional" distribution is given by (see Equation 4.5 in (Neal, 2020)):

$$p\left(\mathrm{suf} \mid \mathbf{do}(\mathrm{pre})\right) = \sum_{c \in \mathcal{C}} p\left(\mathbf{c} = c\right) \cdot p\left(\mathrm{suf} \mid \mathrm{pre}, \mathbf{c} = c\right). \quad (1)$$

A language model, by contrast, learns the "conditional" distribution, which by marginalization is:

$$p\left(\mathrm{suf} \mid \mathrm{pre}\right) = \sum_{c \in \mathcal{C}} p\left(\mathbf{c} = c \mid \mathrm{pre}\right) \cdot p\left(\mathrm{suf} \mid \mathrm{pre}, \mathbf{c} = c\right). \quad (2)$$

We provide an example that illustrates such differences in Appendix D. Under this hypothesis, LLMs may struggle to fully replicate the fractal nature of human language because the conditional distribution uses $p\left(\mathbf{c} = c \mid \mathrm{prefix}\right)$ instead of the marginal $p\left(\mathbf{c} = c\right)$, which should be used if prompting corresponds to an intervention. To account for this, we investigate the impact of availing various amounts of contextual information in the prompt. These prompting strategies range from minimal cues (e.g. few keywords) to detailed prompts (e.g. summaries or ordered excerpts). Interestingly, increasing the information density in the prompt does not always improve fractal characteristics, as shown in Figure 7. In fact, the relationship for self-similarity seems to exhibit a "double descent", where providing a summary in the prompt generates texts that are *less* similar to natural language than providing either an unordered set of keywords (less information) or an ordered set of excerpts (more).

Do fractal parameters in LLM-generated texts differ substantially from natural language? Our findings suggest that they partially do. Specifically, the range of fractal parameters in natural language is contained with a *narrow* range, whereas those of LLMs' output vary widely, as demonstrated for the Hölder exponent S in Figure 10. Large values of S indicate less self-similar structure (i.e. lack of rich details) so we expect LLMs to fail occasionally to produce texts with small values of S. This is indeed what we observe.

To conduct our study, we build a dataset comprising of over 240,000 LLM-generated articles. These differ by the model that generated them, the contextual information provided in the prompt, the decoding temperature, and the data domain (e.g. science, news, etc). To facilitate research in domains, such as detecting LLM-generated contents, we make this dataset public. The data card is provided in Appendix E and samples from the data are shown in Appendix G. In addition, we show that our main conclusions continue to hold using the RAID dataset (Dugan et al., 2024), which contains generated texts by 11 models (e.g. ChatGPT, Cohere, Llama 2, etc) in domains such as Reddit and reviews.

**Statement of Contribution.** In summary, we:

1. provide a comprehensive analysis of how various factors—such as decoding temperatures, instruction-tuning and model size—affect the ability of LLMs to replicate the fractal properties of natural language.

2. investigate how prompting affects the fractal structure of texts, showing evidence for a non-monotone behavior. We demonstrate that the range of fractal parameters in natural language is much narrower than in LLM's. This shows that fractal parameters might prove useful in identifying some (but not all) synthetic texts. We also connect the differences in fractal parameters to the quality of the texts.

3. show these results hold across a variety of model architectures, demonstrating the generality of our findings and making them relevant for diverse LLM families.

4. release a dataset containing over 240,000 articles generated by various LLMs (both pretrained and instruction-tuned) across various settings (e.g. data domain and temperature).

## 2. Related Works

Several studies have looked into the statistical properties of LLM-generated texts. For instance, (Guo et al., 2023) analyzed ChatGPT's responses in comparison to human-generated text and found that ChatGPT-produced outputs tend to have lower log-perplexity scores. Based on these findings, the authors developed detection systems for identifying LLM-generated content, concluding that short texts are more challenging to detect than longer documents. Similarly, Tulchinskii et al. (2023) propose using the intrinsic dimension of the data manifold as a metric that distinguishes natural language from LLM-generated texts.

The observation that LLM-generated texts exhibit lower log-perplexity scores has been utilized by several other works for detecting such content, including (Solaiman et al., 2019; Gehrmann et al., 2019; Ippolito et al., 2020; Vasilatos et al., 2023) and (Yang et al., 2023), among others. This raises the question of which models are most suitable for evaluating text perplexity. In (Mireshghallah et al., 2024), the authors study this issue and suggest that smaller models are more effective! Additionally, (Hans et al., 2024) proposed normalizing the average log-perplexity score using a cross-PPL metric calculated across two different models before applying a global detection threshold.

However, such approaches, which are based on $1^{st}$-order statistics of log-perplexity scores, are not always effective (Hans et al., 2024). For instance, methods, such as DetectGPT, which incorporate $2^{nd}$-order information by estimating the curvature of the loss (Mitchell et al., 2023), have been found to yield better results.

Our work contributes to this line of research by focusing primarily on fractal parameters. As argued in (Meister & Cotterell, 2021), the evaluation of LLMs should go beyond log-perplexity and also consider how well LLMs capture other "statistical tendencies" observed in natural language. Our study has a similar goal. We conduct a comprehensive quantitative analysis involving a range of model architectures, decoding temperatures, and prompting methods, and we explore the differences between pretrained and instruction-tuned models as well, among other considerations.

Although our study may have implications for detecting LLM-generated content, detection is *not* the primary focus of this work. It is important to acknowledge, however, that detecting LLM-generated content is critical and has been the subject of several noteworthy studies. These include the perplexity-based detection methods mentioned above, as well as supervised classification approaches (Verma et al., 2024; Pu et al., 2022; Jawahar et al., 2020; Ghosal et al., 2023; Tang et al., 2024; Dhaini et al., 2023; Guo et al., 2023), with fine-tuning of pretrained models being especially effective (Zellers et al., 2020; Solaiman et al., 2019;

Fagni et al., 2021). While detecting such content has inherent limitations (Varshney et al., 2020; Sadasivan et al., 2024)—as LLMs are trained to model the full joint distribution of human language—continued progress in this area is essential for mitigating societal risks. These include, but are not limited to, the spread of misinformation (Zellers et al., 2020), fake online reviews (Yao et al., 2017), potentially harmful medical advice (Guo et al., 2023), academic dishonesty (Susnjak, 2022), and extremist propaganda (McGuffie & Newhouse, 2020). The importance of these efforts is underscored by incidents where LLM-generated news articles containing factual inaccuracies were published with minimal human oversight (Christian, 2023). We hope our work will help along these directions.

In this work, we examine the impact of various factors, including the model size. Prior literature has consistently shown that larger models tend to perform better, with the benefits of scaling being predictable empirically (Hestness et al., 2017; Kaplan et al., 2020; Alabdulmohsin et al., 2022; Zhai et al., 2022). These improvements are not limited to perplexity scores. For example, (Dou et al., 2022) found that larger models produce texts with fewer factual and coherence-related issues. Not surprisingly, we also find that bigger pretrained models yield fractal parameters that are closer to those of natural language (see Figure 3).

Additionally, we investigate other factors, such as temperature settings. Previous research has demonstrated that while improved decoding methods may deceive humans, they may also introduce detectable statistical anomalies (Ippolito et al., 2020). Our work explores these effects in detail from the lens of fractals, contributing to the broader understanding of how model parameters influence LLM output.

## 3. Experimental Setup and Dataset

Our goal is to investigate when and how LLM-generated texts can vary substantially from natural language in their fractal structure. For that we need to generate synthetic texts. In order to correctly identify the impact of each factor we consider in our study (e.g. temperature setting, prompting, model size), we generate the data ourselves from scratch. We follow a similar setup to the one used in (Verma et al., 2024), in which we restrict analysis to long documents or paragraphs, as opposed to short answers to questions. Similar to (Verma et al., 2024), we query a capable LLM via its API, which is always Gemini 1.0 Pro (Anil et al., 2024) in our experiments, to generate some contextual information about an article (such as keywords or a summary) before asking another model to write an article based on this contextual information. Since each article is matched with a corresponding human-generated text (ground truth), we name this dataset "Generated And Grounded Language

*Table 1.* A summary of the contextual information used during prompting, ordered from the least informative (top) to the most (bottom). Each prompting method provides more contextual information than the one above it. Keywords and summaries were generated using Gemini 1.0 Pro as discussed in Section 3. See Appendix C for prompt templates.

| Abbreviation | Description |
|---|---|
| continue (cont) | Simple continuation based on a short prefix (no prompting). |
| chain-of-thought (cot) | Ask the model to generate an outline before generating the article, using a short prefix. |
| short keywords (kw) | A few, unordered keywords. |
| keywords (kw+) | Many unordered keywords. |
| summary (su) | A summary of the article. |
| summary + keywords (su+) | Both a summary and many keywords. |
| excerpt (exc) | Ordered list of long excerpts from the original article. |

Examples" (GAGLE). It contains over 240,000 articles[1].

The contextual information we use were chosen such that they can be *ordered* from the least informative to the most, as shown in Table 1. The datasets we use are from five domains: (1) WIKIPEDIA (Wikimedia, 2024), (2) BIG-PATENT, consisting of over one million records of U.S. patents (Sharma et al., 2019), (3) NEWSROOM, containing over one million news articles (Grusky et al., 2018), (4) SCIENTIFIC, a collection of research papers obtained from ArXiv and PubMed repositories (Cohan et al., 2018), and (5) BILLSUM, containing US Congressional and California state bills (Kornilova & Eidelman, 2019). We only use, at most, 1,000 articles from each domain.

In our experiments, we use pretrained models (with simple continuation only) and instruction-tuned models with various prompting strategies as discussed earlier. During text generation, we experiment with three decoding temperatures: $\beta = 0$ (greedy decoding), $\beta = 0.5$, and $\beta = 1$ (pretraining temperature). The three models we use are Gemini 1.0 Pro (Anil et al., 2024), Mistral-7B (Jiang et al., 2023), and Gemma-2B (Mesnard et al., 2024).

Once the texts are generated, we score them using pretrained models. One goal is to identify if our findings remain robust across different scoring models. As mentioned earlier, some previous works suggest that smaller models might be better for scoring and detecting LLM-generated texts (Mireshghallah et al., 2024) while other works suggest that using the same model for both generation and scoring yields better results (Fagni et al., 2021; Mitchell et al., 2023).

---

[1]The GAGLE dataset is available at: https://huggingface.co/datasets/ibomohsin/gagle

Finally, once all the log-perplexity scores are calculated, we compute fractal parameters. Because S and H are exponents of power laws, sufficiently long documents are required. Hence, we encourage the model to generate long documents in the prompt (see prompting templates in Appendix C). We drop the first 64 tokens to remove any warm-up effects, and ignore documents that are less than 400 tokens in length. When a document is ignored, we also ignore the corresponding ground-truth document, to remove this confounding effect. Then, we clip all documents (both human- and LLM-generated) to 400 tokens to have equal lengths. We estimate fractal parameters using the scales (time gaps) $\tau \in \{8, 16, 32, 48, 64, 96, 128, 160, 192, 256, 320\}$ with $\epsilon = 10^{-2}$; see (Alabdulmohsin et al., 2024) for details on how to calculate them. We also use bootstrapping (Efron & Tibshirani, 1994) to estimate confidence intervals by subsampling with replacement 10 independent samples. Figure 2 illustrates quality of fit for the setting in Appendix G where samples of documents are provided.

**Disclaimer.** Our estimates of S and H differ from those in (Alabdulmohsin et al., 2024) for two reasons. Because LLM-generated texts are short (typically of about 500 tokens), we restrict the range of the scale term $\tau$ to at most 320 tokens. Also, we use a slightly larger value of $\epsilon$ because we found that smaller values in short documents lead to high variance. Both imply that our estimates are less accurate. Nonetheless, our goal is to use S and H as statistical *probes* to compare natural texts from LLM-generated, so we use the same hyperparameters for both types of documents.

## 4. Detailed Analysis

**Q1. How do log-perplexity scores in LLM-generated documents differ from those in language?** To answer this question, results are shown in Figure 4. In agreement with prior works, we observe that LLM-generated texts have a lower log-perplexity scores (negative values in the $y$ axis) but not for large pretrained models when prompted using their pretraining temperature $\beta = 1$. In the latter setting, LLM-generated texts have a similar average log-perplexity score to natural language. We illustrate this using the GLTR tool (Gehrmann et al., 2019) in Figure 5. Gemma-2B is an exception, probably because it is much smaller than the rest of the models. Instruction-tuning, by contrast, lowers the log-perplexity of generated texts compared to natural language even at temperature $\beta = 1$, although no prompting instructions are used in Figure 4. Similar findings are also obtained using RAID dataset, as shown in Appendix A. Overall, this suggests that detection methods relying solely on log-perplexity scores, such as GLTR, may not be adequate for identifying contents generated by pretrained models as they become increasingly more capable in the future.

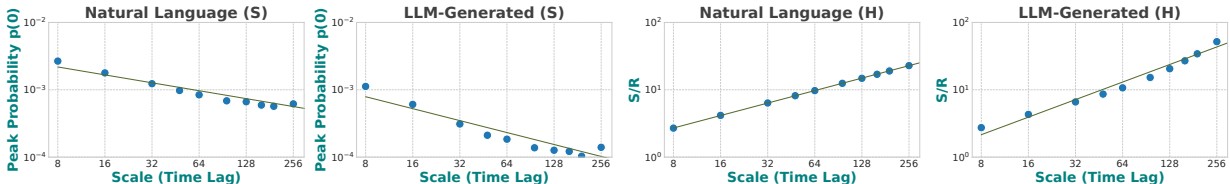

*Figure 2.* Quality of power law fit for fractal parameters in both human- and LLM-generated documents, for the same setting as in Appendix G where samples of documents are provided.

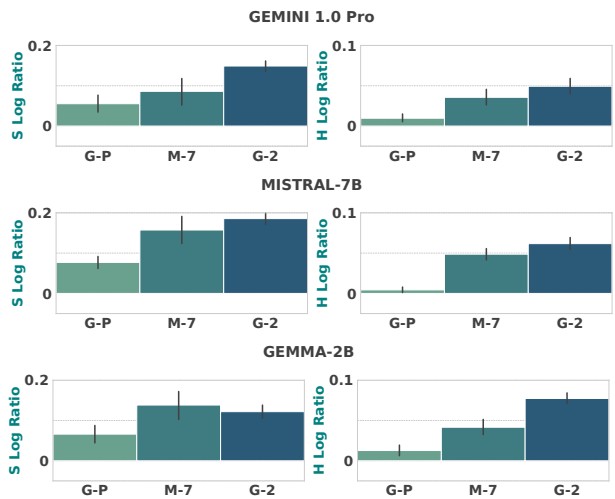

*Figure 3.* The $y$-axis is either $\log \tilde{S}/S$ (left column) or $\log \tilde{H}/H$ (right column), where $\tilde{S}$ is the Hölder exponent of LLM-generated texts while $S$ is of natural language, and the same holds for $\tilde{H}$ and H. The $x$-axis are the generating models: Gemini 1.0 Pro (denoted G-P), Mistral-7B (denoted M-7), and Gemma-2B (denoted G-2), all are pretrained models with temperature $\beta = 1$. Subtitles indicate the model used for scoring the texts. As expected, we observe that larger models tend to replicate the fractal properties of natural language better than smaller models. In addition, LLM-generated texts has higher values of both S (less self-similarity) and H (more dependence).

**Q2. If large pretrained LLMs at their pretraining temperature $\beta = 1$ can replicate the 1st-order statistics of log-perplexity scores in language, do they also replicate its fractal parameters?** Figure 3 summarizes the results. We observe that larger pretrained models at temperature $\beta = 1$ replicate the fractal properties of natural language better, regardless of which model is used for scoring. In addition, LLM-generated texts are systematically biased towards higher values of both S (less self-similarity) and H (more dependence) than in natural language. As we will discuss later in Q6, this means they are biased towards generating texts with *lower* quality than in natural language.

**Q3. What about fractal parameters in instruction-tuned models?** We examine the impact of instruction tun-

ing on fractal parameters when all texts are generated in both pretrained and instruction-tuned models using simple continuation (no prompting). We focus on simple continuation here to isolate the impact of instruction-tuning alone. To recall, Figure 4 shows that instruction tuning generates texts with lower log-perplexity scores than natural language. Figure 6 shows that texts generated by instruction-tuned models have higher values of the Hurst exponent at low temperatures $\beta < 1$, indicating more dependence over time. Self-similarity is not impacted, however. Similar findings are also obtained using RAID dataset, as shown in Appendix A.

**Q4. What if contextual information is provided in the prompt to instruction-tuned models?** As mentioned in Section 1, we also consider the causal graph shown in Figure 1, and examine the impact of adding various contextual cues in the prompt (see Table 1). Figure 7 summarizes the results across all combinations of generating models, scoring models, and datasets. For self-similarity, we observe a double descent. While the second descent is expected, given that LLMs should be eventually capable of replicating the original article if its entire content is provided in the prompt, the fact that providing a sample of unordered keywords is better than a summary is surprising! We also observe that asking the model to generate an outline first, similar to chain-of-thought (CoT) prompting (Wei et al., 2023), yields fractal parameters that are closer to those of natural language than simple continuation. In addition, as shown in Appendix F, generating patent and science articles seem to be more sensitive to prompting than in other domains.

**Q5. How sensitive are fractal parameters of LLM-generated articles to the contextual information provided in the prompt?** To answer this question, we first calculate for each of the three architectures Gemini 1.0 Pro, Mistral-7B, and Gemma-2B the average Hölder and Hurst exponents disaggregated by prompting method, where averages are calculated over all remaining variables (e.g. decoding temperature, scoring algorithm, and dataset). Then, we plot the standard deviation calculated across the prompting methods. The results are displayed in Figure 8. As

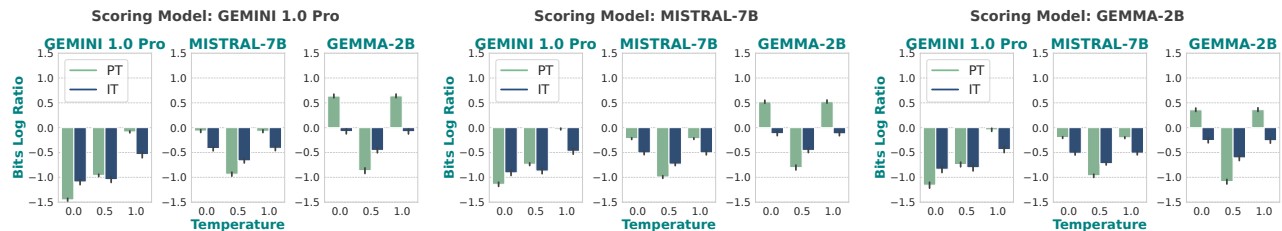

*Figure 4.* *y*-axis is the log-ratio of log-PPL scores for both pretrained (PT) and instruction-tuned (IT) models with simple continuation, when Gemini 1.0 Pro (left), Mistral-7B (center), and Gemma-2B (right) is used to score texts. Texts generated by large pretrained models do *not* have a lower log-PPL than natural texts; instruction tuning and the use of small temperatures lead to that effect.

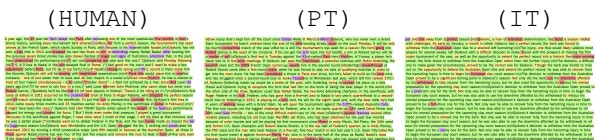

*Figure 5.* Output of the GLTR tool (Gehrmann et al., 2019) on texts generated by humans (left), Mistral-7B pretrained (middle) and Mistral-7B instruction-tuned (right) at temperature $\beta = 1.0$. Both the pretrained and instruction-tuned models are provided with a short prefix. Colors indicate perplexity scores. The output of the pretrained model looks similar to the human-generated text in terms of log-PPL scores (more orange, red, and purple tokens indicating surprise), in agreement with Figure 4.

expected, larger models are less sensitive to the choice of the prompting method than smaller models.

**Q6. How do fractal parameters relate to the quality of output?** To answer this question, we first recall that the Hölder exponent S quantifies the level of self-similarity in a stochastic process, with *smaller* values indicating a *more* self-similar structure (heavier tail); i.e. with complex, rich details at all levels of granularity. Hence, lower values of S are desirable. By contrast, the Hurst exponent H quantifies dependence over time. Values close to H ≈ 0.5 indicate no dependence (i.e. the process is random) while values close to H ≈ 1.0 indicate strong predictability (e.g. when the same text is repeated over and over again). Natural language has values close to H ≈ 0.65. In practice, LLMs do not generate words entirely at random, so the correlation between H and model quality is negative. For this reason, H is also strongly and positively correlated with S, with a Pearson coefficient of 0.68 and a p-value $< 10^{-15}$.

In Figure 9, we plot the average quality of documents against their average log-perplexity score and fractal parameters. Here, we use Gemini Pro 1.0 to auto-rate the quality of generated texts. The prompt template and examples of responses are in Appendix C and we provide examples of quality ratings in Appendix G. We observe, as expected, that both S and H are negatively correlated with quality. Interestingly, H is a much stronger predictor of average quality

than the other metrics. Similar findings are also obtained using RAID dataset, as shown in Appendix A. The reason log-perplexity is not a good predictor of quality can be illustrated with a simple example. Suppose that the entire document comprises of a single word repeated over and over again. Then, the log-perplexity of each subsequent token gets progressively closer to zero, since the next token can be reliably predicted. Obviously, this does not imply that an article made of a single repeating word has a good quality. The Hurst exponent, on the other hand, will be quite large in the latter case, indicating poor quality. In Appendix G, we provide a sample of documents from hyperparameter settings that yield large and small values of H for comparison.

**Q7. How well does the range of fractal parameters overlap between natural language and LLM-generated texts?** Figure 10 shows that the range of fractal parameters in natural language is mostly a *narrow* subset of those observed in LLM-generated texts. Clearly, while there is an overlap, natural language maintains S and H in a narrow range whereas they vary widely in LLM-generated texts. Similar findings are also obtained using RAID dataset, as shown in Appendix A. This is consistent with the earlier observation about the relation between fractal parameters and quality of texts. Among the different factors considered, we find that *prompting* has the biggest impact on S and H as shown in Table 2, which calculates the Shannon mutual information between fractal parameters and other variables.

Nevertheless, it is important to keep in mind that in Figure 10, each value of the fractal parameter is calculated over a *corpus* of texts, not individual documents. A corpus of texts corresponds to a particular combination of generating model, decoding temperature, prompting method, scoring model, and dataset. The reason fractal parameters are calculated over multiple documents is because they describe properties about the *underlying stochastic process*, such as its autocorrelation function $\rho_n = \mathbb{E}[x_{t+n}x_t]$, not properties about a single individual realization of it. Hence, multiple independent measurements are needed.

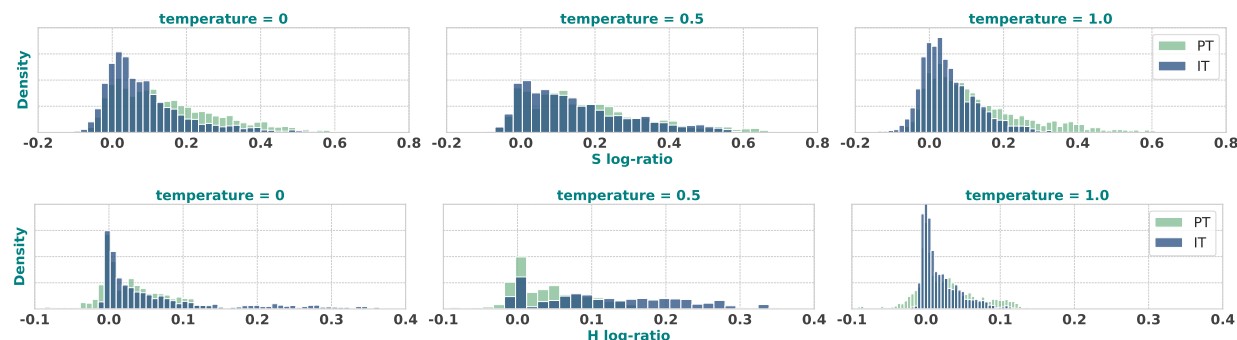

*Figure 6.* Distribution of the log-ratio of Hölder exponent S (top) and Hurst exponent H (bottom) in pretrained (PT) and instruction-tuned (IT) models (all with simple continuation) compared to natural language. Instruction-tuned models at low temperatures $\beta < 1$ have higher values of H; i.e. more dependence over time. See Section 4/Q3 for further discussion.

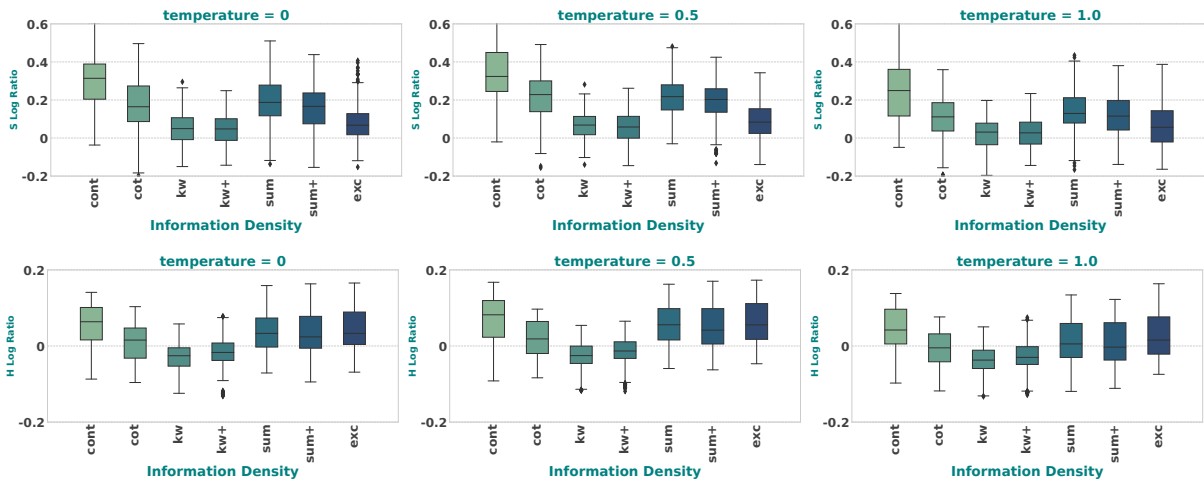

*Figure 7.* $y$-axis is the log of the ratio of the fractal parameters between LLM-generated texts and natural language, similar to Figure 3. Here, we only use instruction-tuned models. $x$-axis is from left to right the 7 prompting strategies in Table 1 (top to bottom). Detailed results are in Appendix F.

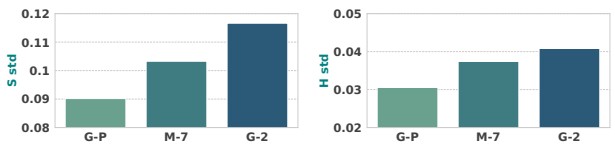

*Figure 8.* Standard deviation of S (left) and H (right) calculated across prompting methods for each instruction-tuned model: Gemini 1.0 Pro (G-P), Mistral-7B (M-7) and Gemma-2B (G-2).

**Q8. Are there notable differences when LLMs are used to score their own outputs?** Inspired by prior observations, which suggest that using *similar* architectures for both scoring and generation might work better for detecting LLM-generated texts (Mitchell et al., 2023; Fagni et al., 2021), we explore if there are differences in fractal parameters when

*Table 2.* Shannon mutual information (Shannon, 1948) between S or H and other variables, normalized by the Shannon entropy of the fractal parameter. We bin values into intervals of length 0.1. Prompting has the biggest impact on fractal parameters, followed by the data domain.

|  | Scoring Model | Generating Model | Temp | Dataset | Prompt |
|---|---|---|---|---|---|
| S | 0.2% | 1.0% | 1.0% | 7.3% | 8.4% |
| H | 1.7% | 4.8% | 4.3% | 7.2% | 19.7% |

LLMs score their own outputs (i.e. using Mistral-7B to score its output, as opposed to the output of other models).

One way to examine this is to look into the reduction in uncertainty:

$$\mathcal{J}(X;Y) \doteq U(X \mid Z) - U(X \mid Z, Y), \qquad (3)$$

where $U(X|Z)$ is a measure of the uncertainty in $X$ when

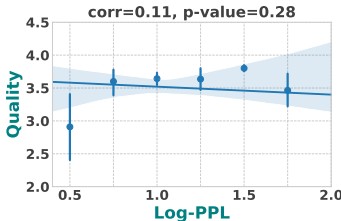 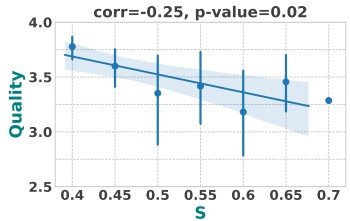 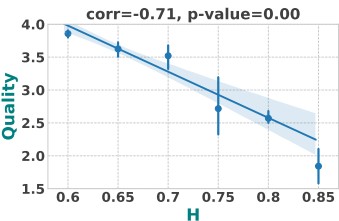

*Figure 9.* Average quality of LLM-generated documents, as judged by Gemini 1.0 Pro, vs. log-PPL (left), Hölder exponent (middle), and Hurst exponent (right). The Hurst parameter is a much better predictor of quality than the other metrics. See Section 4/Q6 for discussion, Appendix C for the exact prompt used in Gemini 1.0 Pro, and Appendix G for examples.

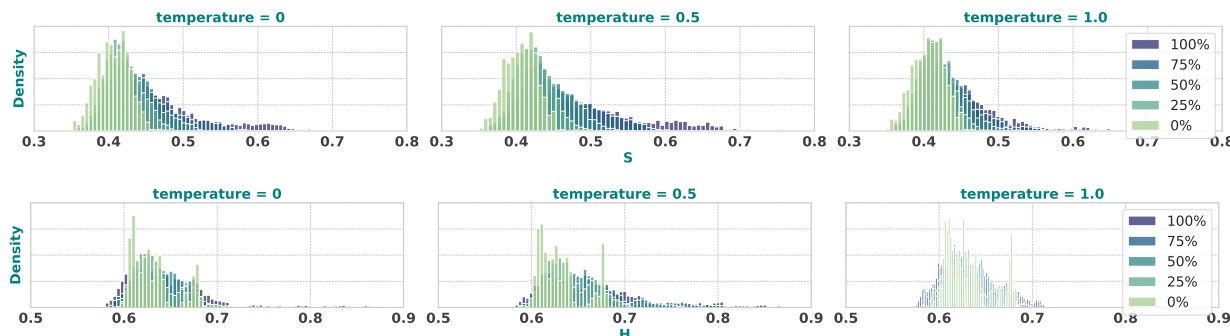

*Figure 10.* Distribution of S and H for collections containing both LLM-generated and human-generated texts, where the values in legend indicates the proportion of LLM-generated texts. See Section 4/Q7 for details. Cases when LLMs repeat the same text mostly occur with greedy decoding (left) but we still see different distributions of S and H in higher temperatures.

conditioning on $Z$, such as the conditional Shannon entropy or the error rate of predicting $X$ given $Z$. In our setting, $X$ and $Y$ are the scoring and generating models while $Z$ are the remaining variables, such as dataset, decoding temperature, and fractal parameters. Intuitively, because fractal parameters are included in the set of predictors, if the error rate of predicting the generating model does not depend on the scoring model even when fractal parameters and other variables are included in the set of predictors, then fractal parameters remain relatively unchanged whether or not the same model is used for generating and scoring texts, which is indeed what we observe. Specifically, a random forest classifier, in its default Sciki-Learn implementation (Pedregosa et al., 2011), can predict the scoring model with a high accuracy of 97.0% without having to include the generating model in the set of predictors and this accuracy remains unchanged when including the generating model. Similarly, the accuracy of predicting the generating model is quite high at 97.8% without using the scoring model as a predictor. Including the scoring model improves accuracy only slightly.

**Q9. Are some domains easier to synthesize?** Table 3 shows that LLM-generated encyclopedic articles and legal documents are closer to those of natural language in terms of average log-perplexity scores (1st order statistics) and

fractal parameters (2nd order). However, this may be a reflection of the weight of those domains during (pre)training, rather than anything fundamental about the domains themselves. Interestingly, when using the RAID dataset, it seems challenging for LLMs to replicate humans in *poetry*, and this only becomes evident when we look into the Hölder exponent.

## 5. Discussion and Limitations

In this work, we investigate whether LLMs are capable of replicating the fractal structure of language. We note that various strategies, such as the decoding temperature and prompting method, can impact fractal parameters even when log-perplexity scores seem to be unaffected. This goal is in line with earlier works, such as Meister & Cotterell (2021), who argued that the evaluation of LLMs should go beyond log-perplexity and also consider how well LLMs capture other "statistical tendencies" observed in language.

Our findings reveal that for pretrained models, larger architectures are more effective at capturing such fractal properties. In addition, with instruction-tuned models, the similarity to human language does not improve monotonically as the amount of contextual information in the prompt increases. Notably, the Hurst parameter emerged as a strong

*Table 3.* Log-ratio of fractal parameters and log-perplexity scores between LLM-generated texts and natural language using instruction-tuned models. Results are averaged across all settings (e.g. decoding temperatures, models and prompts). LLM-generated encyclopedic and legal documents are closer to natural language, than other domains.

| Domain | S | H | log-perplexity |
|---|---|---|---|
| **GAGLE Dataset** | | | |
| NEWSROOM | $0.19 \pm 0.14$ | $0.02 \pm 0.05$ | $-0.58 \pm 0.29$ |
| SCIENTIFIC | $0.17 \pm 0.15$ | $0.07 \pm 0.05$ | $-0.61 \pm 0.27$ |
| BIGPATENT | $0.15 \pm 0.15$ | $0.04 \pm 0.06$ | $-0.53 \pm 0.26$ |
| BILLSUM | $0.06 \pm 0.10$ | $-0.04 \pm 0.05$ | $-0.12 \pm 0.28$ |
| WIKIPEDIA | $0.06 \pm 0.12$ | $-0.01 \pm 0.04$ | $-0.46 \pm 0.29$ |
| **RAID Dataset** | | | |
| ABSTRACTS | $-0.13 \pm 0.05$ | $0.16 \pm 0.02$ | $-0.59 \pm 0.09$ |
| BOOKS | $-0.10 \pm 0.04$ | $0.07 \pm 0.01$ | $-0.66 \pm 0.04$ |
| NEWS | $0.18 \pm 0.02$ | $0.11 \pm 0.01$ | $-0.53 \pm 0.04$ |
| POETRY | $0.50 \pm 0.02$ | $0.05 \pm 0.01$ | $-0.67 \pm 0.10$ |
| RECIPES | $0.10 \pm 0.05$ | $0.05 \pm 0.01$ | $-0.75 \pm 0.04$ |
| REDDIT | $-0.07 \pm 0.02$ | $0.11 \pm 0.01$ | $-0.67 \pm 0.06$ |
| REVIEWS | $0.08 \pm 0.02$ | $0.13 \pm 0.02$ | $-1.23 \pm 0.11$ |

predictor of quality in generated texts, among other significant findings. To facilitate further research in this area, we release our GAGLE dataset, which comprises over 240,000 LLM-generated articles.

**Limitations.** In terms of limitations, estimating fractal parameters requires analyzing large corpora of lengthy documents because these parameters describe properties of the underlying stochastic processes. Therefore, they may not be reliable at making conclusions about *individual* documents or short texts. This limitation prevents us from making claims about the ability to detect AI-generated content using these metrics alone. However, we note that perplexity-based detection methods might be enhanced by incorporating second-order statistics such as the Hölder and Hurst exponents, since the range of those parameters in LLM-generated articles varies widely compared to human-generated texts. We leave the exploration of detection strategies that leverage these fractal characteristics for future work.

In addition, we focus on auto-regressive models. Exploring whether our findings hold for fundamentally different architectures, such as state space models (SSMs) (Gu & Dao, 2024), is a valuable direction for future research.

## Impact Statement

There are many potential societal consequences of advancing the field of artificial intelligence (AI), both positive, such as improved accessibility to high-quality healthcare and education, and negative, if such systems are misused. The goal in this work is to contribute to the ongoing effort to improve understanding of language models and the mechanisms behind their success. While we recognize the general ethical considerations that accompany the advancement of language models and AI, we do not feel that this particular work raises unique or unaddressed ethical concerns beyond the established considerations within the field.

## Acknowledgement

We thank Vinh Tran and Jeremiah Harmsen for their insightful reviews and suggestions, Mostafa Dehghani and Mike Mozer for early discussions, and Google DeepMind at large for providing a supportive research environment.

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

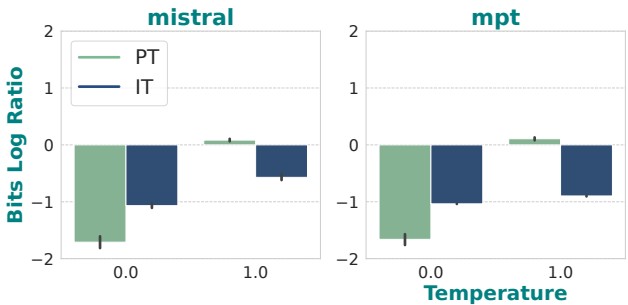

*Figure 11.* Similar to Figure 4, *y*-axis is the log-ratio of log-PPL scores for both pretrained (PT) and instruction-tuned (IT) models, when Gemini 1.0 Pro is used to score texts.

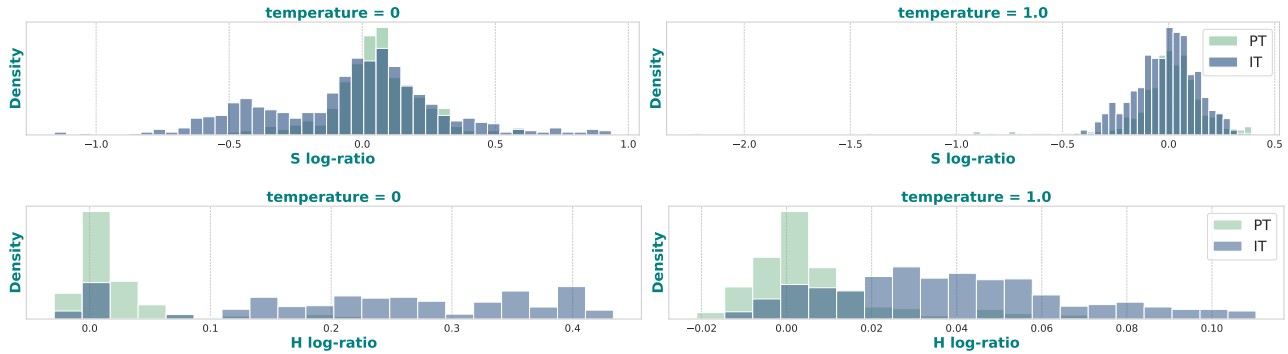

*Figure 12.* Similar to Figure 6, this figure shows the distribution of the log-ratio of the Hölder exponent S (top) and Hurst exponent H (bottom) across all LLM-generated articles in RAID dataset, when scored by Gemini 1.0 Pro.

## A. Additional Experiments using RAID Dataset

In this section, we conduct additional experiments using RAID dataset (Dugan et al., 2024), which contains articles generated by 11 models (e.g. ChatGPT, Cohere, Llama 2, etc) in domains such as Reddit and reviews, among others. The goal is to verify if our main conclusions continue to hold. Since we do not control the prompts in RAID and we only score texts using Gemini Pro 1.0, Q2/4/5/8 are omitted in this analysis.

Overall, we find a substantial agreement. We summarize our findings below:

1. **Q1 (Log-perplexity)**: Consistent with our results, greedy decoding and instruction tuning yield lower perplexity than human text, but pretrained models at $\beta = 1$ show perplexity *similar* to human text. See Figure 11.

2. **Q3 (Fractals in IT Models)**: Our findings still hold as shown in Figure 12: Instruction tuning affects the Hurst exponent (H), especially at low temperatures (leading to higher H), while Self-Similarity (S) remains largely unaffected.

3. **Q6 (Text Quality)**: We use Gemini 1.0 Pro to evaluate the quality of generated articles and compare the average quality against the process of generating articles (e.g. model and hyperparameters). As before, only the Hurst exponent (H) is well-correlated with quality. This observation now holds across the 11 models and 7 domains in RAID, reinforcing our earlier result. Results are shown in Figure 13.

4. **Q7 (Distribution of Fractals)**: Natural language still has a tighter distribution of fractal parameters compared to LLM-generated text, particularly for S at low decoding temperatures. Results are shown in Figure 14.

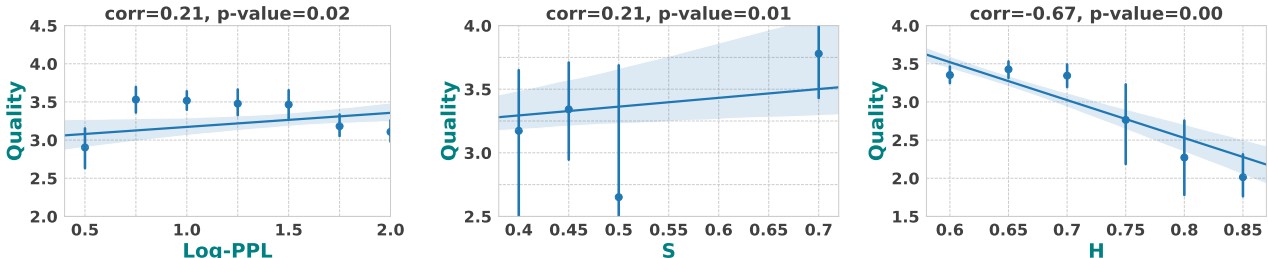

*Figure 13.* Similar to Figure 9, $y$ axis is the average quality of LLM-generated documents, as judged by Gemini 1.0 Pro, vs. log-PPL (left), Hölder exponent (middle), and Hurst exponent (right).

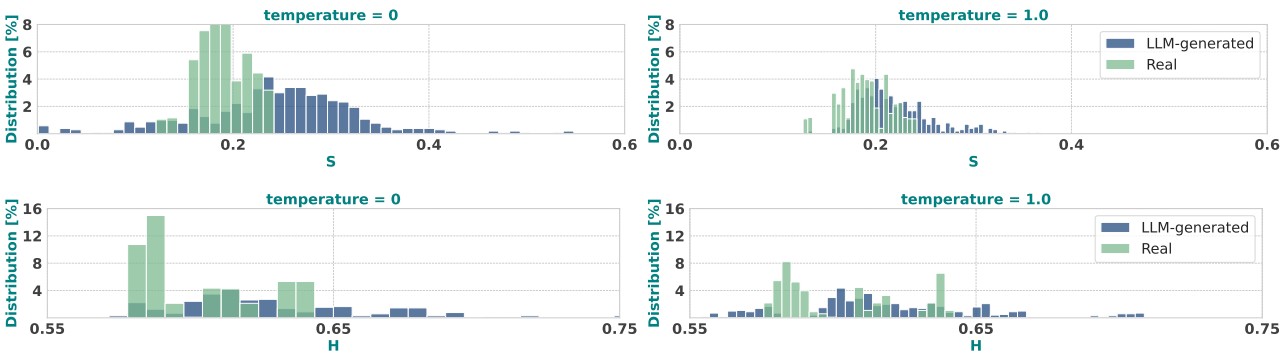

*Figure 14.* Similar to Figure 10, the distribution of S and H is shown for collections containing either LLM-generated texts or original articles.

## B. Definitions of Fractal Parameters

In this appendix, we provide a concise, self-contained description of the two fractal parameters: (1) the Hölder (Self-Similarity) exponent S and (2) the Hurst exponent H. We use the code provided by Alabdulmohsin et al. (2024) to calculate these parameters.

### B.1. Hölder Exponent

An object is "self-similar" if its statistical or geometric properties remain consistent across different scales. Examples include coastlines, snowflakes, the Cantor set, and the Koch curve. Analogously, a stochastic process is self-similar if it is *distributionally* similar to a rescaling of time.

Formally, let $(x_t)_{t \in \mathbb{N}}$ be a stochastic process, such as a sequence of log-perplexity scores, and write $(X_t)_{t \in \mathbb{N}}$ for its integral process: $X_t = \sum_{i=1}^{t} x_i$. Then, the process is said to be self-similar if $(X_{\tau t})_{t \in \mathbb{N}}$ is distributionally equivalent to $(\tau^S X_t)_{t \in \mathbb{N}}$ for some exponent S. Here, S is the Hölder exponent.

One way to calculate S is as follows. Fix $\epsilon \ll 1$ and denote the $\tau$-increments by $(X_{t+\tau} - X_t)_{t \in \mathbb{N}}$. These would correspond, for instance, to the number of bits used for clauses, sentences, paragraphs and longer texts as $\tau$ increases. In terms of the increment process $(x_t)_{t \in \mathbb{N}}$, this corresponds to aggregating increments into "bursts". Let $p_\epsilon(\tau)$ be the probability mass of the event $\{|X_{t+\tau} - X_t| \le \epsilon\}_{t \in \mathbb{N}}$. Then, S can be estimated by fitting a power law relation $p_\epsilon(\tau) \sim \tau^{-S}$ (Watkins, 2019).

### B.2. Hurst Exponent

The Hurst parameter $H \in [0, 1]$, on the other hand, quantifies the degree of predictability or dependence over time (Hurst, 1951). It is calculated using the so-called rescaled-range (R/S) analysis. Let $(x_t)_{t \in \mathbb{N}}$ be an increment process. For each $n \in \mathbb{N}$, write $y_t = x_t - \frac{1}{t} \sum_{k=0}^{t} x_k$ and $Y_t = \sum_{k=0}^{t} y_t$. The range and scale are defined, respectively, as $R(n) =$

$\max_{t \le n} Y_t - \min_{t \le n} Y_t$ and $S(n) = \sigma\left(\{x_k\}_{k \le n}\right)$, where $\sigma$ is the standard deviation. Then, the Hurst parameter H is estimated by fitting a power law relation $R(n)/S(n) \sim n^{\mathrm{H}}$.

# C. Prompting Templates

## C.1. Providing Contextual Information

*Table 4.* A summary of the prompting templates used in this work.

| Abbreviation | Description | Prompting Template |
|---|---|---|
| `continue` | Simple continuation based on a short prefix (no prompting). | None |
| `cot` | Ask the model to generate an outline first before generating the article. A short prefix is provided. | `Extend the following text.  First write an outline.  Then insert **Extended Text:** After that write the text using a minimum of 15 paragraphs containing a minimum of 1000 words, in plain text only without titles or headings.  The text you need to extend is:  <prefix>` |
| `short keywords` | A few, unordered list of keywords. | `Using these keywords:  <keywords>, write an article, in a minimum of 15 paragraphs containing a minimum of 1000 words, in plain text only without titles or headings.` |
| `keywords` | Many unordered keywords. | `Using these keywords:  <keywords>, write an article, in a minimum of 15 paragraphs containing a minimum of 1000 words, in plain text only without titles or headings.` |
| `summary` | A summary of the entire article. | `Write about the following in a minimum of 15 paragraphs containing a minimum of 1000 words, in plain text only without titles or headings:  <summary>.` |
| `summary+keywords` | Both a summary and an unordered list of many keywords. | `Write about the following in a minimum of 15 paragraphs containing a minimum of 1000 words, in plain text only without titles or headings: <summary>.  Using these keywords: <keywords>.` |
| `excerpt` | An ordered list of long excerpts from the original article. | `Write about the following in a minimum of 15 paragraphs containing a minimum of 1000 words, in plain text only without titles or headings:  <ordered excerpts>.` |

## C.2. Generating Contextual Information

To generate the keywords and summary, we use the instruction-tuned Gemini 1.0 Pro model with the following prompts:

- **Keywords**:

```
Generate keywords for the following text, but only respond with the keywords in
    plain text separated by commas. The text is:
<article>
```

- **Summary**:

```
Summarize the following text in few sentences, but only respond with the text
    summary in plain text. The text is:
<article>
```

### C.3. Quality of Texts

As discussed in Section 4/Q6, we use the instruction-tuned Gemini 1.0 Pro model to estimate the quality of texts and relate those to fractal parameters in Figure 9. The prompt we use to estimate the quality of an article is provided below:

```
First, explain briefly what is good and what is bad about the quality of the
    following document. Then, rate it on a scale from '1' to '5', where '1' is
    poorest and '5' is best. The last character in your response must be a single
    digit that corresponds to your rating. The document is:\n <article>
```

Examples of how the model evaluates the quality of articles is provided in Appendix G.

## D. Prompts as Do-Queries Example

To illustrate the differences between Equations 1 and 2, suppose hypothetically that all of the world's conversations are about two topics only: fairy tales or the weather. These two will be our possible contexts. Let us also assume that 75% of the time, people talk about the weather.

In addition, let us suppose that only three possible prefixes listed below are used with equal probability as shown in the table:

| Prefix | $p(\text{weather}|\text{prefix})$ | $p(\text{fairytale}|\text{prefix})$ |
|---|---|---|
| "It is a sunny day" | 50% | 50% |
| "It is a lovely sunny day | 85% | 15% |
| "It is raining" | 90% | 10% |

The causal graph in this setup would be as follows: (1) we first select a context, which is "weather" with probability 75% and "fairy tales" with probability 25%. (2) Then, we select one of the three prefixes. By Bayes rule, for example, the probability we select the first prompt above given that the context is "weather" would be $22.22\%$. Finally, (3) we continue the text, given both the context and the prefix.

If a language model is trained on this data and we prompt it with the phrase "*It is a sunny day*," what should the model predict next? There are two ways to interpret this question. One option is that the model should give a prediction based on *actual* historical data, using the conditional distribution $p\,(\text{suffix} \mid \text{prefix})$. In this case, there is 50% probability that the context is about the weather, so the model will continue the prefix by describing the weather 50% of time and continue the prefix by describing a fairy tale in 50% of the time.

Another interpretation, however, is that by prompting the model, we are asking the model to act as if there was an *intervention* that forced all conversations to start with the prefix "*It is a sunny day*" and predict how the historical distribution would have looked like. In that case, Equation 1 implies that the model should sample a context first from its marginal distribution, which would be "weather" with probability 75% and fairy tales with probability 25%. Once the context is selected, it can use the historical data to predict the continuation of the prefix "*It is a sunny day*" after conditioning on that particular context. The intuition here from the causal graph in Figure 1 is that it is the context that dictates or causes the conversations, not the other way around, so the distribution of underlying contexts would remain the same.

# E. Data Card

*Table 5.* Generated And Grounded Language Examples (GAGLE) data card.

| | |
|---|---|
| **Description** | GAGLE comprises of LLM-generated articles. All articles are generated using the public checkpoints of the open-source models Mistral-7B (Jiang et al., 2023) and Gemma-2B (Mesnard et al., 2024). The seed for all articles are sourced from five academic datasets: (1) WIKIPEDIA articles (Wikimedia, 2024), (2) BIGPATENT, consisting of over one million records of U.S. patent documents (Sharma et al., 2019), (3) NEWSROOM, containing over one million news articles (Grusky et al., 2018), (4) SCIENTIFIC, a collection of research papers obtained from ArXiv and PubMed OpenAccess repositories (Cohan et al., 2018), and (5) BILLSUM, containing US Congressional and California state bills (Kornilova & Eidelman, 2019). So, all articles are of encyclopedic, news, legal, scientific or patents nature. The dataset contains, at most, 1,000 articles from each domain. The articles are generated via the following prompting strategies: 

 1. `continue (pt)`: Simple continuation based on a short prefix (no prompting) using pretrained model. 

 2. `continue (it)`: Simple continuation based on a short prefix (no prompting) using instruction-tuned model. 

 3. `cot`: Ask the model to generate an summary first before generating the article. A short prefix is provided. 

 4. `short keywords`: A few, unordered list of keywords. 

 5. `keywords`: Many unordered keywords. 

 6. `summary`: A summary of the entire article. 

 7. `summary + keywords`: Both a summary and an unordered list of many keywords. 

 8. `excerpt`: An ordered list of long excerpts from the original article. 

 With the exception of `continue (pt)`, all other prompts are used in instruction-tuned models. |
| **Primary Data Modality** | Text. |
| **Data Fields** | 1. **ID**: a unique ID identifying the original ground-truth article. 

 2. **Model**: one of Mistral-7B and Gemma-2B. 

 3. **Domain**: one of WIKIPEDIA, BIGPATENT, NEWSROOM, SCIENTIFIC, and BILLSUM. 

 4. **Prompt**: one of the specified prompting methods. 

 5. **Temperature**: numeric. Either 0.0 (greedy decoding), 0.5, or 1.0 (pretraining temperature). 

 6. **Prefix**: Prefix used to generate the article. 

 7. **Quality**: description of the quality of the article along with a rating from 1 (poorest) to 5 (best). 

 8. **Text**: Actual LLM-generated text. 

 9. **Log-Perplexity Scores**: The scores generated by one of Mistral-7B or Gemma-2B pretrained models. |
| **Intended Use Case** | Facilitate research in domains related to detecting and analyzing LLM-generated tests. |
| **Access Type** | Unrestricted. |
| **CC-by-4.0** | |
| **Sensitive Human Attributes** | N/A |

# F. Contextual Information in the Prompt: Full Figures

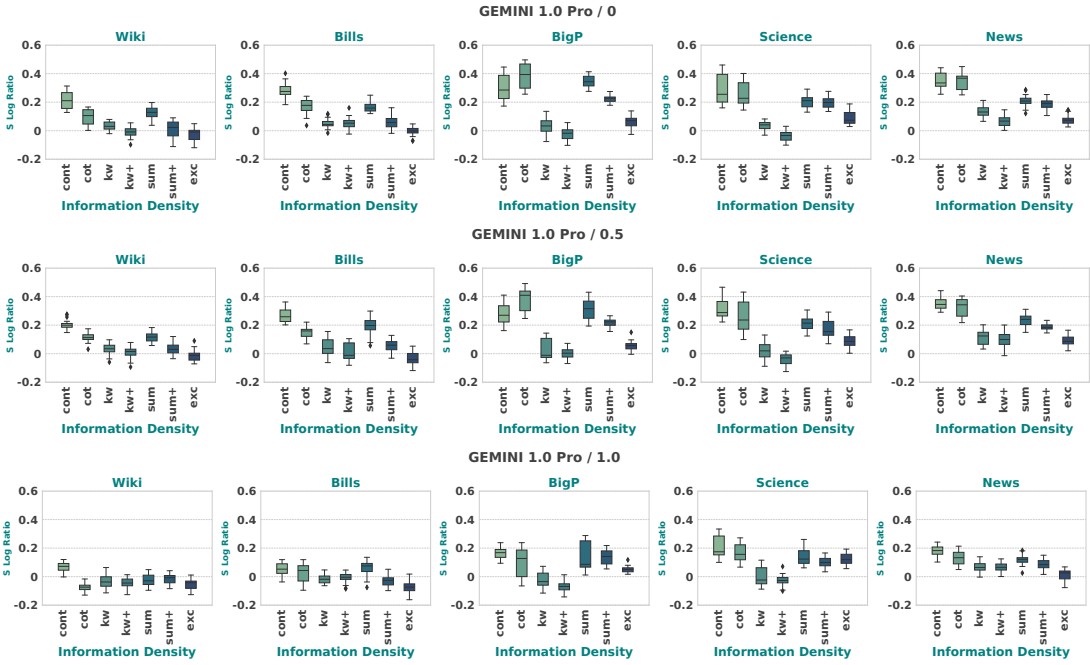

*Figure 15.* Hölder Exponent. Generating model = Gemini 1.0 Pro

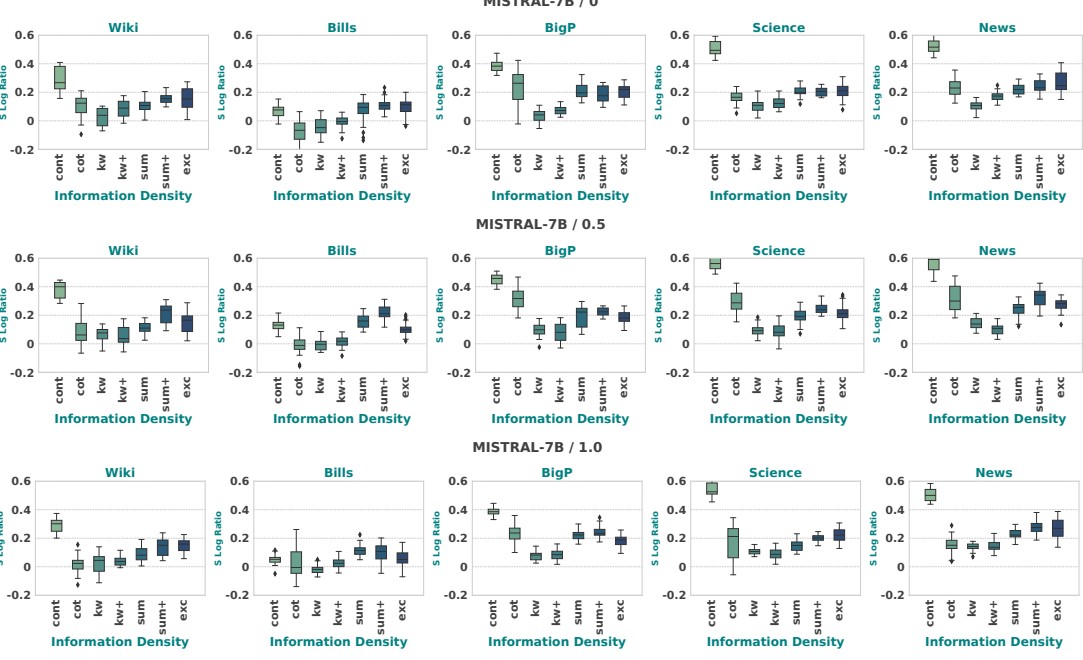

*Figure 16.* Hölder Exponent. Generating model = Mistral-7B

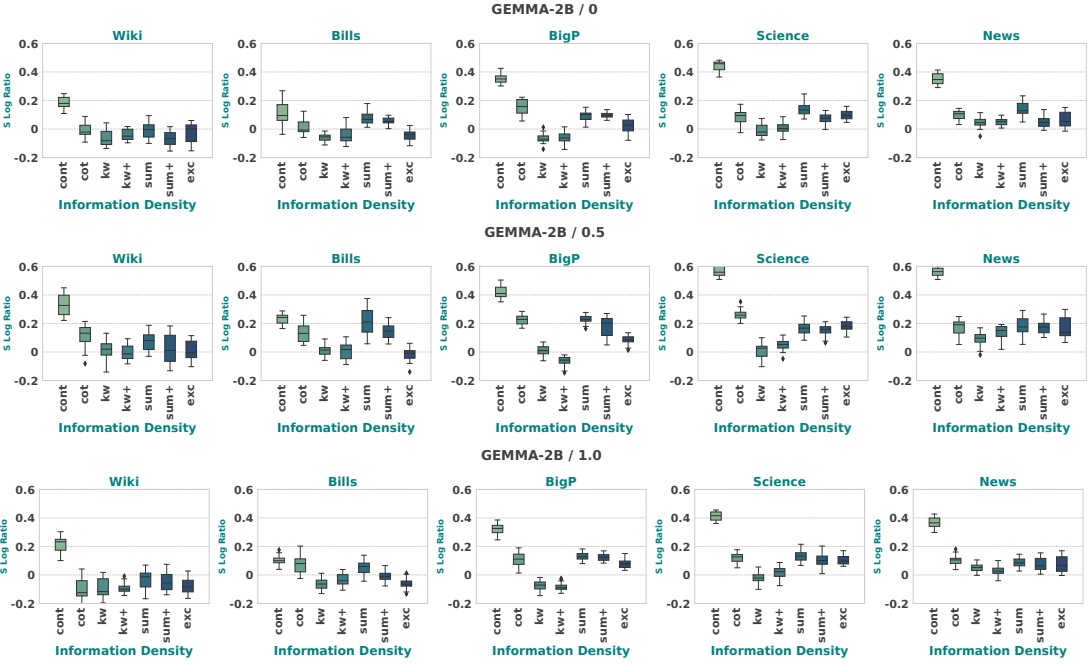

*Figure 17.* Hölder Exponent. Generating model = Gemma-2B

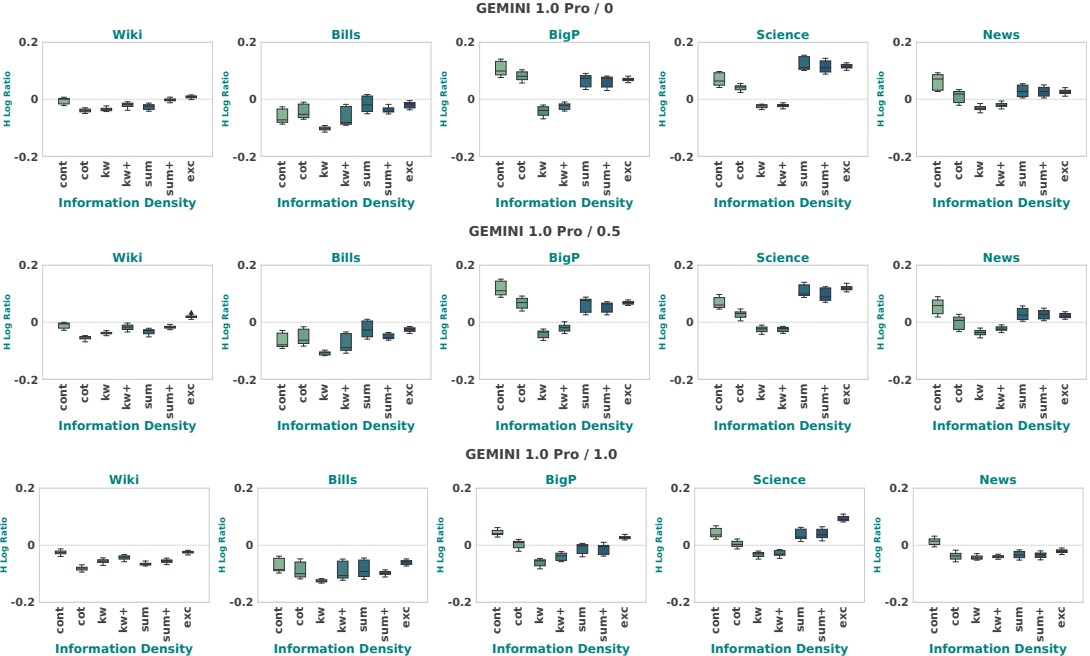

*Figure 18.* Hurst Exponent. Generating model = Gemini 1.0 Pro

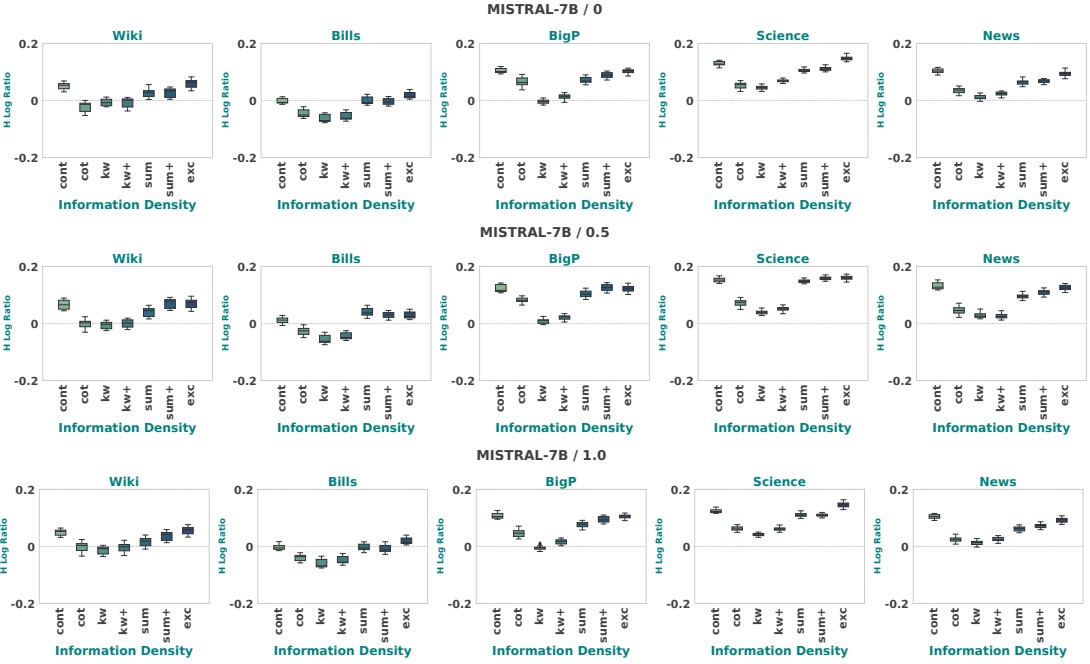

*Figure 19.* Hurst Exponent. Generating model = Mistral-7B

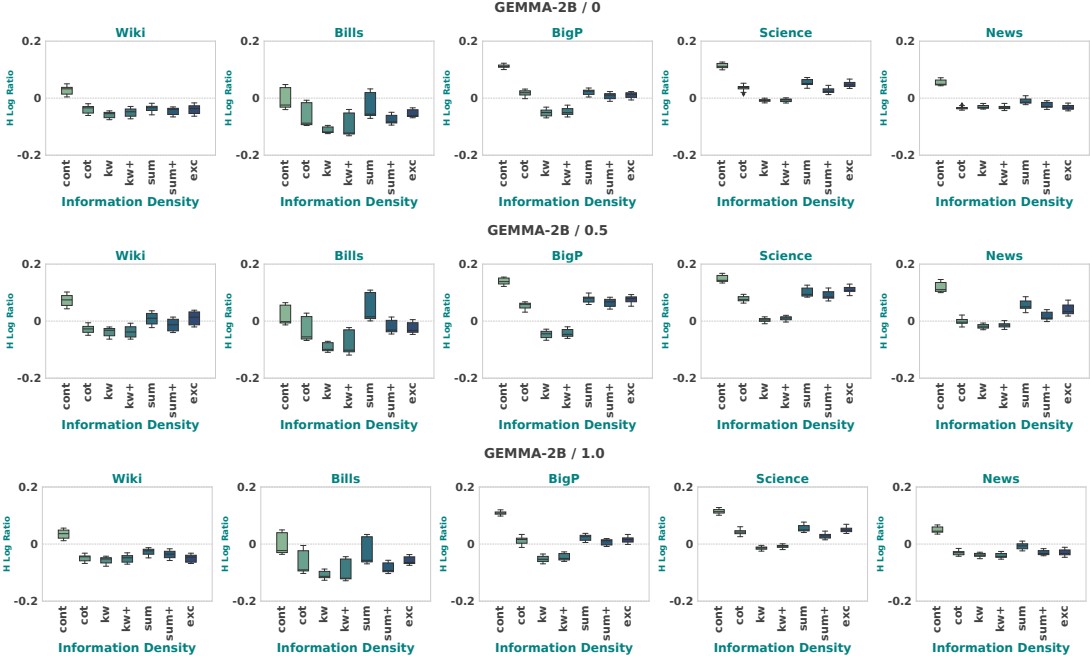

*Figure 20.* Hurst Exponent. Generating model = Gemma-2B

# G. Example Documents of Generating Processes with Large and Small Values of H

## G.1. Large Values of H

- **Data domain**: Newsroom.

- **Generating**:
    - *Model Name:* Mistral-7B (pretrained)
    - *Decoding Temperature*: 0.5
    - *Prompting Method*: `continue`

- **Fractal Parameters**:
    - *Scoring Model Name:* Gemini 1.0 Pro.
    - *Hölder Exponent*: S = 0.630
    - *Hurst Exponent*: H = 0.808

### G.1.1. DOCUMENT I

**Ground-truth Record ID:**

> `"newsroom-00001-00064-6"`

**Prefix**:

*(the following article contains a minimum of 15 paragraphs containing a minimum of 1000 words, in plain text only without titles or headings.)*

*A year ago, the 25-year-old Serb swept into Paris after amassing one of the most spectacular five months in men's tennis history, winning every tournament he'd entered. Now, far from a perfect season, the tournament's top seed arrives at the French Open,*

**Generated:**

> the last major of the year, having lost his last two matches, his ranking and his No. 1 ranking.
>
> In the past year, Djokovic has been beaten by a 17-year-old American, lost the last seven sets he's played, and been beaten by a player ranked 111th in the world.
>
> The Serb, who won the Australian Open and Wimbledon in the first two months of the year, was beaten by 17-year-old American Jared Donaldson in the final of the Stuttgart Open last month.
>
> He then lost to Dominic Thiem in the Rome Masters final, before being beaten by Gael Monfils in the quarterfinals of the Madrid Open.
>
> Djokovic, who has been ranked No. 1 for the last seven years, is now ranked 11th in the world.
>
> The Serb will be hoping to bounce back in Paris, where he has won three of the last four French Open titles.
>
> He will face 23-year-old Russian Andrey Rublev in the first round of the tournament.
>
> Rublev is ranked 22nd in the world, but has been on a roll in the last few months, winning the St. Petersburg Open and the ATP 500 event in Rome.
>
> The Serb has been in fine form in the last few months, winning the Australian Open and Wimbledon in the first two months of the year.
>
> Djokovic has been in fine form in the last few months, winning the Australian Open and Wimbledon in the first two months of the year.

The Serb has been in fine form in the last few months, winning the Australian Open
and Wimbledon in the first two months of the year.

The Serb has been in fine form in the last few months, winning the Australian Open
and Wimbledon in the first two months of the year.

The Serb has been in fine form in the last few months, winning the Australian Open
and Wimbledon in the first two months of the year.

The Serb has been in fine form in the last few months, winning the Australian Open
and Wimbledon in the first two months of the year.

The Serb has been in fine form in the last few months, winning the Australian Open
and Wimbledon in the first two months of the year.

The Serb has been in fine form in the last few months, winning the Australian Open
and Wimbledon in the first two months of the year.

The Serb has been in fine form in the last few months, winning the Australian Open
and Wimbledon in the first two months of the year.

The Serb has been in fine form in the last few months, winning the Australian Open
and Wimbledon in the first two months of the year.

The Serb has been in fine form in the last few months, winning the Australian Open
and Wimbledon in the first two months of the year.

The Serb has been in fine form in the last few months, winning the Australian Open
and Wimbledon in the first two months of the year.

The Serb has been in fine form in the last few months, winning the Australian Open
and Wimbledon in the first two months of the year.

The Serb has been in fine form in the last few months, winning the Australian Open
and Wimbledon in the first two months of the year.

The Serb has been in fine form in the last few months, winning the Australian Open
and Wimbledon in the first two months of the year.

The Serb has been in fine form in the last few months, winning the Australian Open
and Wimbledon in the first two months of the year.

The Serb has been in fine form in the last few months, winning the Australian Open
and Wimbledon in the first two months of the year.

The Serb has been in fine form in the last few months, winning the Australian Open
and Wimbledon in the first two months of the year.

The Serb has been in fine form in the last few months, winning the Australian Open
and Wimbledon in the first two months of the year.

The Serb has been in fine form in the last few months, winning the Australian Open
and Wimbledon in the first two months of the year.

The Serb has been in fine form in the last few months, winning the Australian Open
and Wimbledon in the first two months of the year.

The Serb has been in fine form in the last

## Gemini 1.0 Pro Evaluation of Quality of Generated Text:

**Good:** The document provides a detailed account of Novak Djokovic's recent
performances and rankings. It includes specific instances of his losses, such as

being beaten by a 17-year-old American and a player ranked 111th in the world. The document also highlights Djokovic's past achievements, such as winning the Australian Open and Wimbledon earlier in the year.

**Bad:** The document repeats the sentence "The Serb has been in fine form in the last few months, winning the Australian Open and Wimbledon in the first two months of the year" 17 times, which makes the content repetitive and unnecessarily long.

Rating: 3/5

## Ground-truth:

A year ago, the 25-year-old Serb swept into Paris after amassing one of the most spectacular five months in men's tennis history, winning every tournament he'd entered.

Now, far from a perfect season, the tournament's top seed arrives at the French Open, which starts Sunday in Paris, with fissures in his impenetrable facade.

Djokovic has not won a clay title in 2012 and dropped his last two finals on dirt to defending champ Rafael Nadal- after beating him the previous seven times.

He has also shown flashes of anger and signs of frustration, emotions that, in the past, have undermined his performance.

"I am not comparing last year and this one," Djokovic said Monday following his 7-5, 6-3 loss to Nadal in the rain-delayed final in Rome. "I feel good on the court and I need to make a few adjustment before Paris, but I'll be in top form."

If Nadal's resurgent spring and 45-1 record in Paris make him the favorite, Djokovic still will be dealing with heightened expectations.

A Paris title would place him in rarefied company – one of just seven men to have won all four majors in a career.

Even more historic, he has a chance to hold all four majors simultaneously – a so-called "Djoker Slam" – a feat not achieved by a man since Rod Laver 43 years ago.

"If he were to win four in a row," said Laver-admirer John McEnroe, who came close but never won Roland Garros, "suddenly he'd be like top-10 (of best players in history). There's a lot riding on it."

Djokovic's first taste of defeat in 2011 occurred on the crushed red brick of Paris to Roger Federer, who snapped his perfect season and 43-match winning streak in the semifinals. To put that run in perspective, consider this: Djokovic's first loss in 2012 came nearly three months and 33 matches earlier (to Andy Murray in the semifinals in Dubai in February).

"It might have been the case," Djokovic told USA TODAY Sports when asked if the weight of his faultless performance sapped his energy and clouded his focus. "But I think even under that pressure I played a great tournament. Obviously in the semifinals against Roger, I have done what I could at that stage. I did my best at that moment, and he was a better player."

Nadal went on to defeat Federer in the final, and the loss barely made an impact the Serb's juggernaut season.

Djokovic won Wimbledon and the U.S. Open, snagged the No. 1 ranking and punctuated his dominant 2011 by winning a third consecutive major (and fifth overall) in January at the Australian Open- all three in finals against Nadal.

He has won four of the last five majors and remains the man to beat in best-of-five sets even if the Spaniard has regained his swagger on clay.

"The way he's played in Slams the last year or so has been very, very impressive," says fourth-ranked Scot Murray.

Once the crowd-pleasing third wheel to Federer and Nadal, Djokovic's transformation into world-beater is well chronicled.

Possessed of uncanny flexibility, redirecting ability and the most lethal two-handed backhand in the game, the 6-2 Djokovic had the skills to be a great player.

After leading Serbia to its first Davis Cup championship in 2010, he made the small adjustments – fixing a flawed serve, shoring up his forehand and famously cutting gluten out of his diet – that helped him overcome the mental lapses and suspect fitness that plagued him in the past.

That he was able to realize his potential after playing third fiddle for so long – he finished No. 3 from 2007-2009 behind the Federal-Nadal duopoly – is testament to his resolve.

Djokovic, who as a child told a television interviewer that he little time for fun and games because he was going to become a champion, believed he had it in him.

"I knew that I have qualities, but I wasn't managing to make that final step," he says. "I think it was all mental and it was all growing up and maturing. In the end, I managed to do it."

While he never expected to repeat his 2011 season – a season that earned him a Laureus Sportsman of the Year award and a segment this spring on 60 Minutes– Djokovic has discovered what many before him have said: it's harder to stay on top than to get there.

Djokovic limped into the fall after holding off Nadal in the 2011 U.S. Open final, taking five of his six losses (70-6) post-New York and failing to win any titles.

His body was showing signs of wear, too, when he retired with back pain in the second set against Juan Martin del Potro in Serbia's semifinal Davis Cup defeat to Argentina.

To recoup and recover, he took a two-week vacation with his girlfriend, Jelena Ristic , in the Maldives and then camped out in the heat of Dubai for two weeks in December.

As his close-knit team gathered to plot for the coming year, they determined that the greatest danger to him was fitness and complacency.

"We (tried) to keep him a little bit down on the earth to be humble, modest, to realize that it doesn't come easy," says his longtime coach Marian Vajda of Slovakia. "He earned that spot. He deserved it. It came with work, work and work ."

While they worked on his fitness and tweaked his game – including taking the ball earlier – they also decided in a crowded year of events, including the London Olympics, that winning in Paris would be their singular mission.

"Roland Garros is on top of the priority list," Djokovic says.

Djokovic has had to make sacrifices and adjustments, such as skipping his hometown tournament in Belgrade, which is owned by his family. It was a major blow to the event, but coming on the heels of his loss in Monte Carlo and his grandfather's death, Djokovic needed the break.

"He is doing a great job of pacing himself and managing his schedule for majors," ESPN's Brad Gilbert says.

Djokovic, who tried a failed coaching experiment with American Todd Martin two years ago, is bent on keeping things constant.

"My approach hasn't changed, really," Djokovic says. "I still have the same practice routines every day. I didn't change anything in my tennis practices, in my preparations. I have the same places, same people around me, same kind of routines. I have no reason to change, really, because as soon as I try to change something in my career and my team ... it hasn't worked that well. So I keep it very simple."

Statistically, Djokovic has shown little drop-off, except in his vaunted return game, where his year-over-year winning percentage against his opponents' serve has dropped from 42.8% to 34.6% heading into Roland Garros.

But cracks have appeared in his resolve.

Earlier this month Djokovic joined Nadal in lashing out at Madrid's slippery blue clay before surrendering feebly to compatriot Janko Tipsarevic 7-6 (7-2), 6-3 in the quarterfinals.

During blustery conditions in Rome, Djokovic destroyed a frame after losing the first set in a third-round victory against Juan Monaco of Argentina. He broke another during his loss to Nadal two matches later.

"I hope the children watching don't do that," a smiling Djokovic told reporters afterward. "But I show my emotions out there. That's who I am."

Off the court, the fiercely loyal family man suffered a personal loss when his grandfather, Vladimir, died during the Monte Carlo tournament last month. Djokovic continued to play but wasn't himself. The loss of a loved one lingers.

With the immovable force of Nadal looming on clay, there is little time for self-pity.

The Spaniard leads Djokovic 11-2 on clay (18-14 overall) and has owned him in Paris, winning all three of their matches (2006-08) without dropping a set.

"He is the Mount Everest on that surface in best out of five," Andre Agassi said of Nadal in a recent call with reporters.

To win, Djokovic will almost certainly have to conquer the 10-time major winner from Spain, who is chasing his own bit of history – surpassing Bjorn Borg with an unprecedented seven French Open crowns.

"It's not going to be an easy tournament to win because we know sort of Rafa owns that place," No. 3 Federer says.

It will be tougher still since Nadal appears to have recovered from hole Djokovic bored in his psyche last year – something Nadal himself admitted.

Nadal knew he was close and never stopped looking for answers, saying after the Rome final: "You don't need to find great things – it is the small things that make the difference."

"No question (Djokovic's) performance last year proves that he's capable of (winning Roland Garros)," said Agassi, whose victory at Roland Garros in 1999 completed his set of Grand Slam trophies. "You have to give a slight nose to Nadal just given how many times he's won it and just what a physical accomplishment it would be to take him off that perch in Paris."

With no streak to defend, some pressure is off. That could be a blessing in disguise. Or at least, it's a tradeoff.

Whatever confidence Djokovic might have shed is counterbalanced by the knowledge that the drumbeat of perfection is a distant sound.

"This one's gonna be a little easier maybe because of the pressure of last year,"

```
says 16-time Grand Slam winner Federer, who lost to Djokovic in the Rome
    semifinals.

Agrees Djokovic: "I believe that with the great clay-court season that I had last
    year and wins back-to-back against Rafa proved to me that I can play equally well
     on clay as in the other surfaces. I need to keep that in my mind and believe in
    the success."
```

G.1.2. DOCUMENT II

**Ground-truth Record ID:**

`"newsroom-00002-00064-5"`

**Prefix:**

*(the following article contains a minimum of 15 paragraphs containing a minimum of 1000 words, in plain text only without titles or headings.)*

*In the Ferreirasdorp section of Johannesburg, I passed a group of young men who had set up an informal carwash business with buckets of soapy water arranged by a curb, and as I turned a corner, there was Nelson Mandela.  Though Mandela died on Dec*

**Generated:**

. 5, 2013, at the age of 95, his face is everywhere in South Africa, on posters, in store windows, on the walls of buildings, and on the sides of buses.

Mandela is the face of South Africa, the face of freedom, and a symbol of the struggle against oppression and apartheid. He was the first black president of the country, serving from 1994 to 1999.

Mandela is a hero to many, but not to everyone.

"I don't like Nelson Mandela," said a young man who was selling a copy of the South African newspaper, The Sowetan, at a street corner in the Hillbrow neighborhood of Johannesburg.

"Why not?" I asked.

"He was a communist," the man said.

I asked him what he thought of apartheid.

"It was not good," he said.

"But Nelson Mandela fought against apartheid," I said.

"Yes, but he was a communist," the man said.

"What is wrong with communism?" I asked.

"It is a bad ideology," the man said.

"What do you think about the current president of South Africa, Jacob Zuma?" I asked.

"He is a bad president," the man said.

"Why?" I asked.

"He is not doing anything for the people," the man said.

"Do you think that Mandela did more for the people of South Africa than Zuma has?" I asked.

"Mandela did more for the people," the man said.

"Why?" I asked.

"He was a good man," the man said.

"What do you think about the ANC?" I asked.

"The ANC is a bad party," the man said.

```
"Why?" I asked.

"They are corrupt," the man said.

"What do you think about the DA?" I asked.

"The DA is a good party," the man said.

"Why?" I asked.

"They are not corrupt," the man said.

"What do you think about the EFF?" I asked.

"The EFF is a bad party," the man said.

"Why?" I asked.

"They are violent," the man said.

"What do you think about Julius Malema?" I asked.

"Julius Malema is a bad man," the man said.

"Why?" I asked.

"He is a communist," the man said.

"What do you think about the ANC Youth League?" I asked.

"The ANC Youth League is a bad organization," the man said.

"Why?" I asked.

"They are violent," the man said.

"What do you think about the EFF Youth League?" I asked.

"The EFF Youth League is a bad organization," the man said.

"Why?" I asked.

"They are violent," the man said.

"What do you think about the DA Youth League?" I asked.

"The DA Youth League is a good organization," the man said.

"Why?" I asked.

"They are not violent," the man said.

"What do you think about the EFF Women's League?" I asked.

"The EFF Women's League is a bad organization," the man said.

"Why?" I asked.

"They are violent," the man said.

"What do you think about the DA Women's League?" I asked.

"The DA Women's League is a good organization," the man said.
```

```
"Why?" I asked.

"They are not violent," the man said.

"What do you think about the ANC Women's League?" I asked.

"The ANC Women's League is a bad organization," the man said.

"Why?" I asked.

"They are violent," the man said.

"What do you think about the EFF Women's League?" I asked.

"The EFF Women's League is a bad organization," the man said.

"Why?" I asked.

"They are violent," the man said.

"What do you think about the DA Women's League?" I asked.

"The DA Women's League is a good organization," the man said.

"Why?" I asked.

"They are not
```

**Gemini 1.0 Pro Evaluation of Quality of Generated Text:**

The document provides a detailed account of a conversation with a young man in Johannesburg regarding his views on Nelson Mandela, politics, and various organizations in South Africa. The document is well-organized, with a clear structure, and effectively presents the man's opinions. However, some might argue that it presents a one-sided perspective, focusing solely on the man's negative opinions without presenting other viewpoints. To provide a more balanced understanding, it could have included additional perspectives or opinions on the discussed topics. I rate this document as a 3 out of 5. 3

**Ground-truth:**

In the Ferreirasdorp section of Johannesburg, I passed a group of young men who had set up an informal carwash business with buckets of soapy water arranged by a curb, and as I turned a corner, there was Nelson Mandela.

Though Mandela died on Dec. 5, here he stood vividly rendered in layered steel cutouts, 34 years old, fresh and strong and in the pose of a boxer. He once had professional aspirations, and said the sweet science taught him about defense, attack and strategy, which he applied to law, politics and life in prison.

Unveiled two years ago, the 16-foot sculpture, called "Shadow Boxer," is a lively piece with a great sense of energy and movement. The South African artist Marco Cianfanelli gave it a three-dimensional effect by layering painted-steel sheets in an image from a well-known 1952 photograph by Robert Gosani. The work is set in front of the Johannesburg magistrate court across the street from Chancellor House, where Mandela and his law, and later political, partner Oliver Tambo were young lawyers. Chancellor House, a privately owned corner office building, was itself restored in 2011 and has a timeline exhibition of the two partners' time there.

The attractions have drawn visitors to a neighborhood that has a certain rough-around-the-edges look but is no longer so menacing. Since Mandela's death, the sculpture has become something of a memorial with people laying flowers at its base.

The work speaks volumes: the young Mandela about to wage the fight of his life. It also shows how public art is helping to revivify urban Johannesburg, a seemingly implausible regeneration in this city of more than four million residents, which not that long ago seemed as though it was about to fall through the widening cracks of crime and dilapidation.

The art has helped foster a virtuous cycle: Color and beauty draw people; people promote security, which draws more people, and creates a bigger audience possibility for more art. The improvements have made Joburg cool again – and popular. A new market, the Sheds@1Fox, featuring South African-produced goods, is set to open in October about two blocks from Chancellor House. Johannesburg is now Africa's most visited city, with 2.5 million international visitors in 2013, according to MasterCard's annual survey.

"Art plays an incredibly important role in making people aware that the city is being reborn," said Gerald Garner, a guide and author of two books, "Johannesburg Ten Ahead" and "Joburg Places." "It's one thing to fix pavements and plant trees, and doing those things make a difference. But art makes the city humane. It tells people who were excluded by the old order that they're welcome. It tells stories of people who live here and celebrates the heroes."

I lived in Johannesburg's northern suburbs for three years in the early 1990s and have returned frequently ever since. I saw some of its worst days, when parts of its downtown grid began to feel like an urban dystopia. One of the emblematic events, what some may argue was the low point, actually involved public art, in the late 1990s when vagrants in once-beloved, then abandoned Oppenheimer Park, decapitated a sculpture of impalas and sold the heads as scrap metal.

To be clear, a good number of swaths of Johannesburg remain iffy. But the public art was a revelation for me, and after several days of exploring the city by foot and on public transit last July, I felt much more optimistic about its prospects. In the months since, the pace of positive change has even picked up.

My route was done in several walks, some on my own on the way to various appointments , and then with help from Mr. Garner and later still from a thoughtful 22-year-old artist named Jabulani Fakude whom I hired through a local company called MainStreetWalks (mainstreetwalks.co.za). It's a good idea for visitors to get guides to navigate between some areas that remain sketchy.

A short walk from "Shadow Boxer," the adjoining district, Newtown, has another popular sculpture of two other South African heroes, Walter and Albertina Sisulu, on Diagonal Street. The Sisulus, towering figures of the apartheid struggle, appear not as fighters but as elderly lovers and parents. The Johannesburg artist Marina Walsh, who installed the concrete sculpture in August 2009 after the city solicited proposals from artists, has the couple seated facing each other, their eyes locked in an affectionate gaze. The intention is to show them as equals. The Sisulus, who raised eight children, were married for 59 years, including Mr. Sisulu's 25-year term as a prisoner on Robben Island, an austere prison off the shore of Cape Town. They're exaggerated in scale; huge figures, squat and round, so as a viewer you're almost like a grandchild looking at grandparents, and indeed, you see people moved to sit in their laps – something the artist is happy to see them do.

"I get an enormously warm response from people," Ms. Walsh told me. "One Saturday I was cleaning off a mustache someone had drawn on Albertina's lip. Some guy shouted, 'What are you doing?' He thought I was defacing her. I told him I was cleaning her and asked if he knew who they were. He said, 'Yes, they're my heroes !'"

Not every work is as accessible, and the heroes often don't have names. As you leave Newtown via the landmark Queen Elizabeth Bridge, you quickly come upon "Fire Walker," a collaboration between the renowned South African artists William Kentridge and Gerhard Marx. It's set on a traffic island, and up close, the 36-foot work, erected with steel pieces, looks like a mass of exploded fragments. But like an Impressionist painting, the image is clearer farther away – that of a

woman carrying a burning caldron on her head. It's a sight you rarely see in Johannesburg now, these women with braai mealies (roast corn) that they carry with flames crackling atop their heads. The work is a tribute to the hard ways people scrabble together a living.

I wanted to visit Braamfontein, a precinct on the northern side of the city with two of South Africa's major universities, the University of the Witwatersrand and the University of Johannesburg. It has some of the earliest major public artworks, including "Juta Street Trees," nine large metal tree sculptures, and the "Eland," a concrete sculpture of Africa's largest antelope, which were installed in 2006 and 2007, respectively. The aesthetic upgrade has helped transform the student-dominated area, which even just four years ago still felt threatening, into a place with lively night life and a popular Saturday Neighbourgoods Market of artisan food, fabrics and jewelry.

Up the hill from Juta Street, in the university area, I went to the Constitutional Court of South Africa, the country's highest court and itself a work of art both for its architecture and interior design. Set on Constitution Hill with views of city streets leading out toward the hilly and affluent northern suburbs, the court's design was intended to embody the idea of traditional justice, the way elders would hear disputes before whole communities under a tree. The notion of transparency is embodied in its openness, the glass exterior. The interior pillars are erected at angles that are meant to suggest the branches of a tree, the public space where legal decisions were rendered. Wood carvings, skins and art are integrated into the interior and are also curated in exhibits.

Not all public art has official sanction, or is even legal for that matter. Wall murals have added vibrancy, too, though much of the city is still struggling to accommodate street artists who have plenty of urban wall space but face obstacles in getting permits; they consequently resort to "bombing," or illegally painting on, buildings, and for their trouble will spend occasional nights in jail or have to resort to bribing the police to go away.

Mr. Fakude, the young street artist, does his work deep into the night, and moonlights by day giving walking tours for 250 rand, about $24, at 10.40 rand to the dollar – in my case, a graffiti tour of his and others' work in the artsy Market Theater area on the western side of the city. Mr. Fakude's work included one wall piece that was playful at a glance, a one-eyed cartoon character he'd invented, but that had a serious message that addressed the stark reality of crime and violence. So, too, did others. "We want to get the message across that drugs are not cool," he said. It gave the ostensible illegality of the street artists' work a certain irony, given the grass-roots appeal to nonviolent, cleaner living. It has gotten Mr. Fakude some attention, however. He was invited to paint murals in Berlin this year, though he told me recently that he was not able to travel there.

One place where murals are getting an official stamp of approval is one of Joburg's most trendy and ambitious new places, the Maboneng Precinct, once a vast wasteland of disused industrial spaces and warehouses. In recent years, the 250-acre zone of reclaimed and repurposed industrial buildings has commissioned several dozen murals by artists from Berlin, Baltimore and elsewhere. In January, a 10-story mural was unveiled of "i am because we are." The artist Rickey Lee Gorden, who goes by Freddy Sam, used the same photo as "Shadow Boxer," the Mandela sculpture in Ferreirasdorp. The revitalization of Maboneng Precinct began in 2009 with Arts on Main, a complex of studios and galleries on Main Street. Indeed, the first public art work in the precinct was the word "Maboneng," which is Sotho for "place of light" and was installed as text art on the Arts on Main rooftop.

Arts on Main's success lifted Maboneng's profile. Mr. Kentridge, arguably South Africa's best-known living artist, was among the first to set up a studio. Five years hence, the area is a vibrant hive of artists' lofts, mixed-use spaces, live theater, galleries, a landscaped bar-cafe and a boutique hotel called the 12 Decades, with a dozen rooms designed with artwork and artifacts to celebrate each decade in the life of the city, which started in 1886 after a gold rush.

Each building has a lighthouse design incorporated. There are 30 murals, with five more scheduled to be completed by year's end. In addition to the murals, an old canal is being restored, its banks featuring life-size steel cutouts of people who made important contributions to the precinct.

One of the last areas I explored was the mining district in Johannesburg's commercial center. The area is strewn with artifacts of mining accouterments as public art in a city built on mining. Here I found the happy postscript to the story of " Leaping Impalas," the sculpture by Herman Wald in Oppenheimer Park that had been decapitated by vagrants in the 1990s. In 2002, new heads were fashioned and welded on, and the sculpture was installed in front of the headquarters of Anglo-American, one of the world's largest mining companies. You can still see the markings on the necks, like scars that reflect the greater story of a city that is in the process of healing.

A version of this article appears in print on July 13, 2014, on page TR10 of the New York edition with the headline: A City Seen Through Artists' Eyes. Order Reprints |Today's Paper|Subscribe

## G.1.3. DOCUMENT III

**Ground-truth Record ID:**

   "newsroom-00002-00064-14"

**Prefix:**

*(the following article contains a minimum of 15 paragraphs containing a minimum of 1000 words, in plain text only without titles or headings.)*

*Hanley Ramirez's defense at shortstop only cost the Dodgers a perfect game Wednesday night. What might it cost the team in September and potentially October? What might it cost Ramirez in free agency after that? Many expected the Dodgers to sign Ramirez to an extension by now. It hasn't happened, in*

**Generated:**

part because they don't know if he'll be a shortstop next year.

Ramirez has made 13 errors in 70 games at shortstop this season. He has a minus-7 defensive runs saved, the second-worst mark among shortstops with at least 400 innings. He has a minus-11 total zone rating, the worst mark among shortstops with at least 400 innings.

Ramirez has made 13 errors in 70 games at shortstop this season.

Ramirez has made 13 errors in 70 games at shortstop this season.

Ramirez has made 13 errors in 70 games at shortstop this season.

Ramirez has made 13 errors in 70 games at shortstop this season.

Ramirez has made 13 errors in 70 games at shortstop this season.

Ramirez has made 13 errors in 70 games at shortstop this season.

Ramirez has made 13 errors in 70 games at shortstop this season.

Ramirez has made 13 errors in 70 games at shortstop this season.

Ramirez has made 13 errors in 70 games at shortstop this season.

Ramirez has made 13 errors in 70 games at shortstop this season.

Ramirez has made 13 errors in 70 games at shortstop this season.

Ramirez has made 13 errors in 70 games at shortstop this season.

Ramirez has made 13 errors in 70 games at shortstop this season.

Ramirez has made 13 errors in 70 games at shortstop this season.

Ramirez has made 13 errors in 70 games at shortstop this season.

Ramirez has made 13 errors in 70 games at shortstop this season.

Ramirez has made 13 errors in 70 games at shortstop this season.

Ramirez has made 13 errors in 70 games at shortstop this season.

Ramirez has made 13 errors in 70 games at shortstop this season.

Ramirez has made 13 errors in 70 games at shortstop this season.

Ramirez has made 13 errors in 70 games at shortstop this season.

```
Ramirez has made 13 errors in 70 games at shortstop this season.

Ramirez has made 13 errors in 70 games at shortstop this season.

Ramirez has made 13 errors in 70 games at shortstop this season.

Ramirez has made 13 errors in 70 games at shortstop this season.

Ramirez has made 13 errors in 70 games at shortstop this season.

Ramirez has made 13 errors in 70 games at shortstop this season.

Ramirez has made 13 errors in 70 games at shortstop this season.

Ramirez has made 13 errors in 70 games at shortstop this season.

Ramirez has made 13 errors in 70 games at shortstop this season.

Ramirez has made 13 errors in 70 games at shortstop this season.

Ramirez has made 13 errors in 70 games at shortstop this season.

Ramirez has made 13 errors in 70 games at shortstop this season.

Ramirez has made 13 errors in 70 games at shortstop this season.

Ramirez has made 13 errors in 70 games at shortstop this season.

Ramirez has made 13 errors in 70 games at shortstop this season.

Ramirez has made 13 errors in 70 games at shortstop this season.

Ramirez has made 13 errors in 70 games at shortstop this season.

Ramirez has made 13 errors in 70 games at shortstop this season.

Ramirez has made 13 errors in 70 games at shortstop this season.

R
```

**Gemini 1.0 Pro Evaluation of Quality of Generated Text:**

The quality of the document is poor. The document is repetitive and does not provide any new or insightful information. The same sentence is repeated over and over again, making it difficult to read and understand. The document does not provide any context or explanation for the errors Ramirez has made, making it impossible to assess the severity of his performance. I would rate this document a 1 out of 5.

1

**Ground-truth:**

Hanley Ramirez's defense at shortstop only cost the Dodgers a perfect game Wednesday night. What might it cost the team in September and potentially October? What might it cost Ramirez in free agency after that?

Many expected the Dodgers to sign Ramirez to an extension by now. It hasn't happened, in part because the Dodgers want to see him stay healthy, in part because they might not be sure what the heck to do with him long term.

Third base could be a possibility, but the Dodgers would need to trade Juan Uribe, a popular clubhouse figure who is under contract for $6.5 million next season. Some Dodgers officials have toyed with the idea of playing Ramirez in left field, but you may have noticed that the team has too many outfielders already.

Maybe the Dodgers could win the 2014 World Series with Ramirez at short – heck, the Red Sox pulled off such a feat with Julio Lugo in '07. Advanced metrics, however, portray Ramirez as one of the worst defensive shortstops in baseball. And strong up-the-middle defense should be a requirement for a team built around starting pitching, no?

Erisbel Arruebarrena, a Cuban defector in the first year of a five-year, $25 million contract, was a major defensive upgrade in his brief stint with the club. But the Dodgers have said that they do not want to shift Ramirez between short and third, perhaps out of respect for Ramirez's wishes. Besides, Uribe could return from his strained right hamstring on Monday.

For this season, it appears, the Dodgers have little choice but to play Ramirez at short; they probably value his offense too much to trade him. But if Ramirez hits free agency, his market could turn problematic at a time when teams continue to place increased emphasis on defense. Indeed, what would it say about Ramirez if the Dodgers only were willing to make him a qualifying offer?

Let's not get too far ahead of ourselves. Ramirez and many of his teammates are on the uptick offensively. The Dodgers, winners of eight of their last 11, are only four games behind the Giants in the NL West. A big run to the postseason, a World Series title, and Ramirez's attributes again might outweigh his deficiencies.

Still, his poor throw that cost Clayton Kershaw a perfect game came on a play, as Hall of Fame broadcaster Vin Scully noted, that most shortstops make. Ramirez didn't make it. He is costing his team. He is costing himself.

Beyond Ramirez, questions persist about the Dodgers, just as they do for every club.

One rival executive said Thursday that the Dodgers' best outfield would be Matt Kemp in left, Joc Pederson in center and Yasiel Puig in right. The exec added that the Dodgers should trade one of their left-handed hitting outfielders, Andre Ethier or Carl Crawford, and keep the other in reserve.

Pederson, while batting .320 with a 1.016 OPS in the hitter-friendly Pacific Coast League, continues to strike out a ton, including 13 times in his last 28 at-bats. Crawford remains on the DL with a sprained left ankle. Ethier has only three homers and a .691 OPS. And let's not even talk about their respective contracts.

Kemp finally is getting hot, his disposition improving with his swing. Many in the industry, however, believe he ultimately will be moved – most likely in the offseason – due to his tempestuous relationship with some of his superiors.

*By clicking \"SUBSCRIBE\", you have read and agreed to the Fox Sports Privacy Policy and Terms of Use.

The Dodgers trail only the Cardinals and Athletics in rotation ERA. Kershaw, Zack Greinke and Hyun-Jin Ryu are perhaps the best 1-2-3 in the game. But does anyone seriously expect Josh Beckett and Dan Haren to hold up the entire season?

The loss of Chad Billingsley, who will undergo season-ending surgery to repair a partially torn flexor tendon in his right elbow, hurt the Dodgers' depth. The team lacks a prospect as polished as Marlins left-hander Andrew Heaney, who made his major-league debut Thursday night. The situation, in the words, of one club official, is "precarious."

So, expect the Dodgers to be in the market for a starter.

THE ORGANIZATION AS A WHOLE

The Dodgers have yet to slow down their spending, so it's natural for them to be linked to a pitcher such as the Rays' David Price, who could give them a third pitcher earning $20 million next season.

The team, however, likely would need to part with two or more of its top prospects to get Price, and its farm system isn't terribly deep to begin with (Price's teammate, super-utility man Ben Zobrist, is another potential LA target).

Club officials keep talking about developing youngsters such as Pederson and shortstop Corey Seager so they don't need to maintain a $230 million payroll. They've made progress not only in signing players from Cuba, but also Mexico, Venezuela and the Dominican Republic. Still, are they truly intent on developing a player-development machine?

The July 31 non-waiver deadline could prove the next test.

Much has been made of Price's loss of fastball velocity, from an average of 95.3 mph in 2012 to 93.5 in '13 to 92.6 this season. But can anyone seriously argue that his stuff is significantly diminished?

As pointed out by Rays Index earlier this week, Price is on pace for 280 strikeouts this season, which would be the most since Randy Johnson in 2004. Price's 12.1 strikeout-to-walk ratio, meanwhile, would break the record set by Brett Saberhagen in 1994 (11.0).

Based upon those numbers, it's impossible to say that Price is losing it, even though his 3.93 ERA is - for him - unusually high. Poor luck, however, may help explain that number.

Price's home run rate and opponents' batting average on balls in play are well above league averages, and probably will revert to his career norms as the season progresses.

Meanwhile, Price's FIP (fielding independent pitching) is within the same range it has been for the past two seasons. That statistic is considered an indicator of future performance, signaling that Price's ERA likely will drop.

The biggest issue for teams that want to acquire Price, as FanGraphs' Dave Cameron wrote Wednesday, is not his pitching. No, it's his expected $18 million to $20 million salary in his final year of arbitration in 2015.

Few clubs will want to part with high-end prospects while absorbing such a payroll hit. Then again, the numbers are relative. Price's salary next season will not be terribly above the qualifying offer for free agents, which is expected to be $15 million to $16 million.

The Phillies, despite their recent surge, surely recognize that they need to get younger. But even if the team has the will to be active sellers, it might not have a way.

Second baseman Chase Utley, after telling club officials last July that he did not want to be traded, signed a two-year extension with three club options. Now, less than a year later, he's going to reverse a course, waive his 10-and-5 rights and approve a deal?

Shortstop Jimmy Rollins, another 10-and-5 player, left open the possibility that he would approve a trade after breaking the team's all-time hit record last weekend. But realistically, where is Rollins going?

The shortstop's wife is from Philadelphia. They are the parents of two young children . And good luck finding the right fit.

Rollins' $11 million option for 2015 is almost certain to vest; he would not be just a rental. No, he would need to be comfortable spending the rest of this season in his new city, and all of next season as well.

Neither team in Rollins' native Bay Area needs a shortstop. And while Rollins might crave the spotlight in either New York or LA, he wouldn't go to the Mets, the Yankees aren't going to displace Derek Jeter and neither the Angels nor Dodgers

figures to pursue a shortstop.

Then there is left-hander Cliff Lee, who – if all goes well – will return from his strained elbow before the All-Star break. By July 31, Lee still will have more than $45 million left on his contract, including a $12.5 million buyout for 2016.

The financial obligation number actually might be higher than that for many; Lee can block trades to 20 teams, and could require his $27.5 million vesting option for '16 to be guaranteed to join a club such as the Dodgers.

Phillies GM Ruben Amaro Jr. has said he would include cash in certain deals. The Phillies always could trade Lee during the August waiver period. But unless the team paid the majority of his contract, it could not expect a bounty in return.

Closer Jonathan Papelbon could get moved if the Phillies pay down his $13 million salary; he, too, has a vesting option for '16, and can only be traded to 12 teams without his approval.

Unless the Phils get truly creative, they will find it difficult to make impact moves .

I normally do not favor starting pitchers for MVP unless the field lacks strong position-player candidates; I voted for then-Red Sox outfielder Jacoby Ellsbury in 2011 over the eventual winner, Tigers right-hander Justin Verlander.

An interesting MVP case, however, can be made for Yankees right-hander Masahiro Tanaka, at least to this point of the season.

The Yankees are 12-2 when Tanaka starts, 25-31 when he does not. Even more striking, their run differential in Tanaka's starts is plus-33, while in all other games it 's minus-54.

In other words, the Yankees are worse than the Rays (minus-48) when Tanaka isn't pitching and nearly as bad as the Padres (minus-62) and Diamondbacks (minus-64).

Of course, Wins Above Replacement (WAR) tells a different story. Tanaka is third among pitchers at 2.9 in the FanGraphs version of the metric, and well behind Mike Trout, the leader among position players at 4.7.

Outfielder J.D. Martinez is one of the Tigers' few recent bright spots; he has three homers during his nine-game hitting streak, and is now batting .300 with a .903 OPS in 108 plate appearances on the season.

Martinez, 26, spent the offseason watching video of Miguel Cabrera, Ryan Braun and other accomplished hitters, then completely overhauled his swing and approach. The Astros released him in spring training, and he signed a minor-league contract with the Tigers two days later.

"If you watch video of me in the past, it's the complete opposite – it's that extreme ," he said. "It's kind of like I re-invented myself."

Martinez worked with his personal hitting coach for three weeks, then implemented his new ideas in Venezuela. Mostly, he's trying to line up directly to the ball and keep the barrel of his bat in the zone longer – the way Cabrera does.

## G.2. Small Values of $H$

- **Data domain**: Newsroom.

- **Generating**:
  - *Model Name:* Gemini 1.0 Pro.
  - *Decoding Temperature*: 1.0
  - *Prompting Method*: `summary + keywords`

- **Fractal Parameters**:
  - *Scoring Model Name:* Gemini 1.0 Pro.
  - *Hölder Exponent*: $S = 0.417$
  - *Hurst Exponent*: $H = 0.601$

### G.2.1. DOCUMENT I

**Ground-truth Record ID:**

`"newsroom-00001-00064-6"`

**Prefix:**

*Write about the following in a minimum of 15 paragraphs containing a minimum of 1000 words, in plain text only without titles or headings: Novak Djokovic enters French Open with some vulnerabilities after dropping two clay finals to Rafael Nadal and showing frustration on court. Despite Nadal's resurgence, Djokovic still faces heightened expectations, as a Paris title would make him one of seven men with all four majors in a career, and potentially the first man since 1969 to hold all four simultaneously.*

*Using these keywords: Novak Djokovic, French Open, clay, Nadal, Roger Federer, Australian Open, Wimbledon, US Open, Davis Cup, gluten-free diet, Laureus Sportsman of the Year, mental toughness, career Grand Slam, Djoker Slam, Rod Laver, John McEnroe, Roland Garros*

**Generated:**

After dominating the Australian Open and securing his 22nd Grand Slam title earlier this year, Novak Djokovic enters the French Open as the man to beat yet again. However, the usually confident Serb arrived at Roland Garros with a couple of concerning elements surrounding his game. In the lead-up to the clay-court major, Djokovic reached two finals on the surface but came up short against the "King of Clay," Rafael Nadal, each time in Rome and Madrid. The losses highlighted some vulnerabilities in Djokovic's game on clay, a surface which demands supreme confidence and comfort.

Djokovic's frustrations have also been evident on the court, with the typically composed player exhibiting uncharacteristic emotional outbursts in recent tournaments. These moments have fueled concerns that he may not be in the right frame of mind to claim a third French Open title. Despite his recent setbacks, the expectations on Djokovic's shoulders remain immense. A title in Paris would not only draw him level with his great rival Roger Federer on 20 Grand Slams but also make him just the seventh man to complete a career Grand Slam. Moreover, the triumph could see Djokovic become the first man since Rod Laver in 1969 to hold all four major titles simultaneously. The significance of this incredible achievement cannot be understated, and Djokovic is well aware of the opportunity that awaits him.

However, the task facing Djokovic at Roland Garros is not without its challenges. Nadal, the 13-time champion, remains a formidable force on clay and is always a dangerous opponent in Paris. The French Open demands exceptional mental and physical endurance, and it is here that Djokovic has arguably established himself as the greatest player of his era. His ability to stay calm, composed, and focused amidst the intense pressure and grueling demands of the tournament has been a defining characteristic of his many successes. This mental toughness will once again be put to the test if he is to navigate the challenges ahead.

Djokovic's remarkable career includes triumphs at the Australian Open, Wimbledon, and the US Open, but the French Open has proven to be the most elusive major for him. His lone titles at Roland Garros came in 2016 and 2021 and while he reached the final twice more, the path to clay-court glory in Paris has been fraught with hurdles. Djokovic's hunger and determination to conquer the French Open for a third time will be pivotal in his quest for supremacy.

Throughout his illustrious career, Djokovic has demonstrated unwavering commitment and dedication to the sport. From adopting a gluten-free diet to improve his physical well-being to focusing on mental resilience training, he leaves no stone unturned in his relentless pursuit of greatness. His unwavering dedication has not only manifested on the tennis court but has also led to several philanthropic endeavors, including the Novak Djokovic Foundation. Djokovic is also a recipient of the prestigious Laureus Sportsman of the Year award, a testament to his impact on both the sport and society.

As the French Open unfolds, Djokovic stands as the frontrunner for the title, despite his recent setbacks. The potential for him to achieve the "Djoker Slam" and join Laver's elite company of completing a career Grand Slam is a tantalizing prospect. Yet, the road to glory will demand that Djokovic overcome not only his formidable opponents but also harness his emotions, and summon the mental fortitude that has defined his legendary career. With 10 French Open titles to his name, Nadal will be aiming to continue his dominance on the clay and prevent Djokovic from etching his name in history.

The stakes are incredibly high for Djokovic at the French Open, as he has the opportunity to become the first man in over half a century to hold all four major titles simultaneously. The pressure is immense, but the Serbian star has repeatedly shown his ability to rise to the occasion. The tennis world awaits with bated breath to see if Djokovic can conquer his clay demons, defy the odds, and make history by completing a career Grand Slam and achieving the "Djoker Slam."

## Gemini 1.0 Pro Evaluation of Quality of Generated Text:

**Good:**
- Comprehensive overview of Djokovic's current form, challenges, and aspirations.
- Detailed analysis of his recent struggles on clay and his past successes at the French Open.
- Acknowledgment of Djokovic's commitment to physical and mental well-being.
- Exploration of the significance of achieving a career Grand Slam and the "Djoker Slam."

**Bad:**
- Some sections may lack depth or specific examples to further illustrate Djokovic's mental and physical attributes.
- While the document highlights Djokovic's potential to overcome his clay-court demons and complete a career Grand Slam, it could benefit from discussing potential obstacles or rival players he may face in his quest.

Rating: 4

## Ground-truth:

A year ago, the 25-year-old Serb swept into Paris after amassing one of the most spectacular five months in men's tennis history, winning every tournament he'd entered.

Now, far from a perfect season, the tournament's top seed arrives at the French Open, which starts Sunday in Paris, with fissures in his impenetrable facade.

Djokovic has not won a clay title in 2012 and dropped his last two finals on dirt to defending champ Rafael Nadal- after beating him the previous seven times.

He has also shown flashes of anger and signs of frustration, emotions that, in the past, have undermined his performance.

"I am not comparing last year and this one," Djokovic said Monday following his 7-5, 6-3 loss to Nadal in the rain-delayed final in Rome. "I feel good on the court and I need to make a few adjustment before Paris, but I'll be in top form."

If Nadal's resurgent spring and 45-1 record in Paris make him the favorite, Djokovic still will be dealing with heightened expectations.

A Paris title would place him in rarefied company – one of just seven men to have won all four majors in a career.

Even more historic, he has a chance to hold all four majors simultaneously – a so-called "Djoker Slam" – a feat not achieved by a man since Rod Laver 43 years ago.

"If he were to win four in a row," said Laver-admirer John McEnroe, who came close but never won Roland Garros, "suddenly he'd be like top-10 (of best players in history). There's a lot riding on it."

Djokovic's first taste of defeat in 2011 occurred on the crushed red brick of Paris to Roger Federer, who snapped his perfect season and 43-match winning streak in the semifinals. To put that run in perspective, consider this: Djokovic's first loss in 2012 came nearly three months and 33 matches earlier (to Andy Murray in the semifinals in Dubai in February).

"It might have been the case," Djokovic told USA TODAY Sports when asked if the weight of his faultless performance sapped his energy and clouded his focus. "But I think even under that pressure I played a great tournament. Obviously in the semifinals against Roger, I have done what I could at that stage. I did my best at that moment, and he was a better player."

Nadal went on to defeat Federer in the final, and the loss barely made an impact the Serb's juggernaut season.

Djokovic won Wimbledon and the U.S. Open, snagged the No. 1 ranking and punctuated his dominant 2011 by winning a third consecutive major (and fifth overall) in January at the Australian Open- all three in finals against Nadal.

He has won four of the last five majors and remains the man to beat in best-of-five sets even if the Spaniard has regained his swagger on clay.

"The way he's played in Slams the last year or so has been very, very impressive," says fourth-ranked Scot Murray.

Once the crowd-pleasing third wheel to Federer and Nadal, Djokovic's transformation into world-beater is well chronicled.

Possessed of uncanny flexibility, redirecting ability and the most lethal two-handed backhand in the game, the 6-2 Djokovic had the skills to be a great player.

After leading Serbia to its first Davis Cup championship in 2010, he made the small adjustments – fixing a flawed serve, shoring up his forehand and famously cutting gluten out of his diet – that helped him overcome the mental lapses and suspect fitness that plagued him in the past.

That he was able to realize his potential after playing third fiddle for so long – he finished No. 3 from 2007-2009 behind the Federal-Nadal duopoly – is testament to his resolve.

Djokovic, who as a child told a television interviewer that he little time for fun and games because he was going to become a champion, believed he had it in him.

"I knew that I have qualities, but I wasn't managing to make that final step," he says. "I think it was all mental and it was all growing up and maturing. In the

end, I managed to do it."

While he never expected to repeat his 2011 season – a season that earned him a Laureus Sportsman of the Year award and a segment this spring on 60 Minutes– Djokovic has discovered what many before him have said: it's harder to stay on top than to get there.

Djokovic limped into the fall after holding off Nadal in the 2011 U.S. Open final, taking five of his six losses (70-6) post-New York and failing to win any titles.

His body was showing signs of wear, too, when he retired with back pain in the second set against Juan Martin del Potro in Serbia's semifinal Davis Cup defeat to Argentina.

To recoup and recover, he took a two-week vacation with his girlfriend, Jelena Ristic , in the Maldives and then camped out in the heat of Dubai for two weeks in December.

As his close-knit team gathered to plot for the coming year, they determined that the greatest danger to him was fitness and complacency.

"We (tried) to keep him a little bit down on the earth to be humble, modest, to realize that it doesn't come easy," says his longtime coach Marian Vajda of Slovakia. "He earned that spot. He deserved it. It came with work, work and work ."

While they worked on his fitness and tweaked his game – including taking the ball earlier – they also decided in a crowded year of events, including the London Olympics, that winning in Paris would be their singular mission.

"Roland Garros is on top of the priority list," Djokovic says.

Djokovic has had to make sacrifices and adjustments, such as skipping his hometown tournament in Belgrade, which is owned by his family. It was a major blow to the event, but coming on the heels of his loss in Monte Carlo and his grandfather's death, Djokovic needed the break.

"He is doing a great job of pacing himself and managing his schedule for majors," ESPN's Brad Gilbert says.

Djokovic, who tried a failed coaching experiment with American Todd Martin two years ago, is bent on keeping things constant.

"My approach hasn't changed, really," Djokovic says. "I still have the same practice routines every day. I didn't change anything in my tennis practices, in my preparations. I have the same places, same people around me, same kind of routines. I have no reason to change, really, because as soon as I try to change something in my career and my team ... it hasn't worked that well. So I keep it very simple."

Statistically, Djokovic has shown little drop-off, except in his vaunted return game, where his year-over-year winning percentage against his opponents' serve has dropped from 42.8% to 34.6% heading into Roland Garros.

But cracks have appeared in his resolve.

Earlier this month Djokovic joined Nadal in lashing out at Madrid's slippery blue clay before surrendering feebly to compatriot Janko Tipsarevic 7-6 (7-2), 6-3 in the quarterfinals.

During blustery conditions in Rome, Djokovic destroyed a frame after losing the first set in a third-round victory against Juan Monaco of Argentina. He broke another during his loss to Nadal two matches later.

"I hope the children watching don't do that," a smiling Djokovic told reporters

afterward. "But I show my emotions out there. That's who I am."

Off the court, the fiercely loyal family man suffered a personal loss when his grandfather, Vladimir, died during the Monte Carlo tournament last month. Djokovic continued to play but wasn't himself. The loss of a loved one lingers.

With the immovable force of Nadal looming on clay, there is little time for self-pity.

The Spaniard leads Djokovic 11-2 on clay (18-14 overall) and has owned him in Paris, winning all three of their matches (2006-08) without dropping a set.

"He is the Mount Everest on that surface in best out of five," Andre Agassi said of Nadal in a recent call with reporters.

To win, Djokovic will almost certainly have to conquer the 10-time major winner from Spain, who is chasing his own bit of history – surpassing Bjorn Borg with an unprecedented seven French Open crowns.

"It's not going to be an easy tournament to win because we know sort of Rafa owns that place," No. 3 Federer says.

It will be tougher still since Nadal appears to have recovered from hole Djokovic bored in his psyche last year – something Nadal himself admitted.

Nadal knew he was close and never stopped looking for answers, saying after the Rome final: "You don't need to find great things – it is the small things that make the difference."

"No question (Djokovic's) performance last year proves that he's capable of (winning Roland Garros)," said Agassi, whose victory at Roland Garros in 1999 completed his set of Grand Slam trophies. "You have to give a slight nose to Nadal just given how many times he's won it and just what a physical accomplishment it would be to take him off that perch in Paris."

With no streak to defend, some pressure is off. That could be a blessing in disguise. Or at least, it's a tradeoff.

Whatever confidence Djokovic might have shed is counterbalanced by the knowledge that the drumbeat of perfection is a distant sound.

"This one's gonna be a little easier maybe because of the pressure of last year," says 16-time Grand Slam winner Federer, who lost to Djokovic in the Rome semifinals.

Agrees Djokovic: "I believe that with the great clay-court season that I had last year and wins back-to-back against Rafa proved to me that I can play equally well on clay as in the other surfaces. I need to keep that in my mind and believe in the success."

G.2.2. DOCUMENT II

**Ground-truth Record ID:**

"newsroom-00002-00064-5"

**Prefix:**

*Write about the following in a minimum of 15 paragraphs containing a minimum of 1000 words, in plain text only without titles or headings:  The text describes a sculpture of Nelson Mandela in Johannesburg, South Africa, honoring his boxing aspirations and his contributions to law, politics, and life in prison.  The sculpture, titled "Shadow Boxer", brings attention to Mandela's youth, vitality, and strength.  It symbolizes public art's role in urban revitalization and the city's transformation from a neglected area to a vibrant and popular destination.*

*Using these keywords:  Johannesburg, Mandela sculpture, Shadow Boxer, public art, regeneration, urban renewal, city transformation, crime, dilapidation, art as rejuvenation, tourism, African city, art as celebration, guide, navigation, Newtown, Walter and Albertina Sisulu sculpture, apartheid struggle*

**Generated:**

Nestled in the bustling cityscape of Johannesburg, South Africa, a striking sculpture pays homage to the extraordinary life of Nelson Mandela. Known as "Shadow Boxer," this masterpiece celebrates Mandela's aspirations as a boxer, his unwavering commitment to law and politics, and his resilience during his 27 years of imprisonment. Standing tall and proud, the sculpture serves as a poignant reminder of Mandela's indomitable spirit and his pivotal role in shaping South Africa's history.

Crafted by renowned South African sculptor Ledelle Moe, "Shadow Boxer" captures the essence of Mandela's youth, vitality, and unwavering strength. The sculpture depicts a young Mandela poised in a boxing stance, his body taut and muscles rippling with anticipation. Moe's meticulous attention to detail brings Mandela's boxing gloves to life, their weathered surfaces bearing the marks of countless hours spent honing his skills in the ring.

Yet, "Shadow Boxer" transcends the realm of sport, becoming a metaphor for Mandela's relentless determination to challenge injustice and fight for a better future for his country. The sculpture serves as a powerful symbol of his unwavering commitment to the pursuit of social justice, reflecting his belief in the power of peaceful resistance and non-violent action. Through his advocacy and leadership, Mandela left an enduring legacy that continues to inspire generations.

Beyond its artistic merit, "Shadow Boxer" plays a pivotal role in Johannesburg's urban revitalization efforts. Located in Newtown, once a neglected and crime-ridden area, the sculpture has become a catalyst for the neighborhood's transformation into a vibrant and popular destination. The influx of visitors drawn to the iconic artwork has sparked economic growth, creating new jobs and stimulating local businesses.

Furthermore, "Shadow Boxer" exemplifies the transformative power of public art in fostering social cohesion and creating a sense of place. Its prominent position in Newtown serves as a constant reminder of Mandela's life and values, encouraging citizens and visitors alike to reflect on the importance of human rights, equality, and reconciliation. The sculpture serves as a beacon of hope, inspiring individuals to embrace diversity and work together towards a more inclusive and just society.

Additionally, "Shadow Boxer" transcends its physical form, becoming an essential guide and navigation point within the Johannesburg cityscape. Its distinctive silhouette is easily recognizable from afar, serving as a landmark that helps visitors orient themselves and explore the surrounding area. This aspect of the sculpture reinforces its role as a symbol of urban regeneration and renewal,

connecting people with their city's history and cultural heritage.

Situated in close proximity to the Walter and Albertina Sisulu sculpture, "Shadow Boxer" highlights the interconnectedness of South Africa's liberation struggle. Walter Sisulu, alongside Nelson Mandela, played a pivotal role in the anti-apartheid movement, serving as a mentor and comrade. The juxtaposition of these two sculptures underscores the collective efforts of South Africa's fearless leaders who dedicated their lives to dismantling apartheid and ushering in a new era of freedom and democracy.

As the sun sets over Johannesburg, "Shadow Boxer" takes on a new dimension. Illuminated by carefully placed lights, the sculpture seems to come alive, casting long shadows that dance and flicker across the surrounding buildings. This interplay of light and shadow creates a mesmerizing spectacle, transforming "Shadow Boxer" into a vibrant beacon of hope and inspiration.

Visitors from around the world flock to Johannesburg to witness this awe-inspiring sculpture firsthand. It has become an essential stop on the itinerary of any traveler seeking to understand South Africa's complex past and its journey towards reconciliation and transformation. The sculpture is not just a tribute to Nelson Mandela; it is a testament to the resilience and optimism of the South African people who, like Mandela, refused to be defeated by adversity.

"Shadow Boxer" stands as a powerful reminder that even in the darkest of times, the human spirit can triumph over oppression and injustice. It encourages individuals to confront challenges with courage, compassion, and a relentless pursuit of justice. As Johannesburg continues to evolve and thrive, "Shadow Boxer" will undoubtedly remain a cherished landmark, serving as a source of pride for generations to come.

Indeed, "Shadow Boxer" is more than just a sculpture; it is a living testament to the extraordinary life and legacy of Nelson Mandela. Its presence in Johannesburg serves as a constant reminder of the values he held dear and the profound impact he had on South Africa and the world. Through his unwavering determination and unwavering commitment to freedom and equality, Mandela left an enduring mark on history, inspiring countless others to strive for a better future. "Shadow Boxer" stands as a timeless tribute to his indomitable spirit and an enduring symbol of hope and transformation.

## Gemini 1.0 Pro Evaluation of Quality of Generated Text:

The document is well-written and informative, providing a detailed description of the "Shadow Boxer" sculpture in Johannesburg and its significance. It highlights the artistic merit, historical context, and social impact of the artwork, making it a valuable resource for understanding its importance. The language is engaging and descriptive, effectively conveying the essence of the sculpture and its role in Johannesburg's urban revitalization efforts.

Rating: 4

## Ground-truth:

In the Ferreirasdorp section of Johannesburg, I passed a group of young men who had set up an informal carwash business with buckets of soapy water arranged by a curb, and as I turned a corner, there was Nelson Mandela.

Though Mandela died on Dec. 5, here he stood vividly rendered in layered steel cutouts, 34 years old, fresh and strong and in the pose of a boxer. He once had professional aspirations, and said the sweet science taught him about defense, attack and strategy, which he applied to law, politics and life in prison.

Unveiled two years ago, the 16-foot sculpture, called "Shadow Boxer," is a lively piece with a great sense of energy and movement. The South African artist Marco Cianfanelli gave it a three-dimensional effect by layering painted-steel sheets in an image from a well-known 1952 photograph by Robert Gosani. The work is set

in front of the Johannesburg magistrate court across the street from Chancellor House, where Mandela and his law, and later political, partner Oliver Tambo were young lawyers. Chancellor House, a privately owned corner office building, was itself restored in 2011 and has a timeline exhibition of the two partners' time there.

The attractions have drawn visitors to a neighborhood that has a certain rough-around-the-edges look but is no longer so menacing. Since Mandela's death, the sculpture has become something of a memorial with people laying flowers at its base.

The work speaks volumes: the young Mandela about to wage the fight of his life. It also shows how public art is helping to revivify urban Johannesburg, a seemingly implausible regeneration in this city of more than four million residents, which not that long ago seemed as though it was about to fall through the widening cracks of crime and dilapidation.

The art has helped foster a virtuous cycle: Color and beauty draw people; people promote security, which draws more people, and creates a bigger audience possibility for more art. The improvements have made Joburg cool again – and popular. A new market, the Sheds@1Fox, featuring South African-produced goods, is set to open in October about two blocks from Chancellor House. Johannesburg is now Africa's most visited city, with 2.5 million international visitors in 2013, according to MasterCard's annual survey.

"Art plays an incredibly important role in making people aware that the city is being reborn," said Gerald Garner, a guide and author of two books, "Johannesburg Ten Ahead" and "Joburg Places." "It's one thing to fix pavements and plant trees, and doing those things make a difference. But art makes the city humane. It tells people who were excluded by the old order that they're welcome. It tells stories of people who live here and celebrates the heroes."

I lived in Johannesburg's northern suburbs for three years in the early 1990s and have returned frequently ever since. I saw some of its worst days, when parts of its downtown grid began to feel like an urban dystopia. One of the emblematic events, what some may argue was the low point, actually involved public art, in the late 1990s when vagrants in once-beloved, then abandoned Oppenheimer Park, decapitated a sculpture of impalas and sold the heads as scrap metal.

To be clear, a good number of swaths of Johannesburg remain iffy. But the public art was a revelation for me, and after several days of exploring the city by foot and on public transit last July, I felt much more optimistic about its prospects. In the months since, the pace of positive change has even picked up.

My route was done in several walks, some on my own on the way to various appointments , and then with help from Mr. Garner and later still from a thoughtful 22-year-old artist named Jabulani Fakude whom I hired through a local company called MainStreetWalks (mainstreetwalks.co.za). It's a good idea for visitors to get guides to navigate between some areas that remain sketchy.

A short walk from "Shadow Boxer," the adjoining district, Newtown, has another popular sculpture of two other South African heroes, Walter and Albertina Sisulu, on Diagonal Street. The Sisulus, towering figures of the apartheid struggle, appear not as fighters but as elderly lovers and parents. The Johannesburg artist Marina Walsh, who installed the concrete sculpture in August 2009 after the city solicited proposals from artists, has the couple seated facing each other, their eyes locked in an affectionate gaze. The intention is to show them as equals. The Sisulus, who raised eight children, were married for 59 years, including Mr. Sisulu's 25-year term as a prisoner on Robben Island, an austere prison off the shore of Cape Town. They're exaggerated in scale; huge figures, squat and round, so as a viewer you're almost like a grandchild looking at grandparents, and indeed, you see people moved to sit in their laps – something the artist is happy to see them do.

"I get an enormously warm response from people," Ms. Walsh told me. "One Saturday I

was cleaning off a mustache someone had drawn on Albertina's lip. Some guy shouted, 'What are you doing?' He thought I was defacing her. I told him I was cleaning her and asked if he knew who they were. He said, 'Yes, they're my heroes !'"

Not every work is as accessible, and the heroes often don't have names. As you leave Newtown via the landmark Queen Elizabeth Bridge, you quickly come upon "Fire Walker," a collaboration between the renowned South African artists William Kentridge and Gerhard Marx. It's set on a traffic island, and up close, the 36-foot work, erected with steel pieces, looks like a mass of exploded fragments. But like an Impressionist painting, the image is clearer farther away – that of a woman carrying a burning caldron on her head. It's a sight you rarely see in Johannesburg now, these women with braai mealies (roast corn) that they carry with flames crackling atop their heads. The work is a tribute to the hard ways people scrabble together a living.

I wanted to visit Braamfontein, a precinct on the northern side of the city with two of South Africa's major universities, the University of the Witwatersrand and the University of Johannesburg. It has some of the earliest major public artworks, including "Juta Street Trees," nine large metal tree sculptures, and the "Eland," a concrete sculpture of Africa's largest antelope, which were installed in 2006 and 2007, respectively. The aesthetic upgrade has helped transform the student-dominated area, which even just four years ago still felt threatening, into a place with lively night life and a popular Saturday Neighbourgoods Market of artisan food, fabrics and jewelry.

Up the hill from Juta Street, in the university area, I went to the Constitutional Court of South Africa, the country's highest court and itself a work of art both for its architecture and interior design. Set on Constitution Hill with views of city streets leading out toward the hilly and affluent northern suburbs, the court's design was intended to embody the idea of traditional justice, the way elders would hear disputes before whole communities under a tree. The notion of transparency is embodied in its openness, the glass exterior. The interior pillars are erected at angles that are meant to suggest the branches of a tree, the public space where legal decisions were rendered. Wood carvings, skins and art are integrated into the interior and are also curated in exhibits.

Not all public art has official sanction, or is even legal for that matter. Wall murals have added vibrancy, too, though much of the city is still struggling to accommodate street artists who have plenty of urban wall space but face obstacles in getting permits; they consequently resort to "bombing," or illegally painting on, buildings, and for their trouble will spend occasional nights in jail or have to resort to bribing the police to go away.

Mr. Fakude, the young street artist, does his work deep into the night, and moonlights by day giving walking tours for 250 rand, about $24, at 10.40 rand to the dollar – in my case, a graffiti tour of his and others' work in the artsy Market Theater area on the western side of the city. Mr. Fakude's work included one wall piece that was playful at a glance, a one-eyed cartoon character he'd invented, but that had a serious message that addressed the stark reality of crime and violence. So, too, did others. "We want to get the message across that drugs are not cool," he said. It gave the ostensible illegality of the street artists' work a certain irony, given the grass-roots appeal to nonviolent, cleaner living. It has gotten Mr. Fakude some attention, however. He was invited to paint murals in Berlin this year, though he told me recently that he was not able to travel there.

One place where murals are getting an official stamp of approval is one of Joburg's most trendy and ambitious new places, the Maboneng Precinct, once a vast wasteland of disused industrial spaces and warehouses. In recent years, the 250-acre zone of reclaimed and repurposed industrial buildings has commissioned several dozen murals by artists from Berlin, Baltimore and elsewhere. In January, a 10-story mural was unveiled of "i am because we are." The artist Rickey Lee Gorden, who goes by Freddy Sam, used the same photo as "Shadow Boxer," the Mandela sculpture in Ferreirasdorp. The revitalization of Maboneng Precinct began

in 2009 with Arts on Main, a complex of studios and galleries on Main Street. Indeed, the first public art work in the precinct was the word "Maboneng," which is Sotho for "place of light" and was installed as text art on the Arts on Main rooftop.

Arts on Main's success lifted Maboneng's profile. Mr. Kentridge, arguably South Africa's best-known living artist, was among the first to set up a studio. Five years hence, the area is a vibrant hive of artists' lofts, mixed-use spaces, live theater, galleries, a landscaped bar-cafe and a boutique hotel called the 12 Decades, with a dozen rooms designed with artwork and artifacts to celebrate each decade in the life of the city, which started in 1886 after a gold rush.

Each building has a lighthouse design incorporated. There are 30 murals, with five more scheduled to be completed by year's end. In addition to the murals, an old canal is being restored, its banks featuring life-size steel cutouts of people who made important contributions to the precinct.

One of the last areas I explored was the mining district in Johannesburg's commercial center. The area is strewn with artifacts of mining accouterments as public art in a city built on mining. Here I found the happy postscript to the story of " Leaping Impalas," the sculpture by Herman Wald in Oppenheimer Park that had been decapitated by vagrants in the 1990s. In 2002, new heads were fashioned and welded on, and the sculpture was installed in front of the headquarters of Anglo-American, one of the world's largest mining companies. You can still see the markings on the necks, like scars that reflect the greater story of a city that is in the process of healing.

A version of this article appears in print on July 13, 2014, on page TR10 of the New York edition with the headline: A City Seen Through Artists' Eyes. Order Reprints |Today's Paper|Subscribe

G.2.3. DOCUMENT III

**Ground-truth Record ID:**

`"newsroom-00002-00064-14"`

**Prefix:**

*Write about the following in a minimum of 15 paragraphs containing a minimum of 1000 words, in plain text only without titles or headings: Hanley Ramirez's poor defense at shortstop has raised concerns about the Dodgers' postseason chances and his own free agency value. Despite his offensive contributions, Ramirez's defensive liabilities are a concern, especially considering the Dodgers' focus on starting pitching. The team has considered moving Ramirez to third base or left field, but those positions are already filled. His market could be problematic in free agency, given the increased emphasis on defense among teams. The Dodgers' outfield situation is also uncertain, with too many players and inconsistent performances. Their rotation is strong, but concerns about Josh Beckett and Dan Haren's ability to sustain their success throughout the season remain. The loss of Chad Billingsley further hurts their depth. The Dodgers are expected to be in the market for a starting pitcher to address this issue.*

*Using these keywords: Hanley Ramirez, Dodgers, defense, shortstop, free agency, extension, third base, Juan Uribe, left field, Julio Lugo, Erisbel Arruebarrena, Cuban defector, Clayton Kershaw, Matt Kemp, Joc Pederson, Yasiel Puig, Andre Ethier, Carl Crawford, Pacific Coast League, Zack Greinke, Hyun-Jin Ryu, Josh Beckett, Dan Haren, Chad Billingsley, Andrew Heaney, David Price*

**Generated:**

Hanley Ramirez's poor defense at shortstop has become a significant concern for the Los Angeles Dodgers as they approach the postseason. The team made a significant investment in Ramirez's offensive abilities when they signed him to a lucrative contract extension in 2012. However, his defensive deficiencies could potentially undermine the Dodgers' World Series aspirations.

Ramirez's struggles at shortstop have been well-documented. He has made numerous errors in the field and his range has declined significantly in recent years. This has raised questions about the Dodgers' overall defensive alignment, especially considering their focus on starting pitching.

To address Ramirez's defensive woes, the Dodgers have considered moving him to third base or left field. However, these positions are already filled by Juan Uribe and Carl Crawford, respectively. This leaves the Dodgers with few options to improve their defense in the middle of the infield.

Ramirez's defensive liabilities could also impact his value in free agency. The market for shortstops is expected to be strong this winter, with several teams in need of a solid defender at the position. However, Ramirez's defensive struggles may make it difficult for him to secure a long-term contract at a premium price.

The Dodgers' outfield situation is also uncertain. With Matt Kemp, Joc Pederson, Yasiel Puig, and Andre Ethier all vying for playing time, the team has an abundance of outfielders. However, their performances have been inconsistent, leaving the Dodgers unsure of their best lineup configuration.

On the pitching side, the Dodgers' rotation led by Clayton Kershaw is strong on paper. Kershaw is one of the best pitchers in baseball, and Zack Greinke and Hyun-Jin Ryu are both solid starters. However, there are concerns about Josh Beckett and Dan Haren's ability to sustain their success throughout the season, especially given their injury histories. The loss of Chad Billingsley to Tommy John surgery further hurts the Dodgers' depth in the rotation.

To address their rotation concerns, the Dodgers are expected to be in the market for a starting pitcher. David Price is a potential trade target, although his contract demands could make him difficult to acquire. Andrew Heaney, a top pitching prospect, could also be an option to contribute this season.

Amidst the uncertainties surrounding their defense and pitching staff, the Dodgers' path to the World Series is unclear. They have the potential to be a championship contender, but their ability to overcome their defensive and pitching issues will be crucial to their success.

**Gemini 1.0 Pro Evaluation of Quality of Generated Text:**

\*\*Good\*\*:
- The document provides a comprehensive analysis of the Los Angeles Dodgers' defensive and pitching concerns.
- It highlights the potential impact of Hanley Ramirez's poor defense on the team's World Series aspirations.
- It discusses the Dodgers' limited options to improve their defense and the challenges they face in free agency.
- It also mentions the uncertain outfield situation and the need for a starting pitcher.

\*\*Bad\*\*:
- The document lacks specific examples or statistics to support the claims made about Hanley Ramirez's defensive struggles.
- It does not provide an in-depth analysis of the Dodgers' offensive strengths or weaknesses.

Rating: 3

**Ground-truth:**

Hanley Ramirez's defense at shortstop only cost the Dodgers a perfect game Wednesday night. What might it cost the team in September and potentially October? What might it cost Ramirez in free agency after that?

Many expected the Dodgers to sign Ramirez to an extension by now. It hasn't happened, in part because the Dodgers want to see him stay healthy, in part because they might not be sure what the heck to do with him long term.

Third base could be a possibility, but the Dodgers would need to trade Juan Uribe, a popular clubhouse figure who is under contract for $6.5 million next season. Some Dodgers officials have toyed with the idea of playing Ramirez in left field, but you may have noticed that the team has too many outfielders already.

Maybe the Dodgers could win the 2014 World Series with Ramirez at short – heck, the Red Sox pulled off such a feat with Julio Lugo in '07. Advanced metrics, however, portray Ramirez as one of the worst defensive shortstops in baseball. And strong up-the-middle defense should be a requirement for a team built around starting pitching, no?

Erisbel Arruebarrena, a Cuban defector in the first year of a five-year, $25 million contract, was a major defensive upgrade in his brief stint with the club. But the Dodgers have said that they do not want to shift Ramirez between short and third, perhaps out of respect for Ramirez's wishes. Besides, Uribe could return from his strained right hamstring on Monday.

For this season, it appears, the Dodgers have little choice but to play Ramirez at short; they probably value his offense too much to trade him. But if Ramirez hits free agency, his market could turn problematic at a time when teams continue to place increased emphasis on defense. Indeed, what would it say about Ramirez if the Dodgers only were willing to make him a qualifying offer?

Let's not get too far ahead of ourselves. Ramirez and many of his teammates are on the uptick offensively. The Dodgers, winners of eight of their last 11, are only four games behind the Giants in the NL West. A big run to the postseason, a World Series title, and Ramirez's attributes again might outweigh his deficiencies.

Still, his poor throw that cost Clayton Kershaw a perfect game came on a play, as Hall of Fame broadcaster Vin Scully noted, that most shortstops make. Ramirez didn't make it. He is costing his team. He is costing himself.

Beyond Ramirez, questions persist about the Dodgers, just as they do for every club.

One rival executive said Thursday that the Dodgers' best outfield would be Matt Kemp in left, Joc Pederson in center and Yasiel Puig in right. The exec added that the Dodgers should trade one of their left-handed hitting outfielders, Andre Ethier or Carl Crawford, and keep the other in reserve.

Pederson, while batting .320 with a 1.016 OPS in the hitter-friendly Pacific Coast League, continues to strike out a ton, including 13 times in his last 28 at-bats. Crawford remains on the DL with a sprained left ankle. Ethier has only three homers and a .691 OPS. And let's not even talk about their respective contracts.

Kemp finally is getting hot, his disposition improving with his swing. Many in the industry, however, believe he ultimately will be moved – most likely in the offseason – due to his tempestuous relationship with some of his superiors.

*By clicking \"SUBSCRIBE\", you have read and agreed to the Fox Sports Privacy Policy and Terms of Use.

The Dodgers trail only the Cardinals and Athletics in rotation ERA. Kershaw, Zack Greinke and Hyun-Jin Ryu are perhaps the best 1-2-3 in the game. But does anyone seriously expect Josh Beckett and Dan Haren to hold up the entire season?

The loss of Chad Billingsley, who will undergo season-ending surgery to repair a partially torn flexor tendon in his right elbow, hurt the Dodgers' depth. The team lacks a prospect as polished as Marlins left-hander Andrew Heaney, who made his major-league debut Thursday night. The situation, in the words, of one club official, is "precarious."

So, expect the Dodgers to be in the market for a starter.

THE ORGANIZATION AS A WHOLE

The Dodgers have yet to slow down their spending, so it's natural for them to be linked to a pitcher such as the Rays' David Price, who could give them a third pitcher earning $20 million next season.

The team, however, likely would need to part with two or more of its top prospects to get Price, and its farm system isn't terribly deep to begin with (Price's teammate, super-utility man Ben Zobrist, is another potential LA target).

Club officials keep talking about developing youngsters such as Pederson and shortstop Corey Seager so they don't need to maintain a $230 million payroll. They've made progress not only in signing players from Cuba, but also Mexico, Venezuela and the Dominican Republic. Still, are they truly intent on developing a player-development machine?

The July 31 non-waiver deadline could prove the next test.

Much has been made of Price's loss of fastball velocity, from an average of 95.3 mph in 2012 to 93.5 in '13 to 92.6 this season. But can anyone seriously argue that his stuff is significantly diminished?

As pointed out by Rays Index earlier this week, Price is on pace for 280 strikeouts this season, which would be the most since Randy Johnson in 2004. Price's 12.1 strikeout-to-walk ratio, meanwhile, would break the record set by Brett Saberhagen in 1994 (11.0).

Based upon those numbers, it's impossible to say that Price is losing it, even though his 3.93 ERA is – for him – unusually high. Poor luck, however, may help explain that number.

Price's home run rate and opponents' batting average on balls in play are well above league averages, and probably will revert to his career norms as the season progresses.

Meanwhile, Price's FIP (fielding independent pitching) is within the same range it has been for the past two seasons. That statistic is considered an indicator of future performance, signaling that Price's ERA likely will drop.

The biggest issue for teams that want to acquire Price, as FanGraphs' Dave Cameron wrote Wednesday, is not his pitching. No, it's his expected $18 million to $20 million salary in his final year of arbitration in 2015.

Few clubs will want to part with high-end prospects while absorbing such a payroll hit. Then again, the numbers are relative. Price's salary next season will not be terribly above the qualifying offer for free agents, which is expected to be $15 million to $16 million.

The Phillies, despite their recent surge, surely recognize that they need to get younger. But even if the team has the will to be active sellers, it might not have a way.

Second baseman Chase Utley, after telling club officials last July that he did not want to be traded, signed a two-year extension with three club options. Now, less than a year later, he's going to reverse a course, waive his 10-and-5 rights and approve a deal?

Shortstop Jimmy Rollins, another 10-and-5 player, left open the possibility that he would approve a trade after breaking the team's all-time hit record last weekend. But realistically, where is Rollins going?

The shortstop's wife is from Philadelphia. They are the parents of two young children . And good luck finding the right fit.

Rollins' $11 million option for 2015 is almost certain to vest; he would not be just a rental. No, he would need to be comfortable spending the rest of this season in his new city, and all of next season as well.

Neither team in Rollins' native Bay Area needs a shortstop. And while Rollins might crave the spotlight in either New York or LA, he wouldn't go to the Mets, the Yankees aren't going to displace Derek Jeter and neither the Angels nor Dodgers figures to pursue a shortstop.

Then there is left-hander Cliff Lee, who – if all goes well – will return from his strained elbow before the All-Star break. By July 31, Lee still will have more than $45 million left on his contract, including a $12.5 million buyout for 2016.

The financial obligation number actually might be higher than that for many; Lee can block trades to 20 teams, and could require his $27.5 million vesting option for '16 to be guaranteed to join a club such as the Dodgers.

Phillies GM Ruben Amaro Jr. has said he would include cash in certain deals. The Phillies always could trade Lee during the August waiver period. But unless the team paid the majority of his contract, it could not expect a bounty in return.

Closer Jonathan Papelbon could get moved if the Phillies pay down his $13 million salary; he, too, has a vesting option for '16, and can only be traded to 12 teams without his approval.

Unless the Phils get truly creative, they will find it difficult to make impact moves .

I normally do not favor starting pitchers for MVP unless the field lacks strong position-player candidates; I voted for then-Red Sox outfielder Jacoby Ellsbury in 2011 over the eventual winner, Tigers right-hander Justin Verlander.

An interesting MVP case, however, can be made for Yankees right-hander Masahiro Tanaka, at least to this point of the season.

The Yankees are 12-2 when Tanaka starts, 25-31 when he does not. Even more striking, their run differential in Tanaka's starts is plus-33, while in all other games it's minus-54.

In other words, the Yankees are worse than the Rays (minus-48) when Tanaka isn't pitching and nearly as bad as the Padres (minus-62) and Diamondbacks (minus-64).

Of course, Wins Above Replacement (WAR) tells a different story. Tanaka is third among pitchers at 2.9 in the FanGraphs version of the metric, and well behind Mike Trout, the leader among position players at 4.7.

Outfielder J.D. Martinez is one of the Tigers' few recent bright spots; he has three homers during his nine-game hitting streak, and is now batting .300 with a .903 OPS in 108 plate appearances on the season.

Martinez, 26, spent the offseason watching video of Miguel Cabrera, Ryan Braun and other accomplished hitters, then completely overhauled his swing and approach. The Astros released him in spring training, and he signed a minor-league contract with the Tigers two days later.

"If you watch video of me in the past, it's the complete opposite – it's that extreme," he said. "It's kind of like I re-invented myself."

Martinez worked with his personal hitting coach for three weeks, then implemented his new ideas in Venezuela. Mostly, he's trying to line up directly to the ball and keep the barrel of his bat in the zone longer – the way Cabrera does.

# H. Full Figures

The title of each figure is of the format: "**Dataset / Scoring Model / Generating Model**". The $x$-axis is the temperature and hues correspond to different mixing ratio of human-generated and LLM-generated texts. From left to right, the proportion of LLM-generated texts is: (0%, 25%, 50%, 75%, 100%). The purpose of including these is to examine how sensitive fractal parameters are when LLM-generated texts are mixed with natural language. We observe that all fractal parameter indeed vary smoothly.

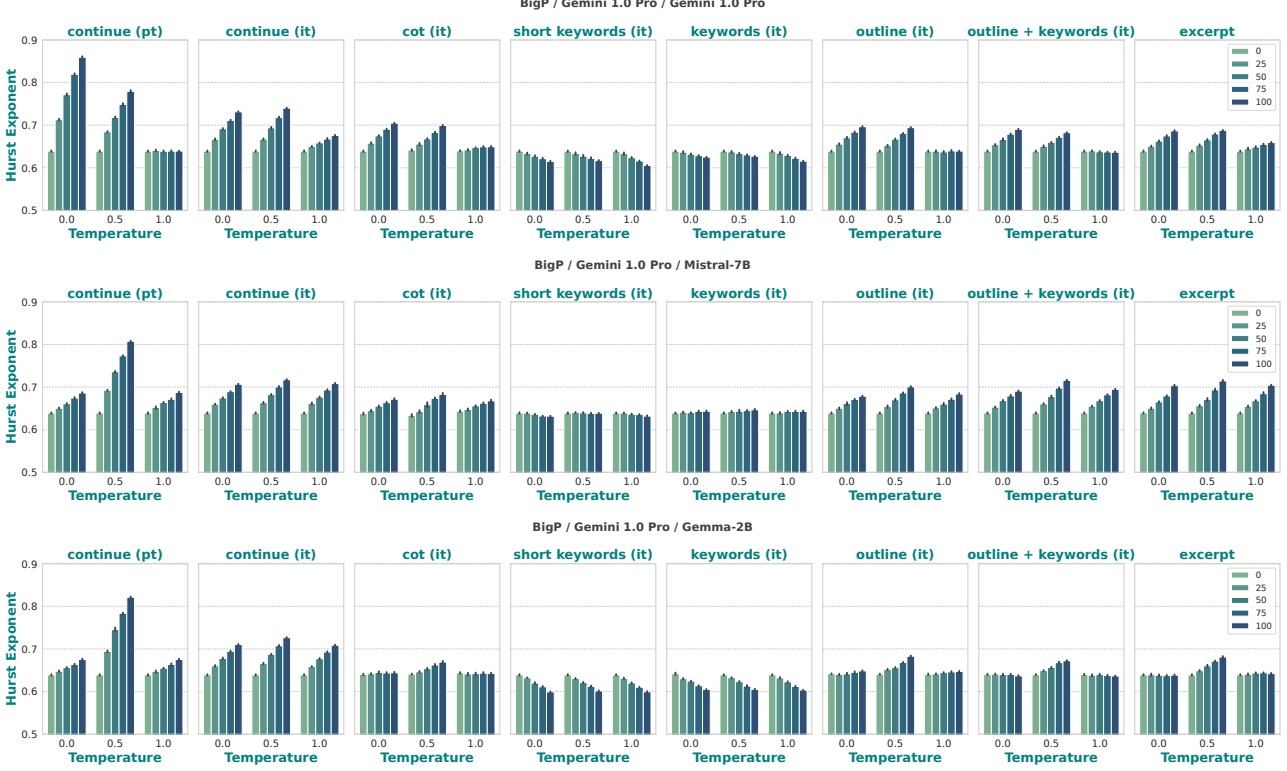

# A Tale of Two Structures: Do LLMs Capture the Fractal Complexity of Language?

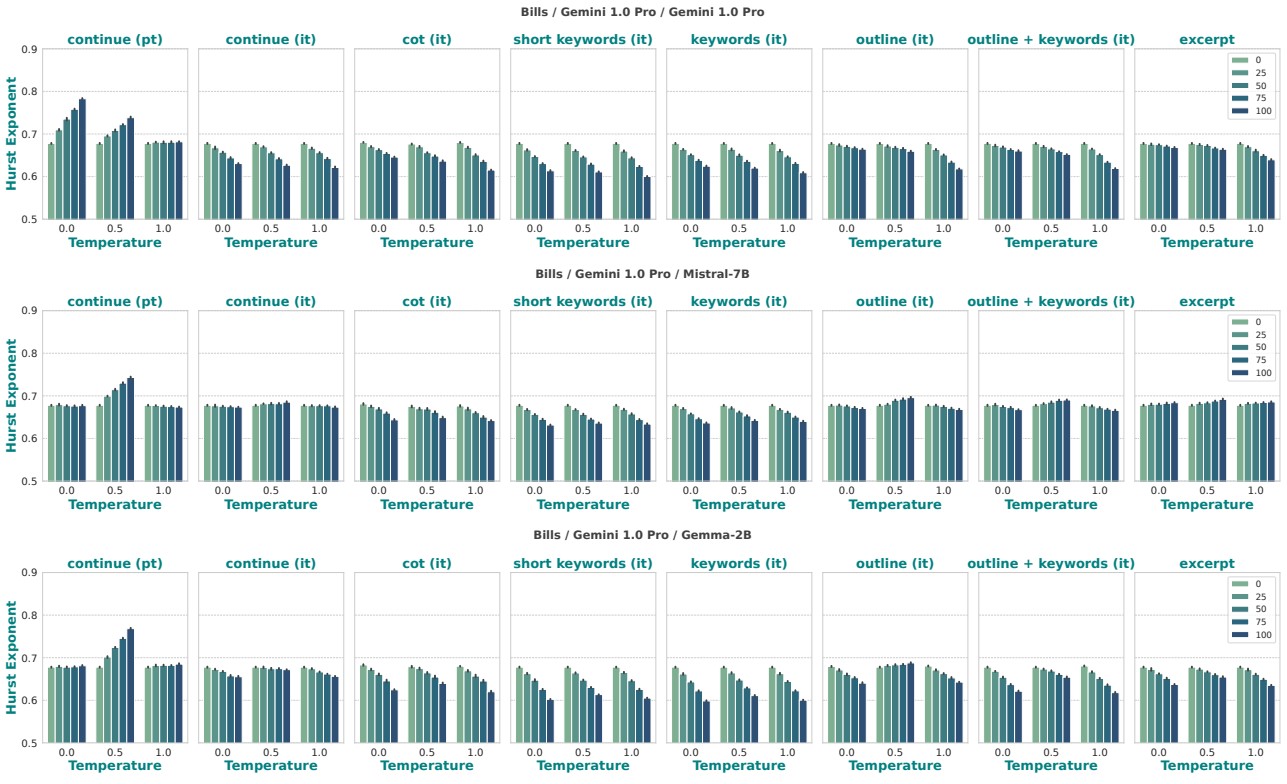

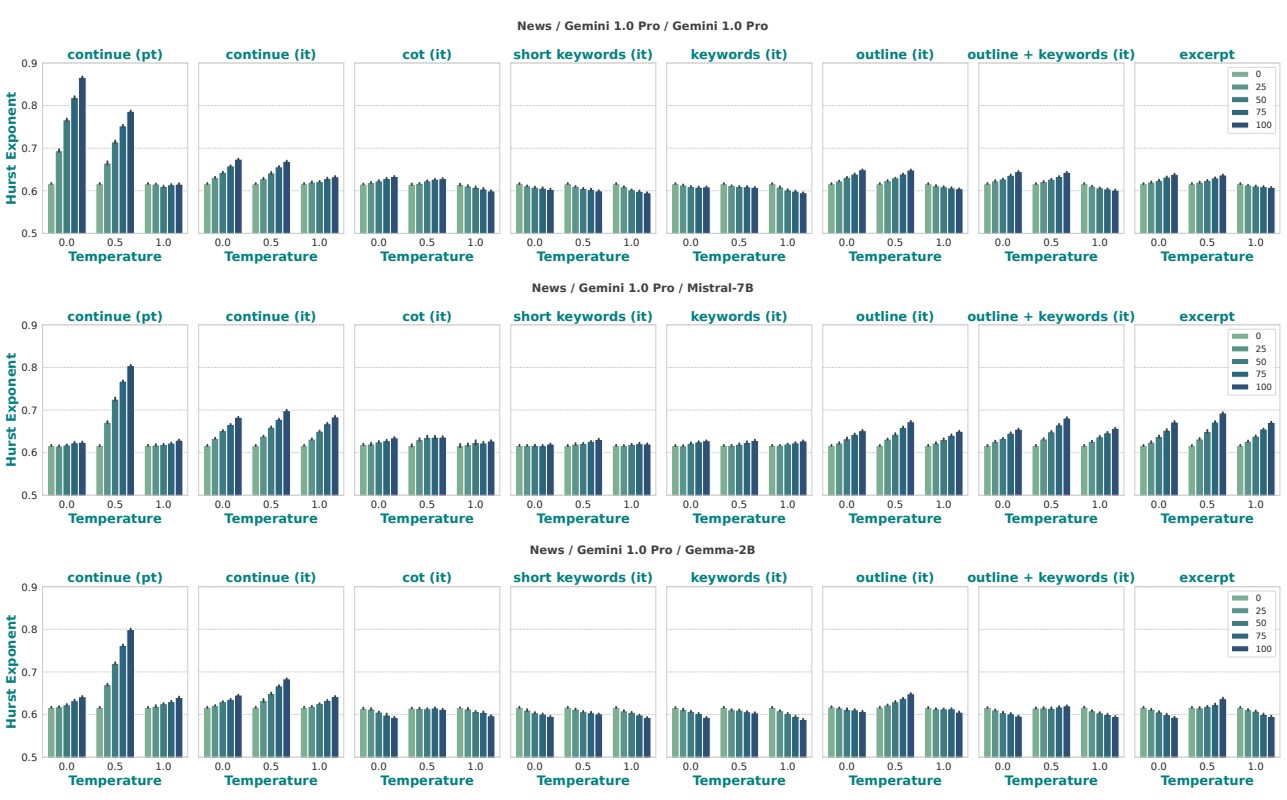

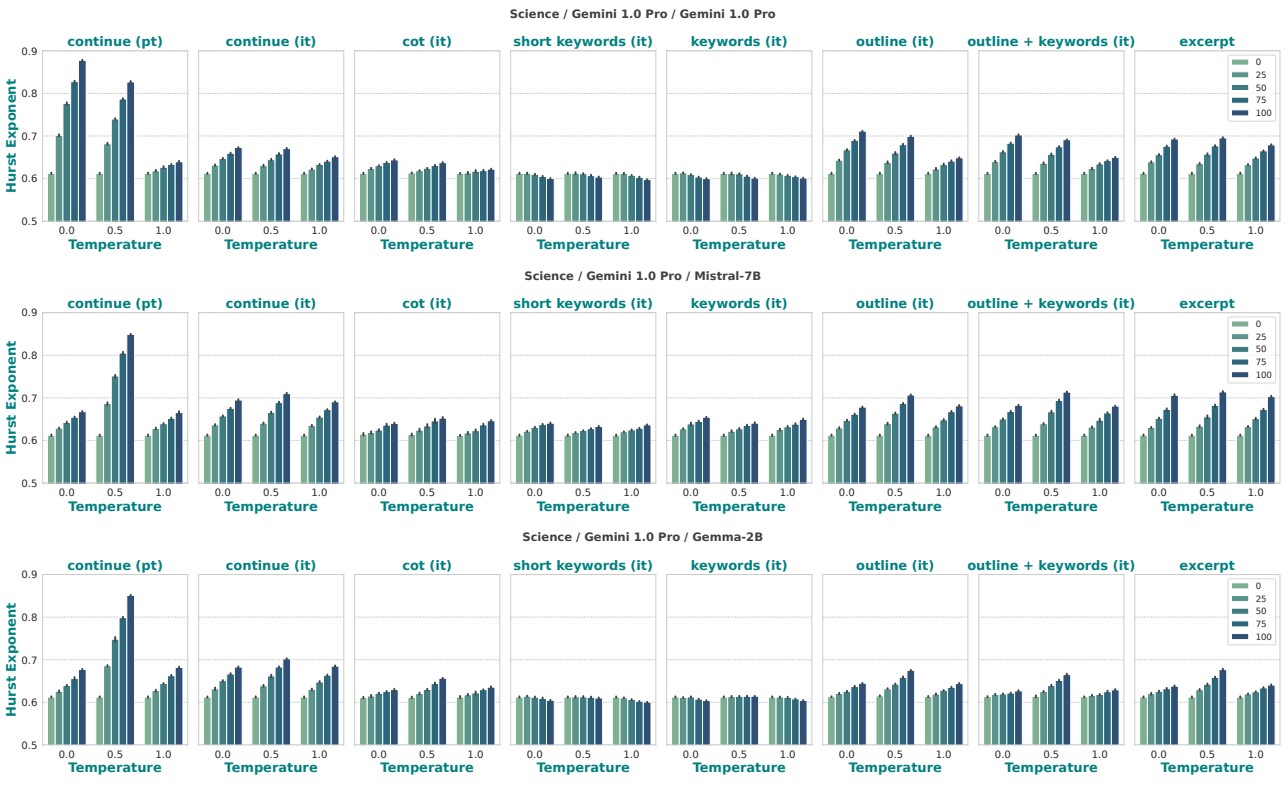

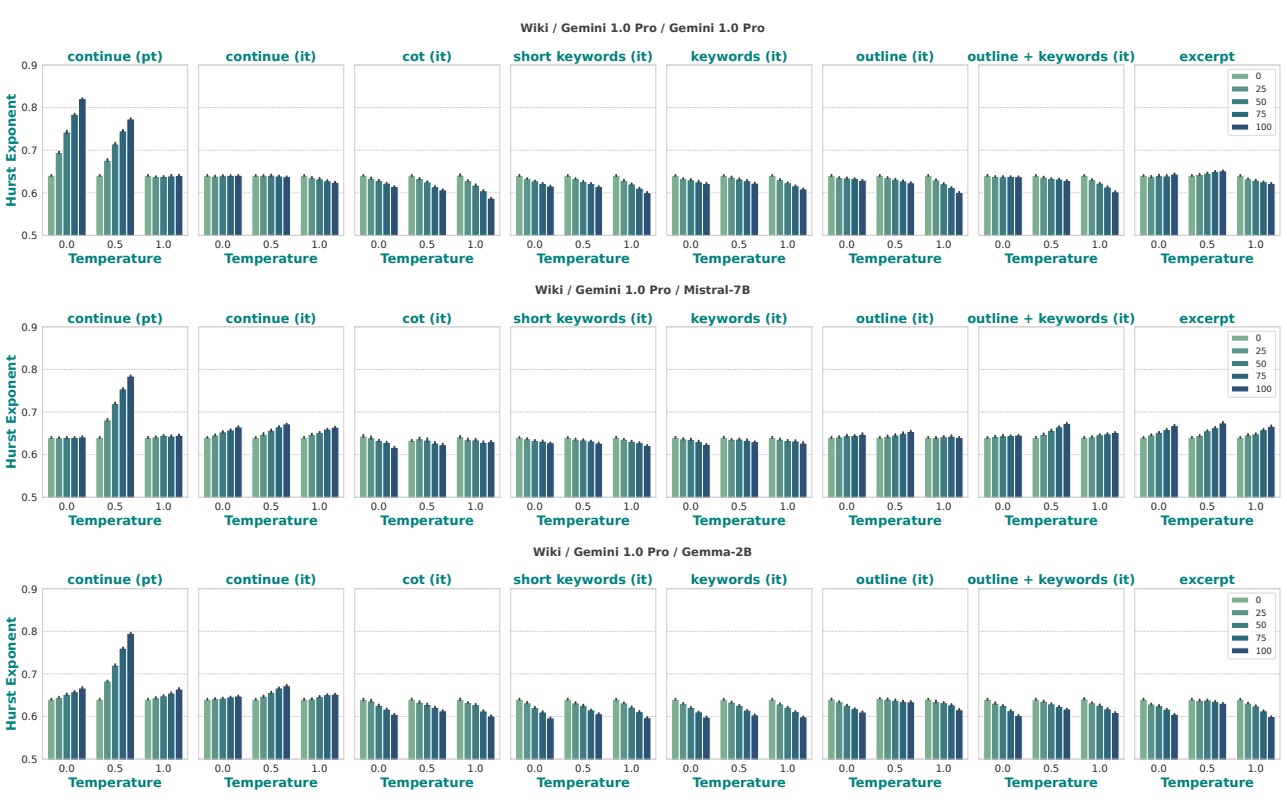

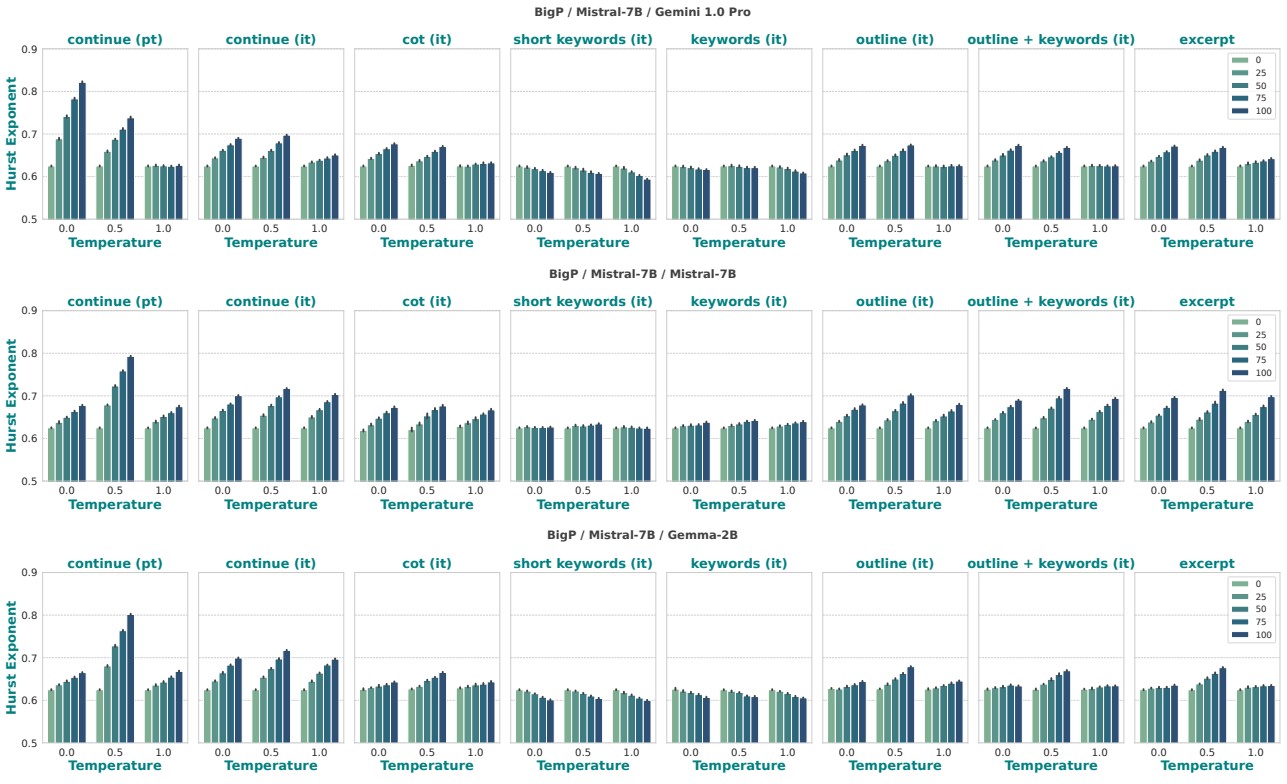

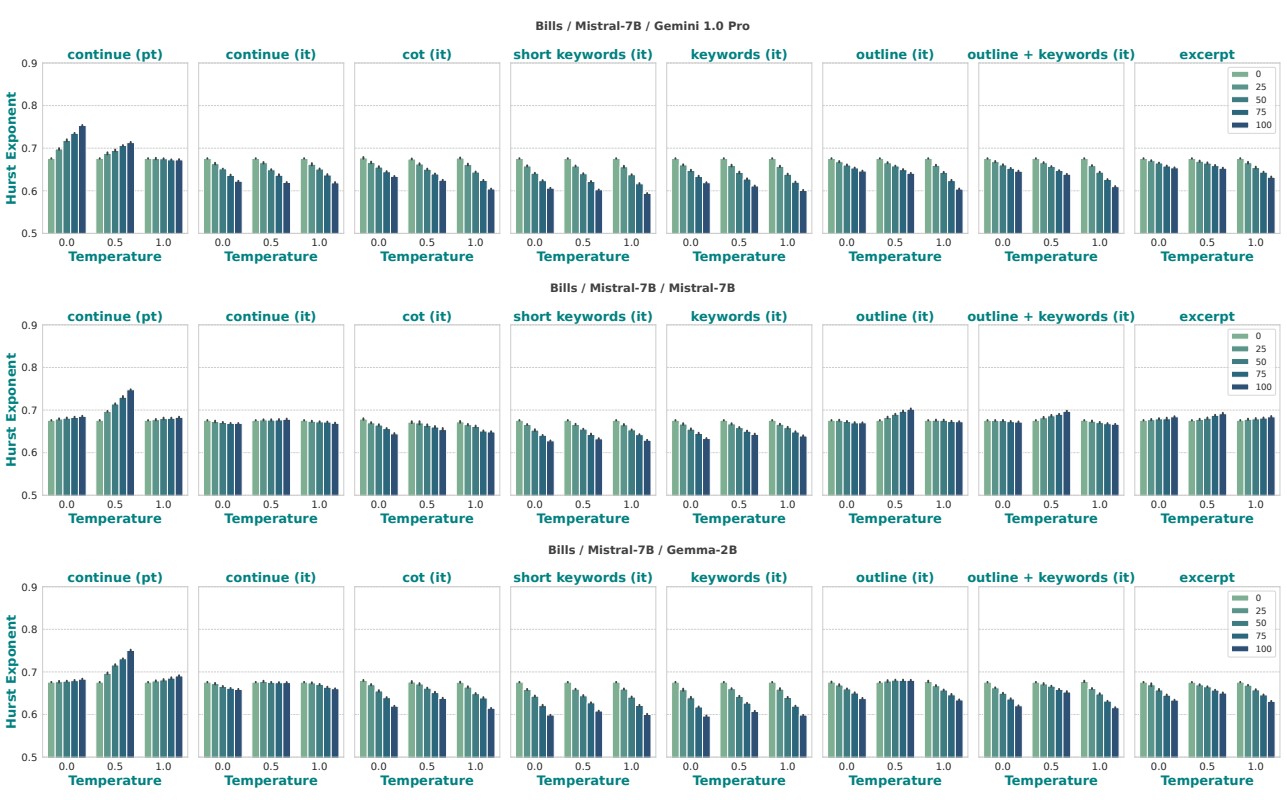

# A Tale of Two Structures: Do LLMs Capture the Fractal Complexity of Language?

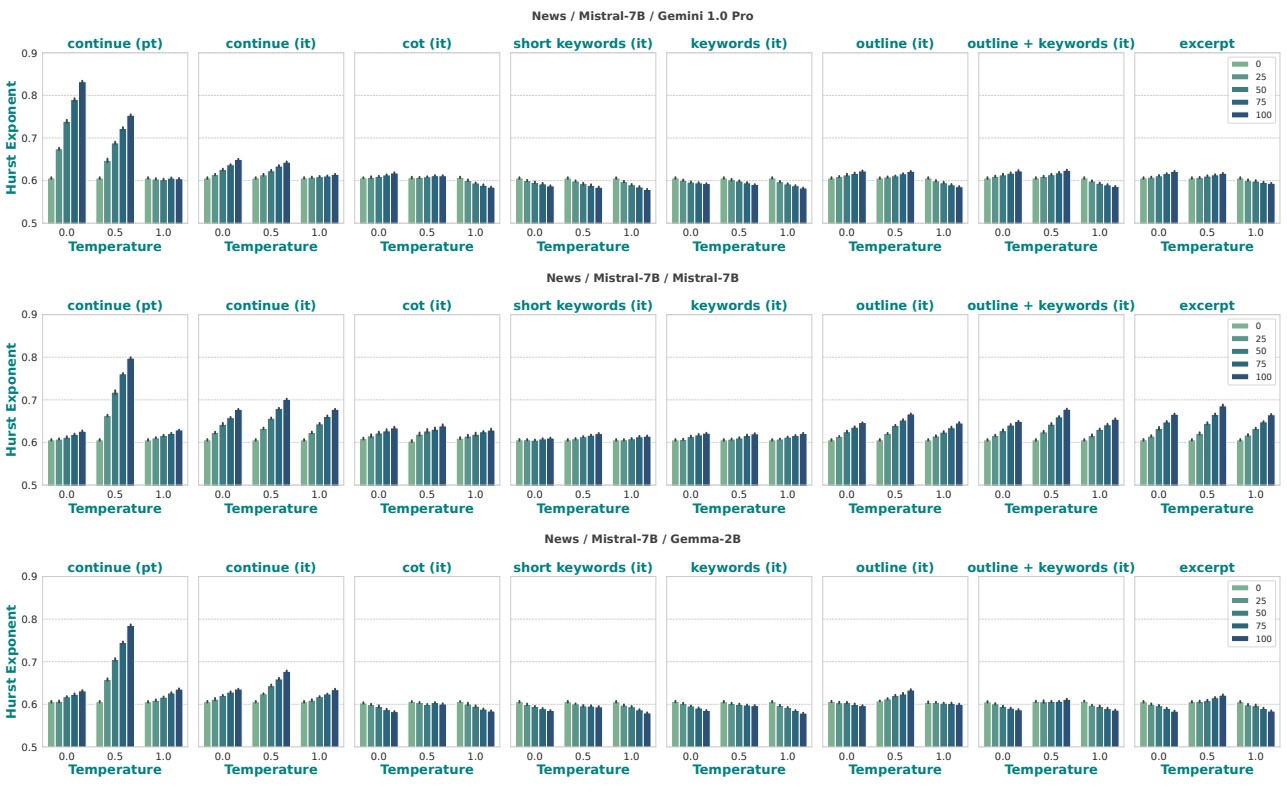

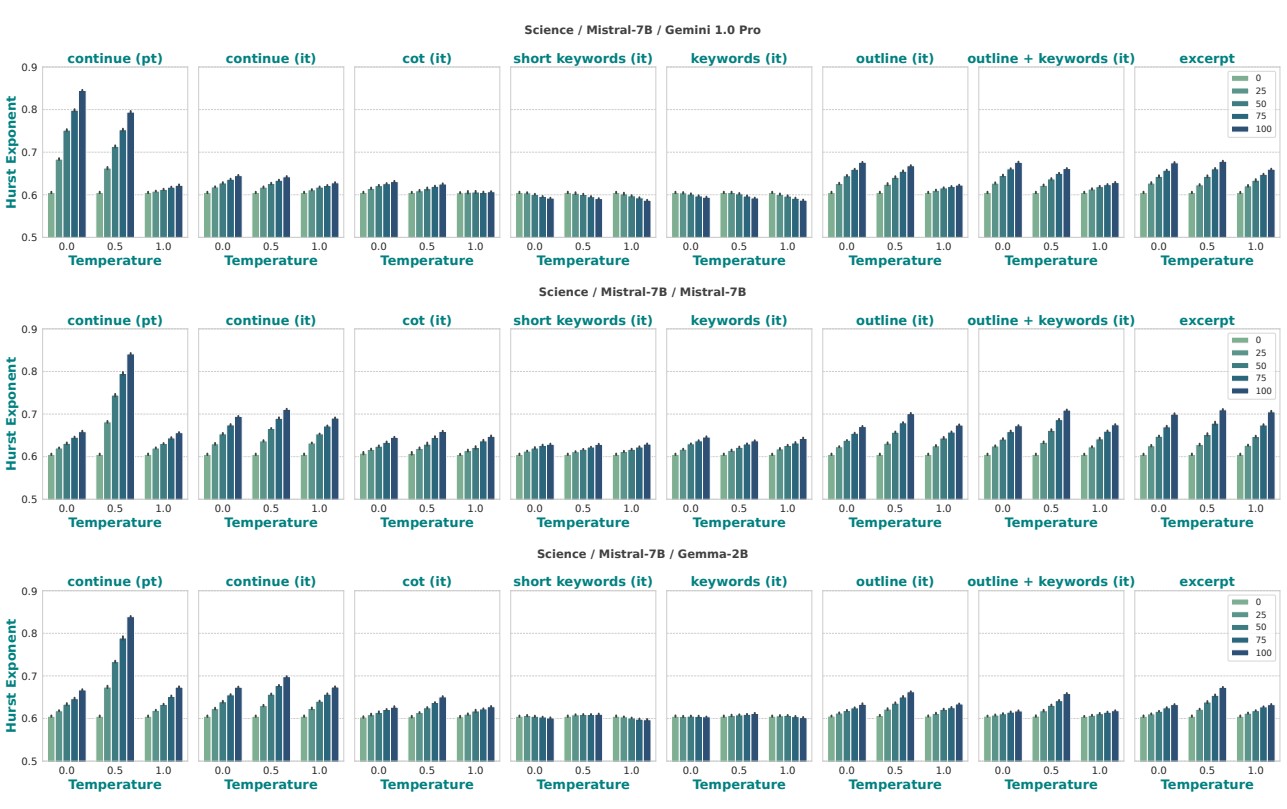

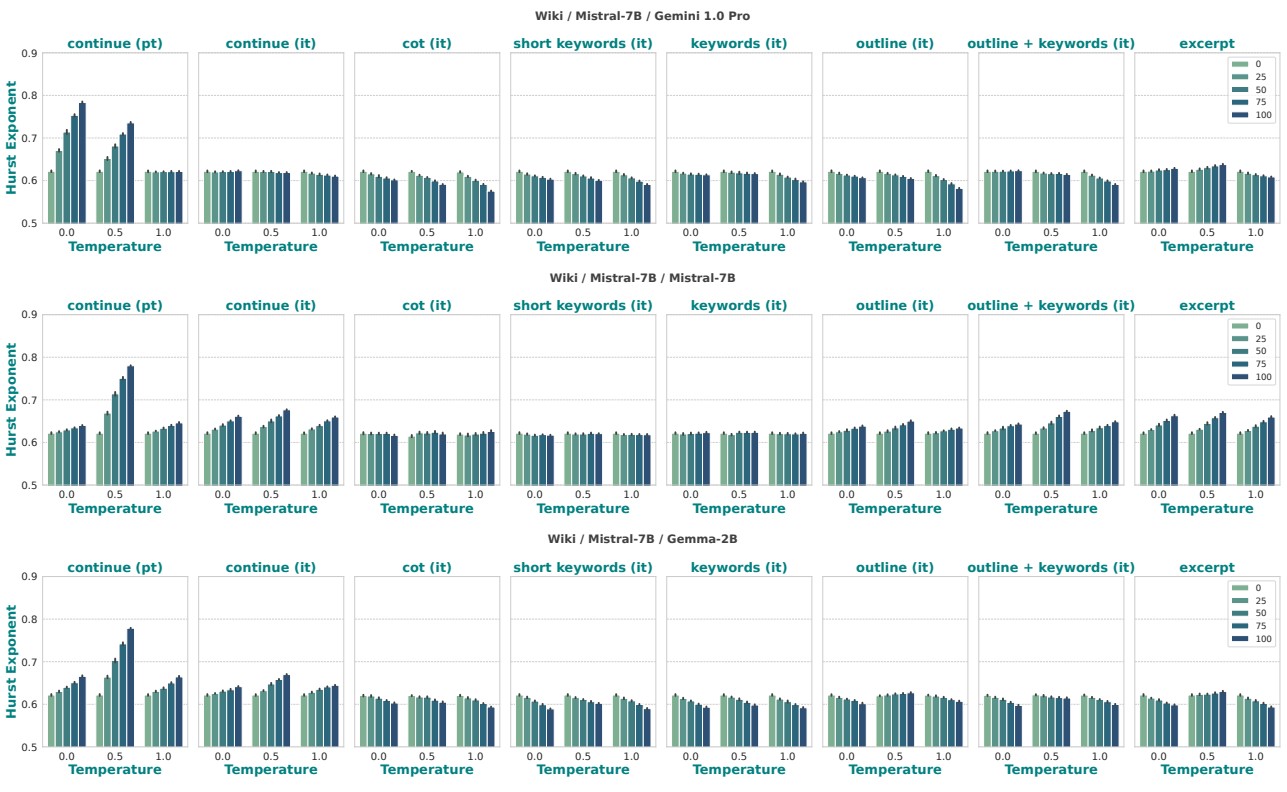

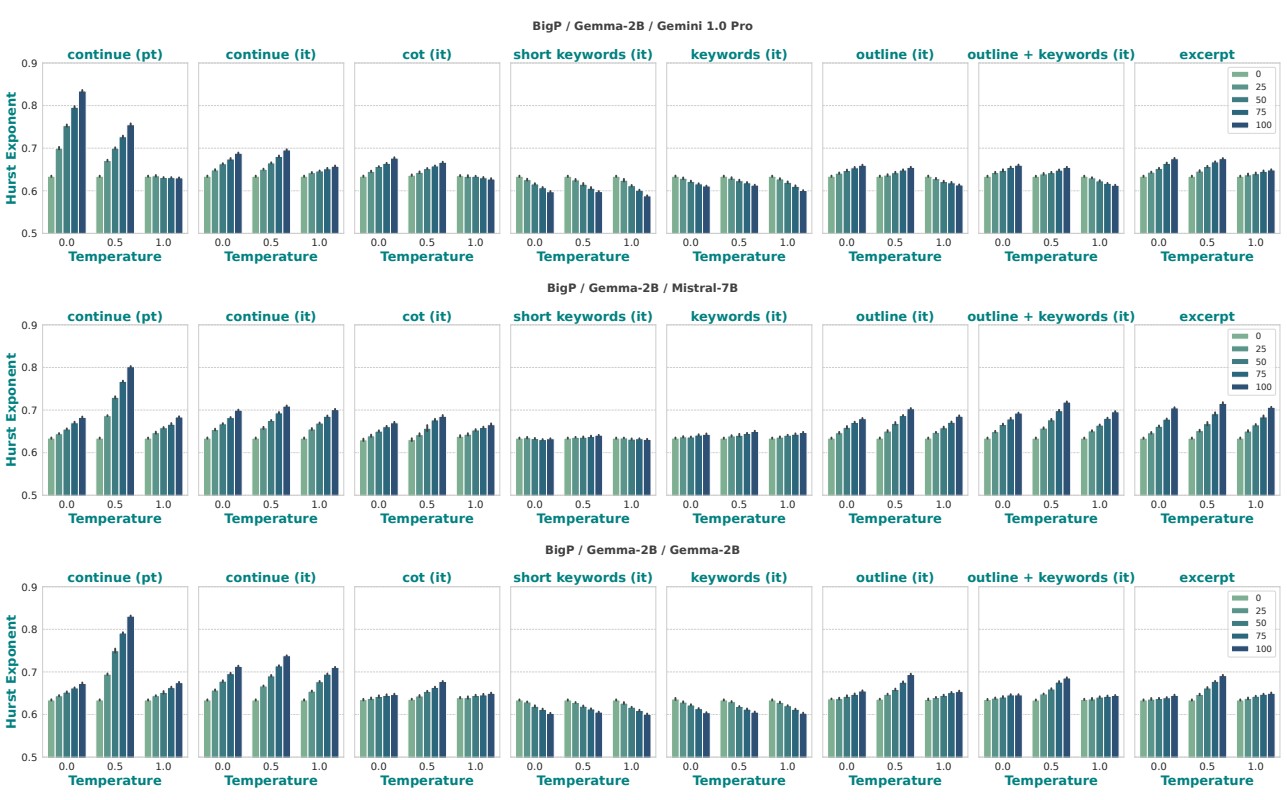

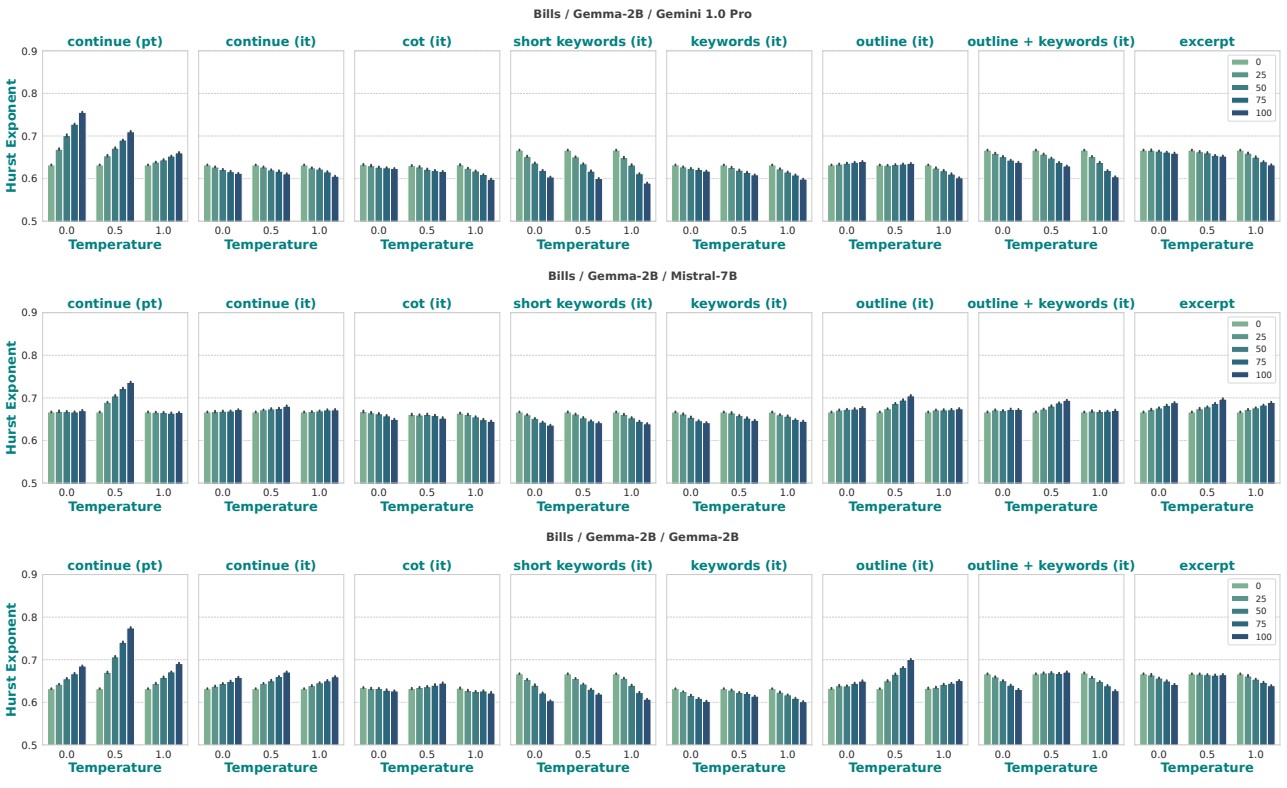

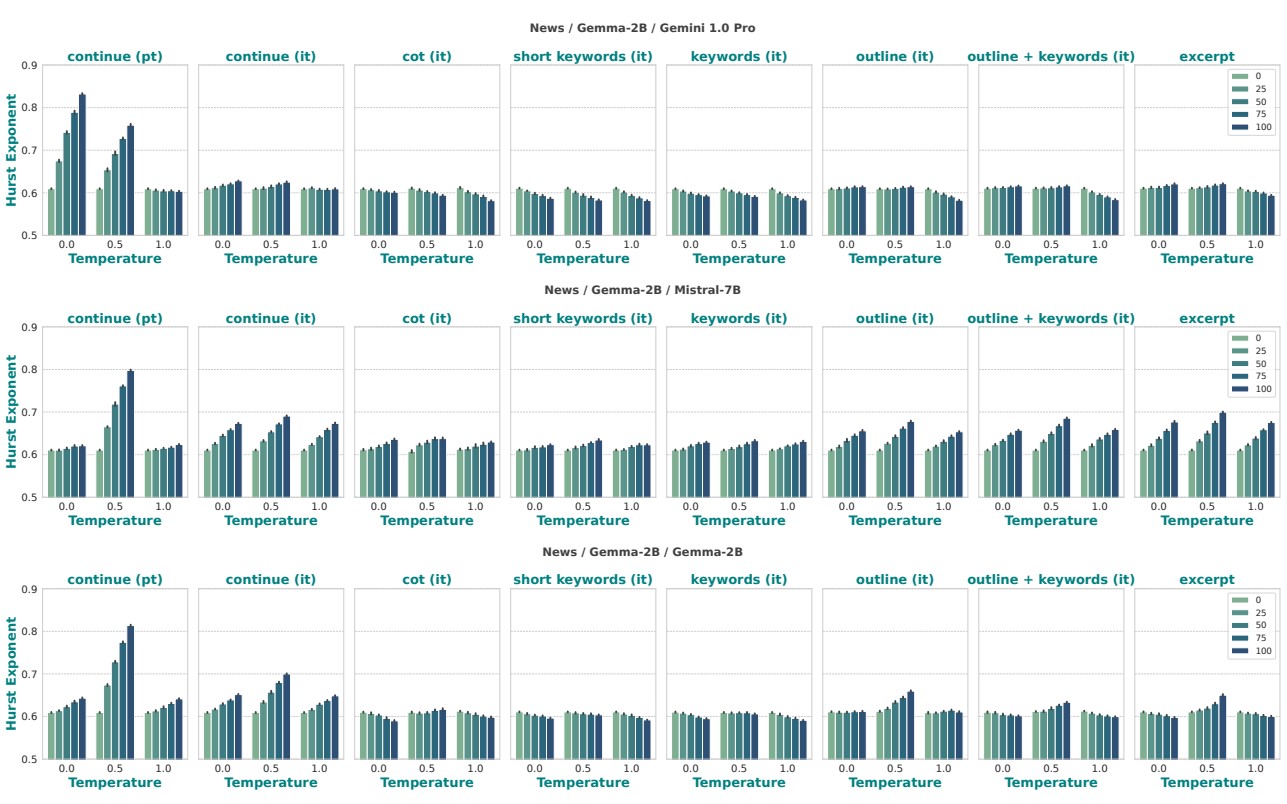

# A Tale of Two Structures: Do LLMs Capture the Fractal Complexity of Language?

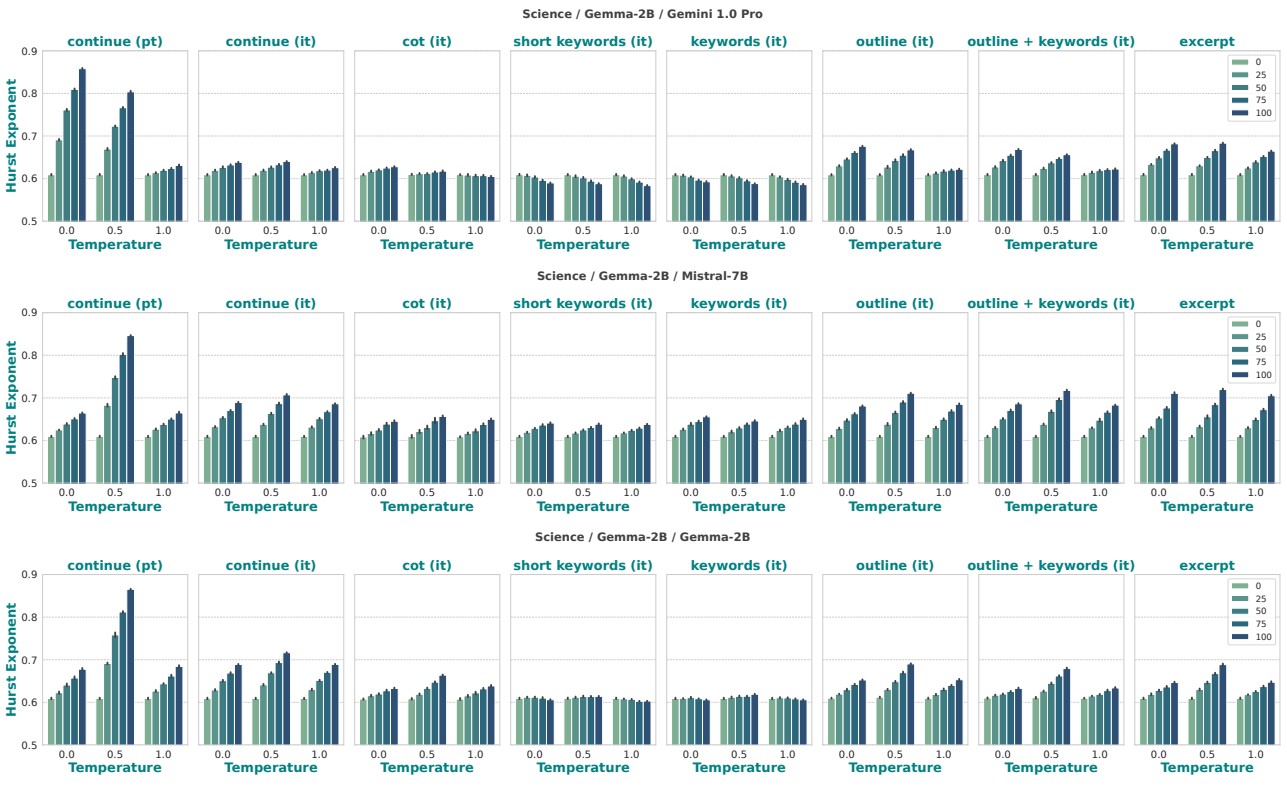

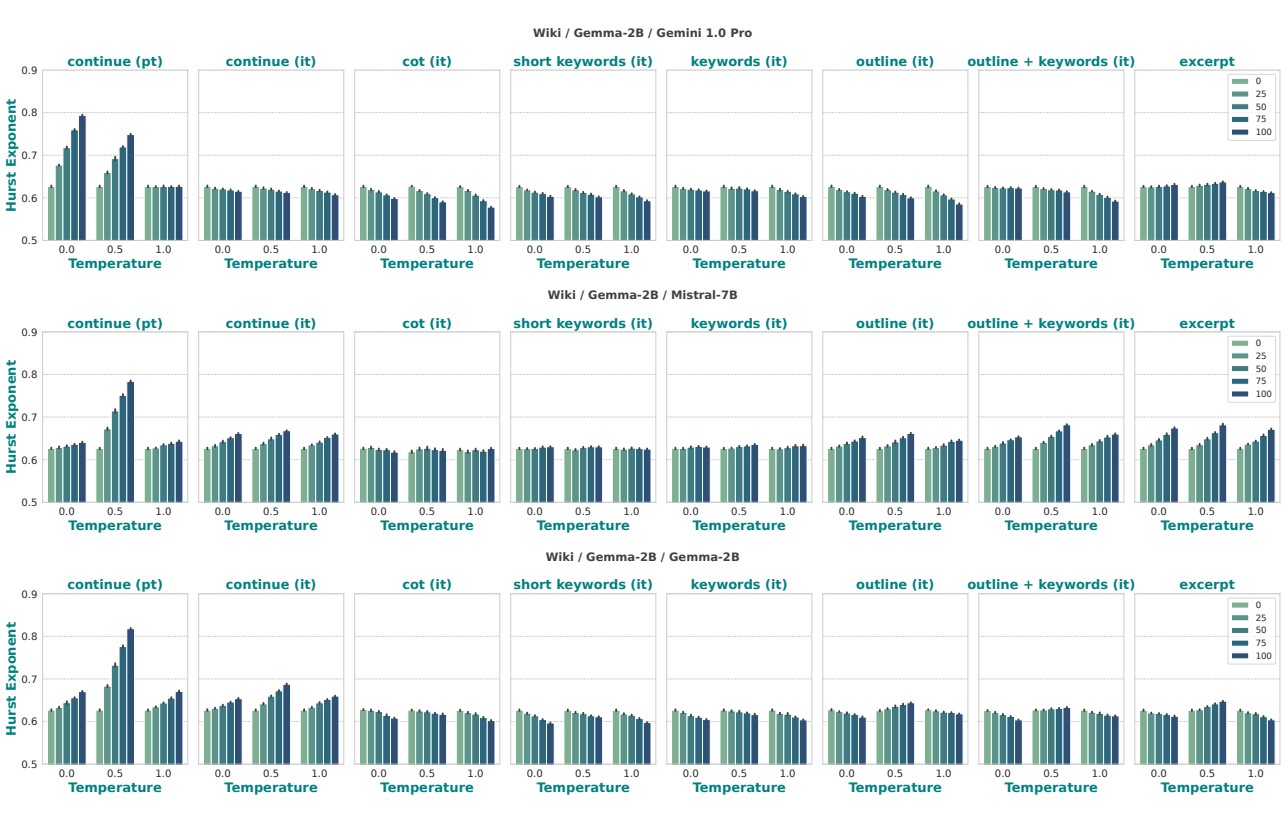

