# OpenReview forum: "A Tale of Two Structures: Do LLMs Capture the Fractal Complexity of Language?"
_ICML.cc/2025/Conference — ICML 2025 poster_

### Official Review · Reviewer_Utsk · 2025-02-21

**Overall Recommendation:** 1

**Summary:**

This article examines whether LLMs exhibit long memory under different conditions. It reports that temperature settings and prompting methods may disrupt long memory.

**Claims And Evidence:**

The authors claim that temperature settings and prompting methods could destroy long memory, and this finding is robust to the choice of architecture, including Gemini, Mistral, and Gemma.

The architectures tested here are very similar, making it unclear to what extent this finding applies more broadly. Additionally, the paper is not well-structured and contains many undefined concepts.

**Essential References Not Discussed:**

Evaluating Computational Language Models with Scaling Properties of Natural Language
Shuntaro Takahashi, Kumiko Tanaka-Ishii, Computational Linguistics, Volume 45, Issue 3 - September 2019

**Experimental Designs Or Analyses:**

The empirical design involves testing Gemini, Mistral, and Gemma using different prompting methods and settings.

**Methods And Evaluation Criteria:**

The method involves examining two important long-memory properties, typically studied as Hurst/Hölder exponents. The authors conducted diverse experiments to determine whether long memory holds.

However, since the experiments were conducted only with Gemini models, the scientific significance of their findings remains unclear.

**Other Comments Or Suggestions:**

This large report may contain valuable findings, but it is the authors' responsibility to demonstrate their significance.

**Other Strengths And Weaknesses:**

In this work, several concepts appear undefined. For example, in the Introduction, the paper states, 'We refer the reader to (Alabulmohsi 2024) for the exact definitions of these quantities.' However, the paper should be self-contained, and it is not the reader's responsibility to seek out definitions elsewhere.

Additionally, different LLM settings, including 'beta' and others, are used as common terms, but these settings must be clearly defined and consistently applied within the paper.


The paper contains mistakes. For example, Willinger et al. (1995) is cited in the statement: 'In language, such self-similarity is attributed to its recursive structure.' However, the title of Willinger et al.'s paper is Self-Similarity in High-Speed Packet Traffic, which has no connection to language.

**Questions For Authors:**

What would the findings be if you conducted your test with Mamba?

**Relation To Broader Scientific Literature:**

There is previous work beyond what the authors cited that has examined long memory in LLMs and its differences from natural language.

**Theoretical Claims:**

This paper is empirical only and there are no theoretical claims.

---

> ### Author Rebuttal · Authors · 2025-03-28
>
> Dear Reviewer,
>
> Thank you for taking the time to review our paper and sharing your concerns. While we wish to clarify certain aspects, we have taken your points seriously and conducted additional experiments to address them. We believe these new results significantly strengthen the paper, demonstrating the robustness and broader applicability of our conclusions. We are pleased to report that our core findings remain consistent across this broader range of experiments.
>
> Please find our detailed responses below:
>
> **More Models**
>
> We appreciate your concern regarding the diversity of models. We want to clarify that our initial experiments were conducted consistently across three models: Gemini 1.0 Pro, Mistral-7B, and Gemma-2B, as detailed throughout the results sections (e.g., Figures 3, 4, 6, 7, etc.). We are not relying solely on Gemini.
>
> However, we agree that demonstrating robustness across a wider range of architectures is beneficial. So, we have now conducted additional experiments using the RAID dataset (https://arxiv.org/pdf/2405.07940), which contains texts generated by 11 other models (e.g. GPT, LLAMA, Cohere, … ) in many domains. We will add these new results to the supplementary material of the revised version of the paper. We have found that our conclusions continue to hold.
>
> For example, as before, only the Hurst exponent (H) is well-correlated with text quality [[Link to Figure]](https://postimg.cc/wt9WXvg0). This observation holds across the 11 models and 7 domains in RAID, reinforcing our earlier result. In addition, natural language still has a tighter distribution of fractal parameters compared to LLM-generated text, particularly for S with low decoding temperature [[Link to Figure]](https://postimg.cc/y3mr7652).
>
> These new experiments provide strong evidence that our conclusions regarding the fractal properties of LLM text are not limited to the initial models but generalize more broadly across the current LLM landscape.
>
> Please refer to our response to Reviewer `YDw714` for a detailed overview of the new experiments as well as the full list of new figures.
>
> **Presentation**
>
> We regret that the paper was perceived as not well-structured. We organized the study around 9 precise research questions (Section 3) to provide a systematic analysis, and all questions were thoroughly answered with extensive experiments. We would appreciate it if you could help us understand what is lacking so we can improve it?
>
> **Missing Reference**
>
> Thank you for highlighting this work. This is indeed relevant, as it studies statistical properties (e.g. Zipf’s law) in language models predating 2019. We will incorporate it into the related works section.
>
> **Self-Containment**
>
> Thank you for this suggestion to improve self-containment. We will add a dedicated section to the supplementary materials providing a clear, self-contained explanation of the Hölder (S) and Hurst (H) exponents as well as the decoding temperature $\beta$.
>
> **Incorrect Citation**
>
> Thank you for catching this typo. We have fixed it.
>
> **Question Regarding Mamba**
>
> This is an excellent question. Exploring whether our findings hold for fundamentally different architectures like State Space Models (SSMs) is a valuable direction for future research. However, analyzing Mamba was beyond the scope of the current study, since our focus was on auto-regressive models. We will mention SSMs as an avenue for future research in the revised paper.
>
> **Clarifying Our Contribution**
>
> Our main contribution is that fractal analysis offers a novel and insightful lens for understanding the capabilities and limitations of LLMs in replicating the complex statistical structures of natural language. As we show in the paper, various strategies, like the decoding temperature and prompting method, can impact fractal parameters even when log-perplexity scores seem to be unaffected.
>
> In addition, this work contributes to “DNN Science” by treating LLMs as phenomena to be studied rigorously, highlighting important questions, and conducting comprehensive experiments to answer them thoroughly. We systematically investigate how controllable variables affect LLMs’ ability to mimic human text structure. This approach not only offers a complementary evaluation methodology (comparing generated text's fractal dimensions to natural language) but also deepens our scientific understanding of how these models function and where they still fall short of human linguistic complexity—a core area of interest for our community.
>
> **Summary**
>
> Thank you again for your valuable feedback. We believe the additional experiments, clarifications, and revisions significantly strengthen the paper and directly address the concerns raised, and we hope that they resolve them. If you have any remaining concerns, please let us know so we can respond to them during the rebuttal period. Otherwise, we would appreciate it if you consider revising your score.

---

### Official Review · Reviewer_sSWy · 2025-02-26

**Overall Recommendation:** 3

**Summary:**

The paper examines whether large language models (LLMs) replicate the fractal characteristics of natural language. Using a dataset of 240,000 LLM-generated articles, the authors analyze fractal parameters (Hölder and Hurst exponents) across three models (Gemini 1.0 Pro, Mistral-7B, Gemma-2B), decoding temperatures, and prompting strategies. Key findings: (1) LLMs exhibit wider fractal parameter variation than natural language, with larger models performing better; (2) temperature and instruction tuning impact self-similarity and long-range dependence; (3) more informative prompts do not always improve fractal alignment, revealing a double descent effect; (4) fractal parameters correlate with text quality and detection potential. The authors claim to release the GAGLE dataset to aid further research.

**Claims And Evidence:**

The paper provides strong empirical support for most claims. However, the claim that results hold across a variety of model architectures is weaker, as only three models (Gemini 1.0 Pro, Mistral-7B, Gemma-2B) are tested, with two from the same ecosystem. Expanding to more diverse architectures (e.g., LLaMA, GPT-4, Claude) would improve generalisability.

**Essential References Not Discussed:**

I believe that, in some sense, the fractal properties of text were investigated in [4] for artificial text detection. While it takes a different approach using fractal dimension, it would be really interesting to understand how it corresponds to your method.


[4] Tulchinskii, Eduard, et al. "Intrinsic dimension estimation for robust detection of ai-generated texts." Advances in Neural Information Processing Systems 36 (2023): 39257-39276.

**Experimental Designs Or Analyses:**

The study is well-structured, using Hölder and Hurst exponents as key statistical metrics and analyzing a large dataset (240,000 articles) across multiple models, temperatures, and prompting strategies. The dataset spans various domains, ensuring diversity in text sources. However, a potential limitation is the limited model diversity—testing only three architectures (Gemini 1.0 Pro, Mistral-7B, and Gemma-2B) may not fully generalise findings across different LLM families.

**Methods And Evaluation Criteria:**

The proposed methods and evaluation criteria are well-aligned with the problem.  The large-scale dataset (240,000 articles) covering diverse domains and generation settings strengthens the empirical foundation. However, while the chosen models and prompting strategies offer useful insights, a broader range of architectures, as I believe, would strengthen the paper.

**Other Comments Or Suggestions:**

I believe, it would be better if authors included the background about fractal characteristics or at least  explicitly state how the computations are done. Now one has to look up to Alabdulmohsin et al. (2024) to understand it.

Some small drawbacks:
- Missing Figure reference in appendix B
- Fig 3, Fig 4, and maybe somewhere else: typo GEMMMA
- Please specify in all figures’ captions what G-P, M-7 and G-2 means.
- Fig 5 is unreadable

**Other Strengths And Weaknesses:**

Strengths:
- **(S1)** Contribution to Existing Research: the paper continues previous work on fractal properties in language (Alabdulmohsin et al., 2024) and expands it by evaluating LLM-generated text across different models, decoding temperatures, and prompting strategies. This provides many valuable insights and findings.
- **(S2)** Dataset: As the authors claim, they will release the GAGLE dataset (240,000 articles), which also explores an interesting aspect – the variation in contextual information provided during prompting. This dimension has not been included in other datasets related to artificial text detection.
- **(S3)** Clarity: The paper is well-structured, with a clear explanation of experimental methodology and key findings. The figures effectively illustrate trends in fractal parameters.

Weaknesses:
- **(W1)** Limited Model Diversity: The study tests only three models (Gemini 1.0 Pro, Mistral-7B, Gemma-2B), which may not be sufficient to generalize conclusions across different LLM architectures. Expanding the analysis to models like LLaMA, GPT-4, or Claude would strengthen the findings.
- **(W2)** Unclear Differentiation Between AI and Human Text: While fractal parameters reveal structural differences between LLM-generated and human text, the study does not demonstrate that these differences enable reliable classification, as authors also mention in Limitations.
- **(W3)** Theoretical Justification: While the empirical results are strong, the paper would benefit from more discussion on why fractal parameters should generalize across different LLMs, rather than just showing observed correlations.

**Questions For Authors:**

- **(Q1)**: How well do fractal parameters generalize across different LLM architectures? If authors could include other models in the dataset it would strengthen the statements. If it is not possible, author could probably use other open-source datasets like RAID [5] or COLING Workshop on MGT Detection [6].
- **(Q2)** Can fractal parameters reliably differentiate AI-generated from human text?
- **(Q3)** I wanted to clarify if a causal model in Figure 1 is something defined in literature? If yes, could you please provide some references. If no, I believe it should be stated in the paper explicitly.


[5] Dugan, Liam, et al. "Raid: A shared benchmark for robust evaluation of machine-generated text detectors." (2024).

[6] Wang, Yuxia, et al. "GenAI content detection task 1: English and multilingual machine-generated text detection: AI vs. human." (2025).

**Relation To Broader Scientific Literature:**

The paper builds on research in statistical properties of natural language and LLM-generated text, expanding beyond log-perplexity-based evaluations. It aligns with findings from [1] on fractal patterns in text. It connects to prior works on LLM-generated text detection, such as Mireshghallah et al. and Gehrmann et al., by proposing fractal parameters as a distinguishing feature [2, 3].


[1] Alabdulmohsin, Ibrahim, Vinh Q. Tran, and Mostafa Dehghani. "Fractal Patterns May Illuminate the Success of Next-Token Prediction." (2024).

[2] Mireshghallah, Niloofar, et al. "Smaller language models are better zero-shot machine-generated text detectors." Proceedings of the 18th Conference of the European Chapter of the Association for Computational Linguistics (Volume 2: Short Papers). 2024.

[3] Gehrmann, Sebastian, Hendrik, Strobelt, Alexander, Rush. "GLTR: Statistical Detection and Visualization of Generated Text." Proceedings of the 57th Annual Meeting of the Association for Computational Linguistics: System Demonstrations. Association for Computational Linguistics, 2019.

**Theoretical Claims:**

The paper primarily focuses on empirical analysis rather than formal theoretical proofs. No explicit mathematical proofs were checked, but the statistical methodology appears sound.

---

> ### Author Rebuttal · Authors · 2025-03-28
>
> Dear Reviewer,
>
> We thank you for your detailed review and constructive feedback. We are pleased that you have found our experiments thorough, claims well-supported, the overall study well-structured, and the findings valuable and insightful. As you stated, GAGLE includes various prompting strategies, unlike other public datasets.
>
> Your primary concerns regarding the diversity of the models were particularly helpful. We have taken these points seriously and conducted substantial additional experiments specifically to address them. We believe these new results significantly strengthen the paper, demonstrating the robustness and broader applicability of our conclusions. We are pleased to report that our core findings remain consistent across this broader range of experiments.
>
> Please find our detailed responses below:
>
> **More Models**
>
> We have followed your advice and used the RAID dataset, which contains texts generated by 11 models (e.g. GPT, LLAMA, Cohere ...) in many domains, addressing the specific gap you noted. We will add these new results to the supplementary material of the revised version of the paper. We have found that our conclusions continue to hold. For example, as before, only the Hurst exponent (H) is well-correlated with text quality [[Link to Figure]](https://postimg.cc/wt9WXvg0). This observation holds across the 11 models and 7 domains in RAID, reinforcing our earlier result.
>
> Please refer to our response to Reviewer `YDw714` for a detailed overview of the new experiments as well as the full list of new figures.
>
> **Missing Reference**
>
> Thank you for bringing up this missing reference. This is quite relevant, since they propose using the intrinsic dimension of the data manifold as a metric that distinguishes natural language from AI-generated texts, similar to how we show separation using fractal parameters.
>
> Tulchinskii et al.'s scope, however, is limited to detecting synthetic texts, whereas we show that fractal analysis offers a novel and insightful lens for understanding the capabilities and limitations of LLMs in replicating the complex statistical structures of language. As we show in the paper, various controllable variables, like the decoding temperature and prompting method, can impact fractal parameters even when average log-perplexity score seems to be unaffected. We believe this approach can deepen our scientific understanding of how these models function and where they still fall short of human linguistic complexity—a core area of interest for our community.
>
> We will add this reference to the related work section with a brief discussion in the revised paper.
>
> **Relevance to Detection**
>
> We acknowledge your observation regarding detecting LLM-generated texts. As we mention in the paper, we do not focus on detecting synthetic texts in this work. However, we do believe that fractal parameters might prove useful for detection and we leave this to future research. Our results demonstrate that fractal structures are often more difficult for LLMs to replicate accurately than simpler statistical properties captured by perplexity (e.g. Figures 4/6/7/10 and Table 3). We plan to explore this direction more in the future.
>
> **Background About Fractals**
>
> Thank you for this suggestion to improve self-containment. We will add a dedicated section to the supplementary materials providing a clear, self-contained explanation of the Self-Similarity (S) and Hurst (H) exponents and how they are computed.
>
> **Causal Model**
>
> Regarding the causal model in Figure 1, this is a conceptual model we introduce to hypothesize why variations in prompt information density might influence the fractal structure of generated text, even if models are calibrated at the next-token level. It serves to motivate our investigation into prompting methods. Appendix B provides a concrete example illustrating this potential effect. We will clarify this in the revised paper.
>
> **Typos**
>
> We greatly appreciate you catching these details! We will correct the typos, fix the missing figure reference in Appendix B, improve the readability of Figure 5, and ensure all captions clearly define abbreviations.
>
> **Summary**
>
> Thank you again for your valuable feedback. We believe the additional experiments, clarifications, and revisions significantly strengthen the paper and directly address the concerns raised, and we hope that they resolve them. If you have any remaining concerns, please let us know so we can respond to them during the rebuttal period. Otherwise, we would appreciate it if you consider revising your score.

---

> > ### Comment · Reviewer_sSWy · 2025-04-05
> >
> > Thank you for your thoughtful and detailed responses! The clarifications and revisions, as I believe, would improve the paper, and I have raised my score accordingly.

---

### Official Review · Reviewer_EJTc · 2025-03-13

**Overall Recommendation:** 2

**Summary:**

This paper investigates whether LLMs can replicate the fractal complexity found in natural language. The authors use the Hölder exponent (S) to examine self-similarity and Hurst exponent (H) for long range dependence. Authors have explored a large range of models, sampling temperature and prompts etc. Through thorough evaluation, they found that

1. Larger models are better at replicate the fractal structures of natural language than small models.

2. High decoding temperature improve similarity to natural text.

3. Prompting strategies impact text fractality non-monotonically.

**Claims And Evidence:**

For the 9 points of analysis, the claims are supported by accompanying figure. I also like that authors experiment with different scoring models to show that results are consistent across different scoring models.

**Essential References Not Discussed:**

N/A

**Experimental Designs Or Analyses:**

The experimental design mainly concerns about (1) how to score those two metrics (2) how to generate synthetic data while ensuring there is a natural text baseline to compare against. The experimental design on data synthesis is valid.

**Methods And Evaluation Criteria:**

The authors uses two main metric (H and S) to evaluate how model generated data might have different fractal structures compared to natural text. Overall, those two metrics are established measures by prior work. Furthermore, authors examine how different factors impact those two metrics - the factors examined by authors are relevant factors that might impact a model's generation. Therefore, the methods overall make sense.

**Other Comments Or Suggestions:**

There are a lot of analysis organized by 9 questions, and I enjoyed reading about these analysis. However, I think because there are so many points, after reading the paper I do not know what the main message of the paper is and what the main contribution is in helping detecting LLM generated text. If authors think that this work contributes to a different line of research that I am missing, please let me know!

**Other Strengths And Weaknesses:**

Strength:

1. The paper introduces fractal parameters (Hölder exponent S and Hurst exponent H) as a new way to analyze self-similarity and long-range dependence in natural and LLM-generated text. The approach goes beyond surface-level comparisons, as mostly done by previous work.

Weakness:

1. If I read this paper as an evaluation paper, I am not too sure if the evaluation results made on three models, which differ by architecture, pretraining data and training paradigm. To this extent, while it is interesting to see how these models differ, whether the conclusion can generalize remain unclear.

2. If I read this paper as trying to understand whether those two metrics could be interesting for the detection community, there is no strong evidence that the two proposed metrics can be used reliably as a detection metric. Furthermore, there are no experiments in the paper examining whether those metrics can be used for detection.

**Questions For Authors:**

See comments.

**Relation To Broader Scientific Literature:**

This paper is closest to the line of works on detecting LLM generated texts. However, the authors emphasizes that the primary goal is not to achieve detection. Based on the experiment results, it does not seem likely that the metrics used in this analysis could be a reliable way to detect LLM generated texts. While the analysis is interesting to see how different axis can change the characteristics of generated text, I it might only have limited implications to this line of work.

**Theoretical Claims:**

N/A

---

> ### Author Rebuttal · Authors · 2025-03-27
>
> Dear Reviewer,
>
> We thank you for your detailed review and constructive feedback. We are pleased that you have found our work comprehensive, the evaluation valid and thorough, the use of fractals novel, and the paper well-organized and easy to read.
>
> Your primary concerns regarding the generalizability of our findings and the clarity of the main contribution were particularly helpful. We have taken these points seriously and conducted substantial additional experiments specifically to address them. We believe these new results significantly strengthen the paper, demonstrating the robustness and broader applicability of our conclusions. We are pleased to report that our core findings remain consistent across this broader range of experiments.
>
> Please find our detailed responses below:
>
> **Generalizability of findings**
>
> Thank you for highlighting this point. We have now conducted additional experiments using the RAID dataset (https://arxiv.org/pdf/2405.07940), which contains texts generated by 11 models (e.g. GPT2/3/4, LLAMA, Cohere, MPT, and Mistral) in many domains. We will add these new results to the supplementary material of the revised version of the paper. We have found that our conclusions continue to hold.
>
> For example, as before, only the Hurst exponent (H) is well-correlated with text quality [[Link to Figure]](https://postimg.cc/wt9WXvg0). This observation holds across the 11 models and 7 domains in RAID, reinforcing our earlier result. In addition, natural language still has a tighter distribution of fractal parameters compared to LLM-generated text, particularly for S with low decoding temperature [[Link to Figure]](https://postimg.cc/y3mr7652).
>
> These new experiments provide strong evidence that our conclusions regarding the fractal properties of LLM text are not limited to the initial models but generalize more broadly across the current LLM landscape.
>
> Please refer to our response to Reviewer `YDw714` for a detailed overview of the new experiments as well as the full list of new figures.
>
> **Main Message**
>
> We appreciate the opportunity to clarify the main message and contribution of our work.
>
> Our central message is that fractal analysis offers a novel and insightful lens for understanding the capabilities and limitations of LLMs in replicating the complex statistical structures of natural language. As we show in the paper, various strategies, like the decoding temperature and prompting method, can impact fractal parameters even when log-perplexity scores seem to be unaffected. This goal is in line with earlier works, such as (Meister & Cotterell, 2021), who argued that the evaluation of LLMs should go beyond log-perplexity and also consider how well LLMs capture other “statistical tendencies” observed in natural language. Our key contribution lies in introducing and validating this fractal analysis framework.
>
> In addition, this work contributes to “DNN Science” by treating LLMs as phenomena to be studied rigorously, highlighting important questions, and conducting comprehensive experiments to answer them thoroughly. We systematically investigate how controllable variables affect LLMs’ ability to mimic human text structure. This approach not only offers a complementary evaluation methodology (comparing generated text's fractal dimensions to natural language) but also deepens our scientific understanding of how these models function and where they still fall short of human linguistic complexity—a core area of interest for our community.
>
> We hope this answers your concern. We will clarify this message in the revised version of the paper.
>
> **Relevance to Detection**
>
> We acknowledge your observation regarding detecting LLM-generated texts. As we mention in the paper, we do not focus on detecting synthetic texts in this work. However, we do believe that fractal parameters might prove useful for detection and we leave this to future research. Our results demonstrate that fractal structures are often more difficult for LLMs to replicate accurately (e.g. Figures 4/6/7/10 and Table 3). We plan to explore this direction more in the future.
>
> **Summary**
>
> Thank you again for your valuable feedback. We believe the additional experiments, clarifications, and revisions significantly strengthen the paper and directly address the concerns raised, and we hope that they resolve them. If you have any remaining concerns, please let us know so we can respond to them during the rebuttal period. Otherwise, we would appreciate it if you consider revising your score.

---

### Official Review · Reviewer_YDw7 · 2025-03-14

**Overall Recommendation:** 3

**Summary:**

This study constructs a dataset named GAGLE, comprising 240,000 AI and human-generated language instances. It employs fractal parameters, including Self-Similarity and Long-Range Dependence (LRD), to examine the differences between various language model sizes and architectures compared to human texts.The research aims to offer a new evaluation perspective for language models by analyzing their fractal characteristics, thereby extending beyond traditional perplexity metrics. By investigating the statistical patterns in model-generated texts, this work enhances our understanding of current language models' capabilities and limitations in replicating natural language complexity and structure. This analysis is crucial for advancing more accurate and natural text generation technologies.

**Claims And Evidence:**

- Some claims in the article are supported by clear and convincing evidence and are consistent with previous research. For example, Q1 mentions that the perplexity of LLM-generated text is lower, and Q2 indicates that generated text with higher temperature is closer to human-like output. However, some claims lack convincing evidence. For instance, in Q6, the study on "how fractal parameters relate to the quality of output" does not clearly specify how the quality of the article is measured, making the claim less convincing. Q8 lacks experimental data to support its argument. Additionally, Q9 overlooks the fact that the dataset types chosen for the study are consistent, failing to address the review/reddit-type data, which represents a notable gap in the research.
- The current dataset (GAGLE), sourced from Wikipedia, BigPatent, Newsroom, and BillSum, leans towards academic texts and may not fully capture the differences between LLM-generated and human text in informal contexts, such as social media.

**Essential References Not Discussed:**

No.

**Experimental Designs Or Analyses:**

**Rationale**:
- The calculation of fractal parameters (such as the Hurst exponent) provides a quantitative analysis of the statistical properties of LLM-generated text, which is innovative. The findings of Q1, Q2, Q3 Q4 are relatively consistent with the previous findings

**Issues**:
- Q6, the study on "how fractal parameters relate to the quality of output" does not clearly specify how the quality of the article is measured, making the claim less convincing. Q8 lacks experimental data to support its argument. Additionally, Q9 overlooks the fact that the dataset types chosen for the study are consistent, failing to address the review/reddit-type data, which represents a notable gap in the research.
- The figures and tables are disorganized: the order of tables and figures is not coherent (e.g., Figure 4/5 do not define IT and PT, and there is a discontinuous reference between Figure 3 and Figure 4). As a result, terms like IT (Instruction Tuning) and PT (Pre-Training) are not clearly defined, which affects readability.

**Methods And Evaluation Criteria:**

The current dataset (GAGLE), sourced from Wikipedia, BigPatent, Newsroom, and BillSum, leans towards academic texts and may not fully capture the differences between LLM-generated and human text in informal contexts, such as social media.  Additionally, the evaluation criteria are somewhat limited.  The impact of fractal analysis could be further explored by examining the accuracy of downstream tasks.

**Other Comments Or Suggestions:**

- Carefully review the relationship between tables and their respective positions in the article to ensure readability.
- Increase the diversity of the data used.
- Carefully check the connection between the proposed arguments and the experiments, as some points (e.g., Q8) lack experimental support.
- Incorporate a wider variety of models into the analysis.

**Other Strengths And Weaknesses:**

Overall, the contributions of this article are claimed as follows:
1.  Analyzing the factors that contribute to LLMs replicating natural language fractal characteristics.
2.  Exploring the impact of prompts on the fractal structure of text.
3.  The results are applicable to various model architectures.
4.  Releasing a dataset containing 240,000 articles.

However, the contributions in points 3 and 4 are limited.  For point 3, the article only uses the Gemini 1.0 Pro, Mistral-7B, and Gemma-2B models, without considering classic and more-widely used models such as GPT, Llama, or the latest o1 and DeepSeek-R1 model.  As for point 4, the dataset is not comprehensive, mainly including academic texts and neglecting the importance of data in informal contexts.

**Questions For Authors:**

- How does the fractal relationship between LLMs and humans evolve with the development of LLMs (e.g., from ChatGPT to DeepSeek-R1)?
- How does the fractal relationship between LLMs and humans differ across different types of datasets?
- Previous studies have shown that substantial portions of text from the training data of LLMs can be extracted using carefully designed prompting techniques. It would be valuable to explore whether incorporating novel human data can help determine if the model's retention of human data leads to a fractal relationship that more closely mirrors human characteristics.

**Relation To Broader Scientific Literature:**

No.

**Theoretical Claims:**

Not applicable.

---

> ### Author Rebuttal · Authors · 2025-03-27
>
> Thank you for your detailed and constructive feedback. We are pleased that you have found our work innovative and valuable.
>
> We have carefully considered your concerns and have conducted new experiments to address them, particularly regarding the diversity of models and datasets. We believe these additions substantially strengthen the paper's contributions and generalizability. We are pleased to report that our core findings remain consistent across this broader range of experiments.
>
> Please find our responses below:
>
> **1. Limited Scope of Models and Datasets**
>
> We have now conducted additional experiments using the RAID dataset (https://arxiv.org/pdf/2405.07940), which contains texts generated by 11 models (e.g. GPT, Llama, ...) in domains that include Reddit and reviews, addressing the gap you noted. We'll add these results to the supplementary material.
>
> **Summary of Findings:**
> *(Please note that Q2/4/5/8 are not applicable here because we don't control the prompts in RAID and we score using Gemini Pro 1.0)*
>
> * **Q1 (Log-perplexity)**: [[link to Figure]](https://postimg.cc/qznYYzVd) Consistent with our results, greedy decoding and instruction tuning yield lower perplexity than human text, but pretrained models at $\beta=1$ show perplexity similar to human text.
>
> * **Q3 (Fractals in IT Models)**: [[link to Figure]](https://postimg.cc/r0Svw9Rh) Our findings still hold: Instruction tuning affects the Hurst exponent (H), especially at low temperatures (leading to higher H), while Self-Similarity (S) remains largely unaffected.
>
> * **Q6 (Text Quality)**: [[link to Figure]](https://postimg.cc/wt9WXvg0) As before, only the Hurst exponent (H) is well-correlated with quality. This observation now holds across the 11 models and 7 domains in RAID, reinforcing our earlier result.
>
> * **Q7 (Distribution of Fractals)**: [[link to Figure]](https://postimg.cc/y3mr7652) Natural language still has a tighter distribution of fractal parameters compared to LLM-generated text, particularly for S with low decoding temperature.
>
> * **Q9 (Data analysis)**: We've repeated the analysis of Table 3 in RAID. The results are below. Interestingly, it seems challenging for LLMs to replicate humans in *poetry*, and this only becomes evident when we look into the Self-Similarity exponent.
>
> |Dataset|S log-ratio|H log-ratio|PPL log-ratio|
> |---|---|---|---|
> |abstracts|$-0.13\pm0.05$|$0.16\pm0.02$|$-0.59\pm0.09$|
> |books|$-0.10\pm0.04$|$0.07\pm0.01$|$-0.66\pm0.04$|
> |news|$0.18\pm0.02$|$0.11\pm0.01$|$-0.53\pm0.04$|
> |poetry|$\bf0.50\pm0.02$|$0.05\pm0.01$| $-0.67\pm0.10$|
> |recipes|$0.10\pm0.05$|$0.05\pm0.01$|$-0.75\pm0.04$|
> |reddit|$-0.07\pm0.02$|$0.11\pm0.01$|$-0.67\pm0.06$|
> |reviews|$0.08\pm0.02$|$0.13\pm0.02$|$-1.23\pm0.11$|
>
> **2. Clarity on Quality Measurement (Q6)**
>
> As stated briefly in Lines 308-310, we use Gemini Pro 1.0 to auto-rate the quality of generated texts. The prompt template & examples of responses are in Appendix A.3 and we provide examples of quality ratings generated by Gemini in Appendix E. Please note that all of the auto-ratings are included in the released GAGLE dataset.
>
> **3. Experimental Support for Claims (Q8)**
>
> In Q8, the experimental results are discussed in Lines 375-385. For instance, when predicting the scoring model, we get an accuracy of 97.0% with and without including the generating model in the predictors. We hope this clarifies your concern.
>
> **4. Organization of Figures**
>
> We apologize for the issues with figure organization and readability. We'll improve them as much as possible within the template constraints, and define PT/IT in Figures 4/5.
>
> **5. Answers to Questions**
> - *Q: Evolution of fractal parameters with the development of LLMs?* This is an insightful question. While our study doesn't provide a longitudinal analysis across model generations, we hypothesize that fractal parameters of LLMs will converge towards those of human language as models improve. One evidence for this is in Figure 3, where more capable models have fractal parameters closer to natural language.
> - *Q. Difference across types of datasets?* We study this question in the paper in Table 3, and have now expanded this analysis with the RAID dataset (see point 1 / Q9 above). We do observe that LLMs seem to be more capable of replicating humans in some domains (e.g. articles) over others (e.g. poetry), as discussed above.
> - *Q. Can memorization affect self-similarity?* We agree this is a very interesting question. Unfortunately, we don't have an answer yet, and will this leave for future research.
>
> **Summary**
>
> Thank you again for your valuable feedback. We believe the additional experiments, clarifications, and revisions significantly strengthen the paper and directly address the concerns raised, and we hope that they resolve them. If you have any remaining concerns, please let us know so we can respond to them during the rebuttal period. Otherwise, we would appreciate it if you consider revising your score.

---

### Decision · Program_Chairs · 2025-05-01

**Decision:**

Accept (poster)

**Comment:**

This paper proposes metrics for comparing machine-generated text against human-written text not only in terms of perplexity but also statistical patterns, particularly self-similarity and long-range dependencies. The authors explore how factors including decoding temperature, instruction-tuning, model size, and prompting may affect such statistics. The findings have implications for detection of machine-generated text. Authors also release a dataset including the texts generated for analysis.

One downside to the current version of the paper is that it looks only at generated articles, rather than texts generated across a wide variety of domains. However, the authors include additional results in their rebuttal on the RAID dataset which covers more domains and models; authors should integrate this into the next version of their paper. Another minor downside is the use of LLMs to generate text quality in Q6 with no human validation. Another concern raised by EJTc is that the authors do not propose a text-detection method based on these metrics; however, I believe the contribution is solid and very interesting by itself without proposing such methods. I also suggest the authors take time to revise the paper's presentation, as mentioned by a couple of reviewers, before publication.